# A theory of synaptic transmission

**Bin Wang, Olga K Dudko***

Department of Physics, University of California, San Diego, La Jolla, United States

**Abstract** Rapid and precise neuronal communication is enabled through a highly synchronous release of signaling molecules neurotransmitters within just milliseconds of the action potential. Yet neurotransmitter release lacks a theoretical framework that is both phenomenologically accurate and mechanistically realistic. Here, we present an analytic theory of the action-potential-triggered neurotransmitter release at the chemical synapse. The theory is demonstrated to be in detailed quantitative agreement with existing data on a wide variety of synapses from electrophysiological recordings *in vivo* and fluorescence experiments *in vitro*. Despite up to ten orders of magnitude of variation in the release rates among the synapses, the theory reveals that synaptic transmission obeys a simple, universal scaling law, which we confirm through a collapse of the data from strikingly diverse synapses onto a single master curve. This universality is complemented by the capacity of the theory to readily extract, through a fit to the data, the kinetic and energetic parameters that uniquely identify each synapse. The theory provides a means to detect cooperativity among the SNARE complexes that mediate vesicle fusion and reveals such cooperativity in several existing data sets. The theory is further applied to establish connections between molecular constituents of synapses and synaptic function. The theory allows competing hypotheses of short-term plasticity to be tested and identifies the regimes where particular mechanisms of synaptic facilitation dominate or, conversely, fail to account for the existing data for the paired-pulse ratio. The derived trade-off relation between the transmission rate and fidelity shows how transmission failure can be controlled by changing the microscopic properties of the vesicle pool and SNARE complexes. The established condition for the maximal synaptic efficacy reveals that no fine tuning is needed for certain synapses to maintain near-optimal transmission. We discuss the limitations of the theory and propose possible routes to extend it. These results provide a quantitative basis for the notion that the molecular-level properties of synapses are crucial determinants of the computational and information-processing functions in synaptic transmission.

**\*For correspondence:**
dudko@physics.ucsd.edu

**Competing interest:** The authors declare that no competing interests exist.

## Editor's evaluation

The present manuscript describes an effort to create a general mathematical model of synaptic neurotransmission. The authors invested great efforts to create a model of the presynaptic mechanisms. This is an exceptionally challenging task and this model makes substantive progress, and highlights where further opportunities lie.

## Introduction

Neurons communicate across special junctions – synapses – using neurotransmitter molecules as a chemical signal (*Südhof, 2013*). Release of neurotransmitters into the synaptic gap occurs when neurotransmitter-loaded vesicles fuse with the membrane of the presynaptic (transmitting) neuron in response to calcium influx during an action potential 'spike'. Synaptic vesicle fusion is remarkably fast and precise: both the duration of fusion and the time between the trigger and fusion initiation are less than a millisecond (*Katz and Miledi, 1965*; *Südhof, 2013*).

The electrical propagation of information along the axon of the presynaptic neuron (the pre-transmission stage) and the response of the postsynaptic neuron to the chemical signal (the

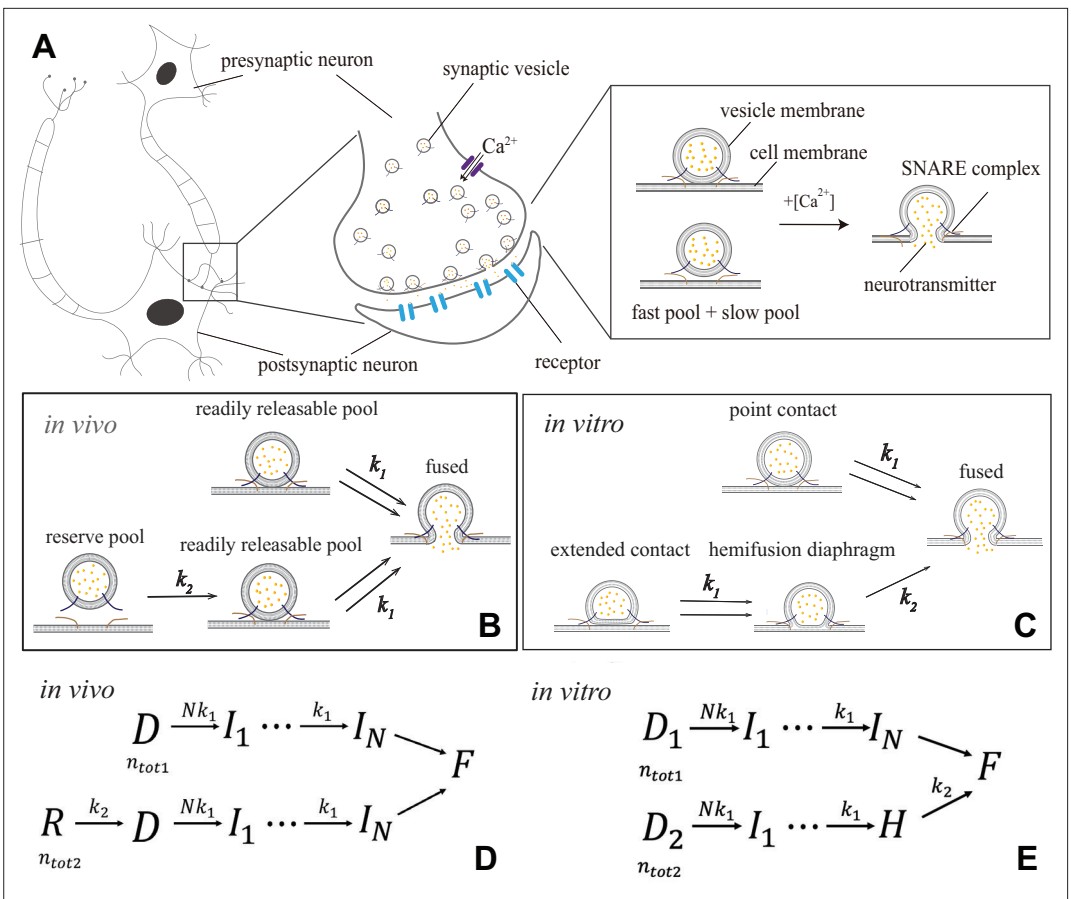

**Figure 1.** Synaptic transmission *in vivo* and *in vitro*. (**A**) Release of neurotransmitters into the synaptic cleft (diameter $\sim 1 - 20\mu m$) occurs when neurotransmitter-loaded vesicles (diameter $\sim 30nm$) fuse with the presynaptic cell membrane in response to $Ca^{2+}$ influx during an action potential. Fusion is facilitated by SNARE protein complexes and proceeds via two parallel pathways that originate in the 'fast' and 'slow' vesicle pools. (**B and C**) Fusion stages *in vivo* and *in vitro*. SNARE conformational transition constitutes the fast step, $k_1$. Vesicle transfer from the reserve pool to the readily releasable pool (RRP) *in vivo* and escape from the hemifusion diaphragm *in vitro* constitute the slow step, $k_2$. (**D and E**) Reaction schemes for (**B**) and (**C**). *In vivo*, state $R$ represents the reserve pool, $D$ the RRP, $I_i$ the state with $i$ independent SNARE assemblies that underwent conformational transitions, $F$ the fused state. *In vitro*, $D_1$ and $D_2$ represent docked vesicles with point- and extended-contact morphologies, $H$ the hemifusion diaphragm. Mathematical equivalence of the reaction schemes *in vivo* and *in vitro* enables the treatment through a unifying theory.

post-transmission stage) have been described by theories that capture phenomenology while connecting to microscopic mechanisms (***Hodgkin and Huxley, 1952***; ***Destexhe et al., 1994***). However, neurotransmitter release, which enables the synaptic transmission itself, lacks a theory that is both phenomenologically accurate and microscopically realistic (***Stevens, 2000***). This void contrasts with detailed experiments, which have revealed the molecular constituents involved. The key to speed and precision of neurotransmitter release is a calcium-triggered conformational transition in SNAREs (soluble N-ethylmaleimide sensitive factor attachment protein receptors) (***Kaeser and Regehr, 2014***; ***Baker and Hughson, 2016***; ***Brunger et al., 2018***). The free energy released during the conformational transition is harnessed by SNAREs to pull the membranes of the vesicle and the cell together, reducing the high kinetic barriers that otherwise hinder fusion. Fusion culminates in the release of neurotransmitters from vesicles into the synaptic cleft (***Figure 1A***).

Here, we present a theory of the action-potential-evoked (AP-evoked) synaptic transmission, which quantitatively reproduces a wide range of data from fluorescence experiments *in vitro* (***Kyoung et al., 2011***; ***Diao et al., 2012***) and electrophysiological experiments *in vivo* (***Barrett and Stevens, 1972***; ***Heidelberger et al., 1994***; ***Schneggenburger and Neher, 2000***; ***Voets, 2000***; ***Beutner et al., 2001***;

*Yang and Gillis, 2004*; *Lou et al., 2005*; *Bollmann et al., 2000*; *Sun et al., 2007*; *Wölfel et al., 2007*; *Sakaba, 2008*; *Kochubey et al., 2009*; *Duncan et al., 2010*; *Miki et al., 2018*; *Fukaya et al., 2021*). The theory yields analytic expressions for measurable quantities, which enables a direct fit to the data. Fitting yields parameters that describe the fusion machinery of each synapse: activation barriers and rates of SNARE conformational transitions at any calcium concentration, the size of vesicle pools, and the number of independent SNARE assemblies necessary for fusion. The analytic expressions explain, quantitatively, the remarkable temporal precision of neurotransmitter release. Perhaps the most striking result of the theory is that the peak release rate as a function of calcium concentration can be written, with proper normalization, in a universal form so that data on different synapses – with release rates spanning ten orders of magnitude – collapse onto a single curve. The established universality is especially remarkable given that these synapses have been known to exhibit strikingly different properties in synaptic transmission due to distinct $Ca^{2+}$-sensors (*Volynski and Krishnakumar, 2018*; *Wolfes and Dean, 2020*) as well as different couplings between the SNAREs and their regulatory proteins or calcium channels (*Kasai et al., 2012*; *Vyleta and Jonas, 2014*; *Stanley, 2016*).

The theory is further applied to relate the properties of neurotransmitter release machinery to the proposed mechanisms of short-term plasticity (*Regehr, 2012*; *Jackman and Regehr, 2017*). A quantitative comparison with experimental data for the paired-pulse ratio enables us to identify the regimes where particular mechanisms of synaptic facilitation dominate or, on the contrary, fail to account for the observed facilitation. We establish how the molecular properties of the transmitter release machinery impose constraints on the tradeoff between transmission rate and fidelity, where fidelity measures the ability of a synapse to generate a desired postsynaptic output in response to a presynaptic input. Finally, we show how the molecular-level properties of synapses determine the optimal synaptic efficacy, or the ability of a synapse to avoid both the transmission errors (lack of a postsynaptic output) and error reads (an output in the absence of an input). Altogether, the theory shows how the key characteristics of synaptic function – plasticity, fidelity, and efficacy – emerge from molecular mechanisms of neurotransmitter release machinery, and thereby provides a mapping from molecular constituents to biological functions in synaptic transmission.

## Results

### Theory

We start from the observation that published data on neurotransmitter release for different synapses and experimental setups (*Barrett and Stevens, 1972*; *Kyoung et al., 2011*; *Diao et al., 2012*; *Heidelberger et al., 1994*; *Schneggenburger and Neher, 2000*; *Lou et al., 2005*; *Bollmann et al., 2000*; *Miki et al., 2018*; *Duncan et al., 2010*; *Kochubey et al., 2009*; *Wölfel et al., 2007*; *Sun et al., 2007*; *Voets, 2000*; *Sakaba, 2008*; *Yang and Gillis, 2004*; *Beutner et al., 2001*; *Fukaya et al., 2021*) can all be encompassed by a unifying kinetic scheme:

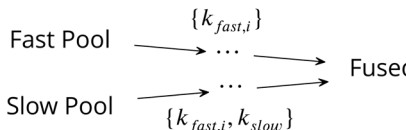

**Scheme 1.** The unifying kinetic scheme for synaptic vesicle fusion.

In this kinetic scheme, synaptic vesicle fusion proceeds through two parallel reaction pathways. Both pathways contain fast steps of rate constants $\{k_{fast,i}\}$. One of the pathways contains an additional, slow, step of rate constant $k_{slow} \ll \{k_{fast,i}\}$. The pathways originate in the 'fast' and 'slow' vesicle pools of sizes $n_{tot1}$ and $n_{tot2}$, respectively. The interpretations of the fast and slow steps as well as the individual states in this unifying kinetic scheme for different experimental setups are summarized in *Figure 1* and detailed below.

In the context of *in vivo* experiments (*Heidelberger et al., 1994*; *Schneggenburger and Neher, 2000*; *Lou et al., 2005*; *Bollmann et al., 2000*; *Miki et al., 2018*; *Duncan et al., 2010*; *Kochubey et al., 2009*; *Wölfel et al., 2007*; *Sun et al., 2007*; *Voets, 2000*; *Sakaba, 2008*; *Yang and Gillis, 2004*; *Beutner et al., 2001*; *Fukaya et al., 2021*), *Scheme 1* concretizes into the kinetic scheme in *Figure 1B and D*. The fast pool represents the readily releasable pool (RRP) comprised of vesicles

that are docked on the presynaptic terminal (state $D$) and fuse readily upon $Ca^{2+}$ influx (**Kaeser and Regehr, 2017**). The slow pool represents the reserve pool (state $R$), which supplies vesicles to the RRP ($R \rightarrow D$) with slow rate $k_2$. Fusion of an RRP vesicle (... $\rightarrow F$) requires $N$ independent SNARE assemblies tethering the vesicle at the cell membrane to concurrently undergo a conformational transition. This transition is $Ca^{2+}$-dependent and involves a single rate-limiting step (**Hui et al., 2005**) of rate constant $k_1([Ca^{2+}])$. Note that $N$ is defined broadly as the critical number of *independent* SNARE assemblies per docked vesicle. Each of the $N$ independent assemblies may consist of a single SNARE or may represent a 'super-assembly' of multiple SNAREs that undergo the conformational transition cooperatively (**Acuna et al., 2014**; **Wang et al., 2014**; **Grushin et al., 2019**; **Tagliatti et al., 2020**; **Zhu et al., 2021**).

In the context of *in vitro* experiments (**Kyoung et al., 2011**; **Diao et al., 2012**), **Scheme 1** becomes the kinetic scheme in **Figure 1C and E**. All vesicles are initially docked (states $D_1$ and $D_2$) but adopt different morphologies (**Figure 1C**) and, consequently, fuse through different pathways (**Gipson et al., 2017**). Vesicles in a point contact with the membrane (state $D_1$) fuse rapidly upon $Ca^{2+}$-triggered SNARE conformational transition, mimicking RRP vesicles *in vivo*. Vesicles in an extended contact (state $D_2$) become trapped in a hemifusion diaphragm intermediate (state $H$), escape from which ($H \rightarrow F$) constitutes the slow step $k_2$.

In all these experiments, the delay due to steps $I_N \rightarrow F$ is negligible compared to both fast and slow steps $k_1$ and $k_2$. Note that a scheme with $N$ independent and concurrent steps of rates $k_1$ (**Figure 1B and C**) is equivalent to a scheme with $N$ sequential steps of rates $Nk_1, (N-1)k_1, ..., k_1$ (**Figure 1D and E**).

Despite the differences in the details of the fusion process *in vivo* and *in vitro* described above, the mathematical equivalence of the corresponding kinetic schemes enables their treatment through a unifying theory. We will assume that the calcium influx is triggered by an action potential that arrives at the presynaptic terminal at $t = 0$. The microsecond timescales (much faster than neurotransmitter release) of the opening of voltage-gated $Ca^{2+}$ channels and diffusion of $Ca^{2+}$ ions across the active zone justify treating the $[Ca^{2+}]$ rising as instantaneous. Since the typical width of $[Ca^{2+}]$ profile is $\sim 1 - 10ms$ (**Bean, 2007**) while most vesicles fuse within $t \sim 100\mu s$ (**Katz and Miledi, 1965**), $[Ca^{2+}]$ can be treated as approximately constant during the fusion process. The theory is thus applicable both for step-like and for spike-like $[Ca^{2+}]$ profiles, as well as for responses to long sequences of spikes of the duration shorter than the timescale $k_2^{-1}$ of RRP replenishment. With the above assumptions, the theory is developed in detail in Appendix 1. Below, we present analytic expressions derived from the theory for the key outputs of the experiments that probe synaptic transmission at the single-synapse level *in vivo* and *in vitro*. These expressions relate experimentally measurable characteristics of synaptic transmission to the molecular parameters of synaptic release machinery, thereby enabling the extraction of these parameters through a fit to experimental data.

An informative characteristic of synaptic transmission is the average release rate. Defined as the average (over an ensemble of repeated stimuli) rate of change in the number of fused vesicles, this quantity is usually reported in experiments on the kinetics of neurotransmitter release (**Schneggenburger and Neher, 2000**; **Bollmann et al., 2000**; **Kyoung et al., 2011**; **Diao et al., 2012**). The rate equations for the kinetic scheme in **Scheme 1** yield the exact solution for the average release rate:

$$\frac{d\langle n(t) \rangle}{dt} = Nk_1 n_{tot1}(1 - e^{-k_1 t})^{N-1}e^{-k_1 t} + Nk_1 k_2 n_{tot2} \sum_{j=0}^{N-1}(-1)^j \binom{N-1}{j}\frac{e^{-k_2 t} - e^{-(j+1)k_1 t}}{(j+1)k_1 - k_2} \tag{1}$$

$$\equiv n_{tot1}p_1(t) + n_{tot2}p_2(t),$$

where $p_{1,2}(t)$ are the probability distributions for the fusion time in the fast and slow pathways, $N$ is the necessary number of independent SNARE assemblies, and $n_{tot1}$ and $n_{tot2}$ are the sizes of the fast and slow pools, respectively. We use the standard notation for binomial coefficient $\binom{N}{m} \equiv \frac{N!}{m!(N-m)!}$.

In practice, the average release rate is obtained from the average cumulative release $\langle n(t) \rangle$, which is defined as the average number of vesicles fused by time $t$ and can be measured directly through electrophysiological recording on the postsynaptic neuron (**Schneggenburger and Neher, 2000**; **Lou et al., 2005**; **Bollmann et al., 2000**; **Wölfel et al., 2007**; **Kochubey et al., 2009**; **Duncan et al., 2010**;

*Miki et al., 2018*) or through fluorescence imaging in synthetic single-vesicle systems (*Kyoung et al., 2011*; *Diao et al., 2012*). Integrating *Equation 1* yields the exact solution for average cumulative release:

$$\langle n(t) \rangle = \int_0^t \frac{d\langle n(t) \rangle}{dt} dt = n_{tot1}(1 - e^{-k_1 t})^N + n_{tot2} \sum_{j=1}^{N} \binom{N}{j}(-1)^{j-1}(1 - \frac{jk_1 e^{-k_2 t} - k_2 e^{-jk_1 t}}{jk_1 - k_2})$$

$$\equiv n_{tot1}F_1(t) + n_{tot2}F_2(t),$$

(2)

where $F_{1,2}(t) = \int_0^t p_{1,2}(t)dt$ are cumulative distributions for the fusion time in the fast and slow pathways and are given by *Equations 15; 18*. In vivo, $F_1(t = T) = (1 - e^{-k_1 T})^N$ is the fusion probability for an RRP vesicle after an action potential of duration $T$ (*Neher, 2015*; *Miki et al., 2018*; *Malagon et al., 2016*). We also derived the full probability distribution of cumulative release by time $t$ (Appendix 1), which, although at present is challenging to measure experimentally, contains more information than the average values in *Equations 1; 2*.

Experiments indicate a separation of timescales, $k_2 \ll k_1$ (*Neher, 2010*; *Kaeser and Regehr, 2014*), which yields useful asymptotic behaviors for AP-evoked neurotransmitter release. At short times, $t \ll 1/k_1, 1/k_2$, the release rate in *Equation 1* is $\frac{d\langle n(t) \rangle}{dt} \sim t^{N-1}$, which can be readily fit to data to extract the number $N$ of independent SNARE assemblies necessary for fusion. At intermediate times, $1/k_1 \ll t \ll 1/k_2$, cumulative release in *Equation 2* becomes $\langle n(t) \rangle \approx n_{tot1} + n_{tot2}k_2 t$, which can be used to determine the RRP size, $n_{tot1}$, by extrapolation (*Neher, 2015*). At long times, $t \sim 1/k_2 \gg 1/k_1$, cumulative release is $\langle n(t) \rangle \approx n_{tot1} + n_{tot2}(1 - e^{-k_2 t})$. As expected, the cumulative release on the intermediate and long timescales is independent of the number $N$ of SNARE assemblies and conformational rate $k_1$ of an assembly as all the fast steps have been completed.

A measure of sensitivity of a synapse to $[Ca^{2+}]$ is the peak release rate (*Schneggenburger and Neher, 2000*; *Lou et al., 2005*; *Bollmann et al., 2000*). The time at which the peak is reached is found from *Equation 1* using $k_2/k_1 \ll 1$: $t_{max} \approx k_1^{-1} \left[ \ln N + (n_{tot2}/n_{tot1}) \left( (N-1)/N^3 \right) (k_2/k_1) \right]$. The peak release rate is then

$$\frac{d\langle n(t) \rangle}{dt} \bigg|_{t=t_{max}} \approx n_{tot1}k_1 \left(1 - \frac{1}{N}\right)^{N-1} \left(1 + \frac{n_{tot2}(N-1)}{n_{tot1}N} \frac{k_2}{k_1}\right).$$

(3)

Now we must establish an explicit form for the calcium-dependence of the rate constant of SNARE conformational transition $k_1([Ca^{2+}])$ in *Equations 1–3*. We utilize the formalism of reaction kinetics (*Kramers, 1940*) generalized to the presence of a bias field (*Dudko et al., 2006*). The formalism treats a conformational transition as thermal escape over a free energy barrier along a reaction coordinate. In the present context, the role of the reaction coordinate is fulfilled by the average number $n_{Ca}$ of $Ca^{2+}$ ions bound to a SNARE assembly at a given $[Ca^{2+}]$, assuming that this average follows the dynamics of the conformational degree of freedom of the SNARE assembly. The generic shape of the free energy profile with a barrier that separates the two conformational states of a SNARE assembly is captured by a cubic polynomial (*Appendix 1—figure 1*). The effect of calcium on the free energy profile is incorporated in analogous manner to the $pH$-dependence of Gibbs free energy of a protein, taking into account the contributions both from the electrostatic energy and from entropy (*Schaefer et al., 1997*; *Zhang and Dudko, 2015*; *Mostafavi et al., 2017*). As shown in Appendix 1, the rate constant of the conformational transition of the SNARE assembly is then

$$k_1([Ca^{2+}]) = k_0 \left(1 - \frac{2}{3}\frac{k_B T n_{Ca}^{\ddagger}}{\Delta G^{\ddagger}} \ln \frac{[Ca^{2+}]}{[Ca^{2+}]_0}\right)^{\frac{1}{2}} \exp\left[\frac{\Delta G^{\ddagger}}{k_B T}\left(1 - \left(1 - \frac{2}{3}\frac{k_B T n_{Ca}^{\ddagger}}{\Delta G^{\ddagger}} \ln \frac{[Ca^{2+}]}{[Ca^{2+}]_0}\right)^{\frac{3}{2}}\right)\right].$$

(4)

Here, $k_0$ is the rate constant and $\Delta G^{\ddagger}$ is the activation barrier for SNARE conformational transition, and $n_{Ca}^{\ddagger}$ is the number of $Ca^{2+}$ ions bound to a SNARE assembly at the transition state, with all three parameters corresponding to a reference calcium concentration $[Ca^{2+}]_0$. *Equation 4* provides a quantitative explanation for the remarkable temporal precision of neurotransmitter release. Indeed, the argument of $\exp(\dots)$ is the change in the barrier height at a given $[Ca^{2+}]$ relative to the reference state. The logarithm of calcium concentration, $\ln[Ca^{2+}]$, is the external force that lowers the barrier (concentrations appear logarithmically because the relevant 'force' on the molecule comes from the

chemical potential, and this helps us to understand how changes in concentration by many orders of magnitude have sensible, graded effects). **Equation 4** shows that the rate $k_1$ is exponentially sensitive to this external force, and so are the release rate (**Equation 1**) and its peak (**Equation 3**) that are both proportional to $k_1$. This exponentially strong sensitivity of the release rate to the force that drives the release explains, quantitatively, the precisely timed character of synaptic release: synaptic fusion machinery turns on rapidly upon $Ca^{2+}$ influx during the action potential and terminates rapidly upon $Ca^{2+}$ depletion (**Sudhof, 2011**).

**Equations 1 and 4** reveal that the number of independent SNARE assemblies $N = 2$ per vesicle provides the optimal balance between stability and temporal precision of release dynamics (**Sinha et al., 2011**). Indeed, at $N = 1$, the release is hypersensitive to sub-millisecond $[Ca^{2+}]$ fluctuations caused by stochastic opening of $Ca^{2+}$ channels (note the high release rate on the sub-millisecond times-cale at $N = 1$ in **Appendix 1—figure 2B**). On the other hand, at $N > 2$, the peak of release following an action potential is delayed. The optimality of $N = 2$ is further supported by the least squares fit of the experimental data (**Kochubey et al., 2009**) to **Equation 1** with different values of $N$: $N = 2$ results in the smallest fitting errors for all calcium concentrations used in the experiment (**Appendix 3—table 1**). However, the theory also reveals that incorporating additional independent SNARE assemblies beyond $N = 2$ may be advantageous for the synapses that require robustness against slower $[Ca^{2+}]$ fluctuations, beyond the sub-millisecond timescale. Indeed, the presynaptic calcium channels are diverse in their intrinsic properties and their interactions with regulatory proteins, and, as the result, generate $[Ca^{2+}]$ fluctuations on a wide range of timescales, $0.5ms - 20ms$ (**Perez-Reyes, 2003**; **Dolphin and Lee, 2020**). The shift of the peak release to longer timescales that accompanies an increase in $N$, as seen in **Appendix 1—figure 2B**, allows the synapses to 'avoid' correspondingly longer-timescale fluctuations in $[Ca^{2+}]$. This point is illustrated further in **Appendix 1—figure 2C**: in synapses with the larger values of $N$, the RRP vesicle release (**Equation 2**) remains low over longer timescales, thereby providing robustness against slower $[Ca^{2+}]$ fluctuations.

In the presence of cooperative interactions among SNAREs that form super-assemblies, $k_1$ in **Equation 4** represents the effective transition rate of a super-assembly. **Appendix 1—figure 2D** illustrates how cooperativity between SNAREs results in a steeper increase of the rate $k_1$ with increasing $[Ca^{2+}]$, and hence in a faster vesicle release. Specifically, every additional SNARE in the super-assembly is esti-mated to increase the release rate by a factor of $\sim 100$ (**Appendix 1—figure 2D**), a result consistent with the previous work (**Manca et al., 2019**) that utilized a different approach.

Now that we have closed-form expressions for the key characteristics of the neurotransmitter release dynamics in hand, we can establish a universal relation for the sensitivity $r$ of a synapse to the strength $c$ of the trigger. Nondimensionalization of **Equations 3 and 4** gives:

$$r = \exp\left[1 - (1 - c)^{\frac{3}{2}}\right],\tag{5}$$

where $c \equiv \frac{2n_{Ca}^{\ddagger}k_BT}{3\Delta G^{\ddagger}}\ln\frac{[Ca^{2+}]}{[Ca^{2+}]_0}$ and $r \equiv \left(\frac{a}{(1-c)^{1/2}}\frac{d\langle n(t)\rangle}{dt}|_{t_{max}}\right)^{\frac{k_BT}{\Delta G^{\ddagger}}}$ are the dimensionless calcium concentra-tion and peak release rate, and $a \equiv \left(1 + \frac{1}{N-1}\right)^{N-1}/(n_{tot1}k_0)$. If the scaling law in **Equation 5** indeed captures universal principles of synaptic transmission, data from different synapses should collapse onto the curve given by **Equation 5**. This prediction is tested in the section 'Application of the theory to experimental data' below.

A postsynaptic response to the action potential events is measured by the peak value of the postsynaptic current (PSC). Using the well-established conductance-based model (**Destexhe et al., 1994**), the average of the peak PSC can be shown to be proportional to the total number of released neurotransmitters (**Katz and Miledi, 1965**):

$$\bar{I}_{PSC} = \gamma \langle n(T)\rangle,\tag{6}$$

where $T$ is the duration of the action potential ($\sim 1ms$) and $\gamma$ depends only on the properties of the postsynaptic neuron. As our focus is on the AP-evoked neurotransmitter release in synaptic transmis-sion, $\gamma$ can be regarded as a constant and postsynaptic receptor saturation can be neglected, so that $\langle n(T)\rangle$ and $\bar{I}_{PSC}$ can be used interchangeably. Note that the presynaptic factors affect the postsynaptic response through $\langle n(t)\rangle$ as described by **Equation 2**, and include $Ca^{2+}$-sensitivity of different $Ca^{2+}$ sensors in SNAREs (captured through $N$, $k_0$, $n_{Ca}^{\ddagger}$ and $\Delta G^{\ddagger}$) and the sizes of both vesicle pools ($n_{tot1}$

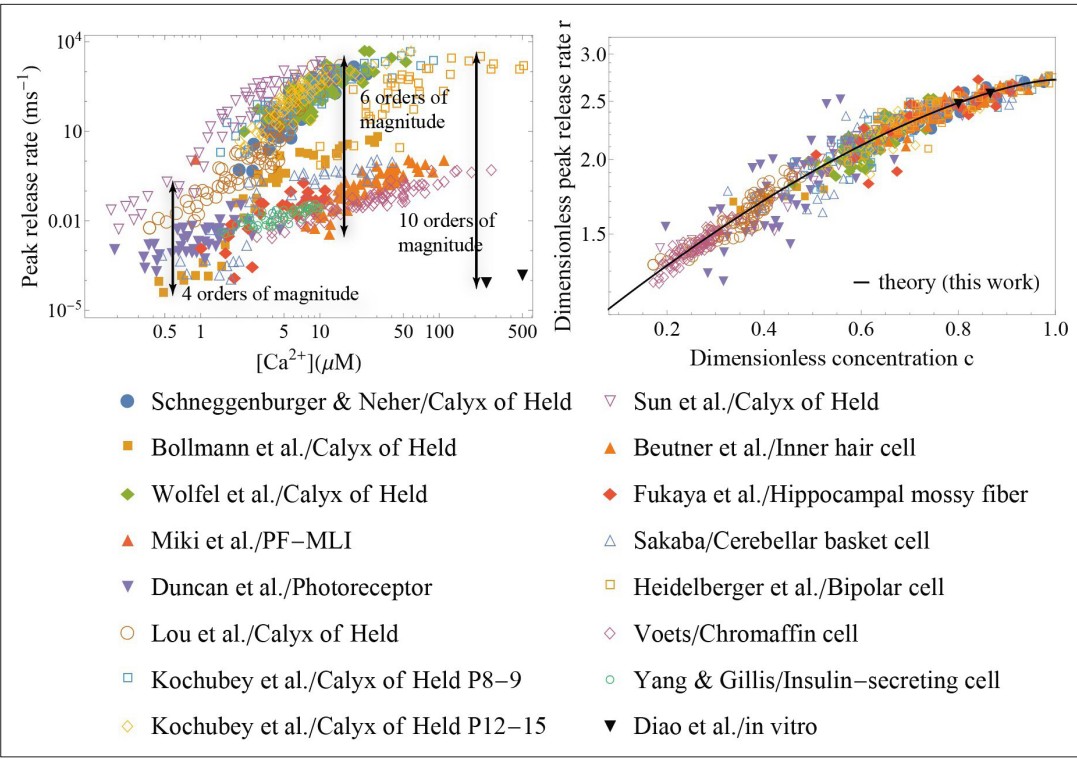

**Figure 2.** Application of the theory to experiments: verifying universality and quantifying specificity. (Left) Measured peak release rate versus calcium concentration for a variety of synapses (*Schneggenburger and Neher, 2000*; *Bollmann et al., 2000*; *Miki et al., 2018*; *Duncan et al., 2010*; *Diao et al., 2012*; *Kochubey et al., 2009*; *Wölfel et al., 2007*; *Lou et al., 2005*; *Heidelberger et al., 1994*; *Beutner et al., 2001*; *Fukaya et al., 2021*; *Sakaba, 2008*; *Voets, 2000*; *Yang and Gillis, 2004*; *Sun et al., 2007*). (Right) The same data as shown on the left, after the peak release rate and calcium concentration have been rescaled. Despite ten orders of magnitude variation in the dynamic range and more than 3 orders of magnitude variation in calcium concentration (left), the data collapse onto a single master curve, *Equation 5* (right). The collapse indicates that the established scaling in *Equation 5* is universal across different synapses. The distinct sets of parameters for each of the synapses (*Appendix 3—table 2*) demonstrate the predictive power of the theory as a tool for extracting the unique properties of individual synapses from experimental data.

and $n_{tot2}$). *Equations 2, 4 and 6* relate the presynaptic action potential to the postsynaptic current response and thus complete our framework for synaptic transmission. Detailed derivations of *Equations 1–6* are given in Appendix 1.

To validate the developed analytic theory, we first compare its predictions to data generated through numerical simulations of the kinetic scheme in *Scheme 1*. A simple least squares fit reliably recovers input parameters of the simulations (*Appendix 2—figure 1* and *Appendix 2—figure 2*). Next, we test the robustness of the theory by comparing it to modified simulations, in which deviations from the assumptions underlying (*Equations 1–4*) are introduced. The modified simulations incorporate (i) the finite-capacity effect of RRP and (ii) heterogeneity of $[Ca^{2+}]$ among different release sites. For deviations within physiological range, the analytic expressions still reliably recover the input parameters (*Appendix 2—figure 3*). Details of the simulations are given in Appendix 2.

## Application of the theory to experimental data

The availability of analytic expressions for measurable quantities enables direct application of the theory to experimental data. A fit of the peak release rate vs. $[Ca^{2+}]$ with *Equations 3 and 4* was performed for a range of synapses to extract a set of parameters $\{\Delta G^{\ddagger}, n_{Ca}^{\ddagger}, k_0\}$ for each synapse. These parameters were then used to rescale the peak release rate and calcium concentration to get the dimensionless quantities $r$ and $c$ that appear in *Equation 5*. We utilized the experimental data from *in vivo* measurements on (i) the calyx of Held, a large synapse (diameter $\sim 20 \mu m$) in the auditory central nervous system, at different developmental stages (*Schneggenburger and Neher, 2000*; *Lou*

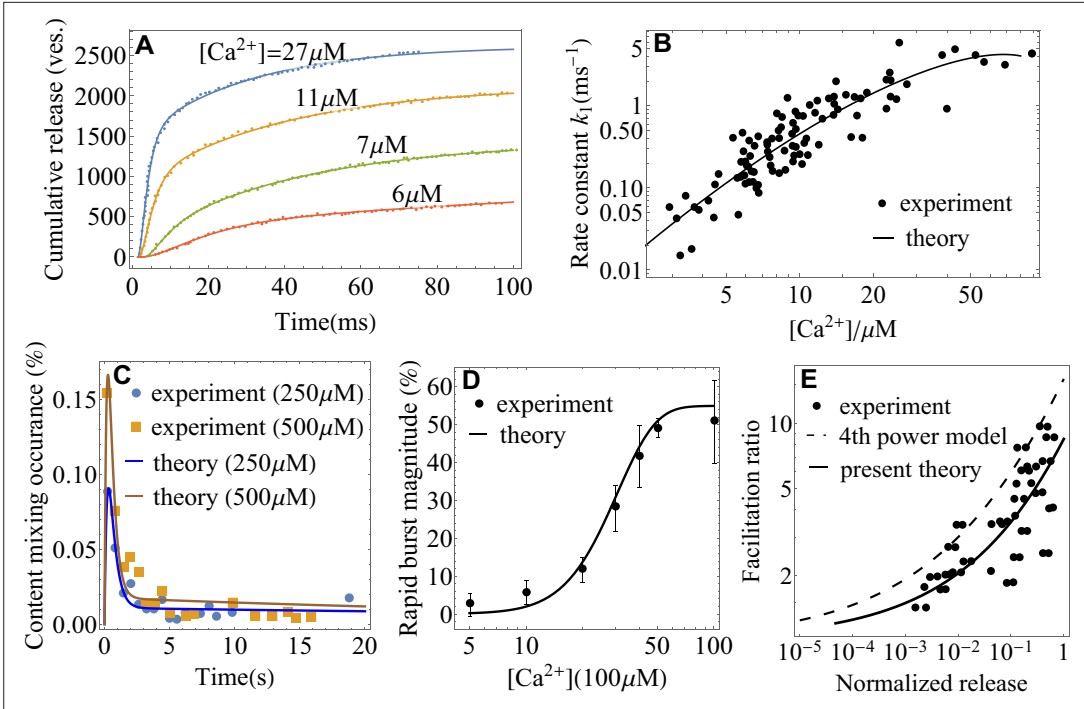

**Figure 3.** Application of the theory to experiments: extracting parameters of synaptic fusion machinery. (**A**) Cumulative release from *in vivo* experiments (*Wölfel et al., 2007*) on the Calyx of Held (symbols) and a fit with *Equation 2* (lines) for different calcium concentrations. (**B**) $Ca^{2+}$-dependent rate constant of SNARE conformational transition from *in vivo* experiments (*Wölfel et al., 2007*) and a fit with *Equation 4*. (**C**) Content mixing occurrence from *in vitro* experiments (*Diao et al., 2012*) and a fit with *Equation 1*. (**D**) Rapid burst magnitude from *in vitro* experiments (*Kyoung et al., 2011*) and *Equation 2*. (**E**) Facilitation as a function of the ratio of residual and control release (as defined in Appendix 3) from the experiment (*Barrett and Stevens, 1972*) on the frog neuromuscular junction (symbols) and from the present theory, *Equation 4* (solid line). The fourth-power model (*Barrett and Stevens, 1972*; *Jackman and Regehr, 2017*) is also shown for comparison (dashed line). Parameters are shown in Appendix 3.

*et al., 2005*; *Bollmann et al., 2000*; *Sun et al., 2007*) (ii) parallel fiber - molecular layer interneuron (PF-MLI), a small synapse ($\sim 1\mu m$) in the cerebellum (*Miki et al., 2018*) (iii) the photoreceptor synapse (*Duncan et al., 2010*) (iv) the inner hair cell (*Beutner et al., 2001*) (v) hippocampal mossy fibre (*Fukaya et al., 2021*) (vi) the cerebellar basket cell (*Sakaba, 2008*) (vii) the retina bipolar cell (*Heidelberger et al., 1994*) (viii) the chromaffin cell (*Voets, 2000*) and (ix) insulin-secreting cell (*Yang and Gillis, 2004*), as well as (x) two *in vitro* measurements (*Kyoung et al., 2011*; *Diao et al., 2012*). *Figure 2* demonstrates that the data from all these synapses collapse on a single curve given by *Equation 5*, consistent with the prediction of the theory. Even though these synapses have been known to have a huge variation in their release rates (up to 10 orders of magnitude) due to the different underlying calcium sensors (*Cohen and Atlas, 2004*; *Kerr et al., 2008*; *Johnson et al., 2010*; *Kochubey et al., 2016*) and different couplings between the SNAREs and their regulatory proteins or calcium channels (*Kasai et al., 2012*; *Vyleta and Jonas, 2014*; *Stanley, 2016*), our theory reveals that all these rates can be brought into a compact, universal form (*Equation 5*). The universal collapse is an indication that synaptic transmission in different synapses is governed by common physical principles and that these principles are captured by the present theory. Variability across synapses on the molecular level is captured through the distinct sets $\{\Delta G^{\ddagger}, n_{Ca}^{\ddagger}, k_0\}$ for each synapse. Notably, the generality of *Equation 5* spans beyond the context of synaptic transmission: the same scaling has appeared in another, seemingly unrelated, instance of biological membrane fusion – infection of a cell by an enveloped virus (*Zhang and Dudko, 2015*).

While a single SNARE can maximally bind $n_{Camax} = 4 - 5$ ions (*Radhakrishnan et al., 2009*; *Brunger et al., 2018*), the fit of some of the experimental data on the calyx of Held analyzed in *Figure 2* produces the transition state values of $n_{Ca}^{\ddagger} > 5$ (*Appendix 3—table 2*). This result indicates that each

SNARE assembly in these synapses is in fact a super-assembly containing two or more cooperative SNAREs. We further note that, since the number of calcium ions bound to a SNARE at the transition state is generally less than the maximum occupancy for the SNARE, $n_{Ca}^{\ddagger} < n_{Camax}$, the synapses with the values of $n_{Ca}^{\ddagger}$ less than but close to five are likely to contain SNARE super-assemblies as well. Interestingly, if we assume that these synapses have the optimal number $N = 2$ of the super-assemblies, and note that the typical rate $k_1 \approx 4ms^{-1}$ at $[Ca^{2+}] = 10\mu M$ would require $\sim 3$ SNAREs per super-assembly (see *Appendix 1—figure 2D*), then the theory estimates that each docked vesicle contains 2 superassemblies × 3 SNAREs/superassembly = 6 SNAREs total. This estimate is consistent with the sixfold symmetric structure recently found using cryoelectron tomography analysis in cultured hippocampal neurons (*Radhakrishnan et al., 2021*).

The utility of the theory as a tool for extracting microscopic parameters of synaptic fusion machinery is further illustrated in *Figure 3A–E*. A fit of *in vivo* data for cumulative release at different levels of $[Ca^{2+}]$ (*Wölfel et al., 2007*) with *Equation 2* extracts the rate of conformational transition of the SNARE assembly, $k_1([Ca^{2+}])$ (*Figure 3A*). A fit of the rate with *Equation 4* extracts activation barrier and rate at reference concentration $[Ca^{2+}]_0$ (*Figure 3B*) of the SNARE assembly. Fits of *in vitro* data (*Kyoung et al., 2011*; *Diao et al., 2012*) with *Equations 1 and 2* are shown in *Figure 3C,D*. In *Figure 3C*, the content mixing occurrence, defined in *Kyoung et al., 2011* as the average release rate normalized by the total number of vesicles, $\frac{d\langle n(t)\rangle}{dt}/(n_{tot1} + n_{tot2})$, is fitted with *Equation 1*. In *Figure 3D*, the rapid burst magnitude, defined in *Diao et al., 2012* as the ratio of the numbers of vesicles fused within the first $1s$ and within $50s$ after calcium trigger, $\langle n(t = 1s)\rangle / \langle n(t = 50s)\rangle$, is fitted with *Equation 2*. *Figure 3E* demonstrates that *Equation 4* yields a significantly better agreement with the experimental data on the frog neuromuscular junction (*Barrett and Stevens, 1972*) than the empirical fourth-power model (*Barrett and Stevens, 1972*; *Jackman and Regehr, 2017*) that was originally used to describe these data. In contrast to the fourth-power model, *Equation 4* accounts for the saturation effect in the dose-response curve of a SNARE assembly at high-calcium concentrations (see, e.g. the nonlinearity in the rate as a function of calcium concentration on the double logarithmic plots in *Figure 2* and *Figure 3B*).

The parameter values extracted from the fits in *Figure 2* and *Figure 3* as well as the least-square fitting algorithm for extracting these parameter values are provided in Appendix 3.

## Linking molecular mechanisms to synaptic function

### Short-term plasticity

Synaptic plasticity, or the ability of synapses to strengthen or weaken over time depending on the history of their activity, underlies learning and memory (*Regehr, 2012*; *Bailey et al., 2015*). A measure of synaptic strength is the peak of the post-synaptic current, which, in turn, is proportional to cumulative release (*Equation 6*). The change in synaptic strength that lasts for less than a minute, known as short-term plasticity (*Regehr, 2012*), can be assessed through the paired-pulse ratio, or the ratio of the cumulative release for two consecutive action potentials of width $T$ (typically $T \sim 1/k_1 \ll 1/k_2$) that are separated by interpulse interval $\tau_{int}$. The weakening of a synapse, or short-term depression, is typically caused by the decrease of RRP size due to depletion of vesicles or inactivation of RRP sites (*Regehr, 2012*). In contrast, the strengthening of synapses, or short-term facilitation, has been attributed to multiple mechanisms (*Jackman and Regehr, 2017*), including the residual calcium hypothesis put forward in the early studies (*Katz and Miledi, 1965*) and recently proposed buffer saturation (*Klingauf and Neher, 1997*; *Neher, 1998*; *Blatow et al., 2003*; *Babai et al., 2014*; *Kawaguchi and Sakaba, 2017*) and syt7-mediated facilitation (*Jackman et al., 2016*; *Turecek et al., 2016*; *Turecek and Regehr, 2018*).

Based on the measured levels of residual calcium concentration of tens to a few hundred nanomolar (*Zucker, 1996*; *Müller et al., 2007*; *Jackman et al., 2016*), *Equation 4* gives an upper bound of $\sim 1.02$ for the paired-pulse ratio. This estimate indicates that the level of residual calcium is far from what is necessary to trigger the large amplitudes of facilitation that are observed in multiple experiments (*Müller et al., 2007*; *Jackman et al., 2016*), in qualitative agreement with the conclusion in *Jackman et al., 2016*.

A more complex version of the residual calcium hypothesis incorporates a facilitation sensor, distinct from the calcium sensor that triggers fusion (usually syt1), which binds to residual $Ca^{2+}$ in between the consecutive action potentials and increases the release probability by interacting with

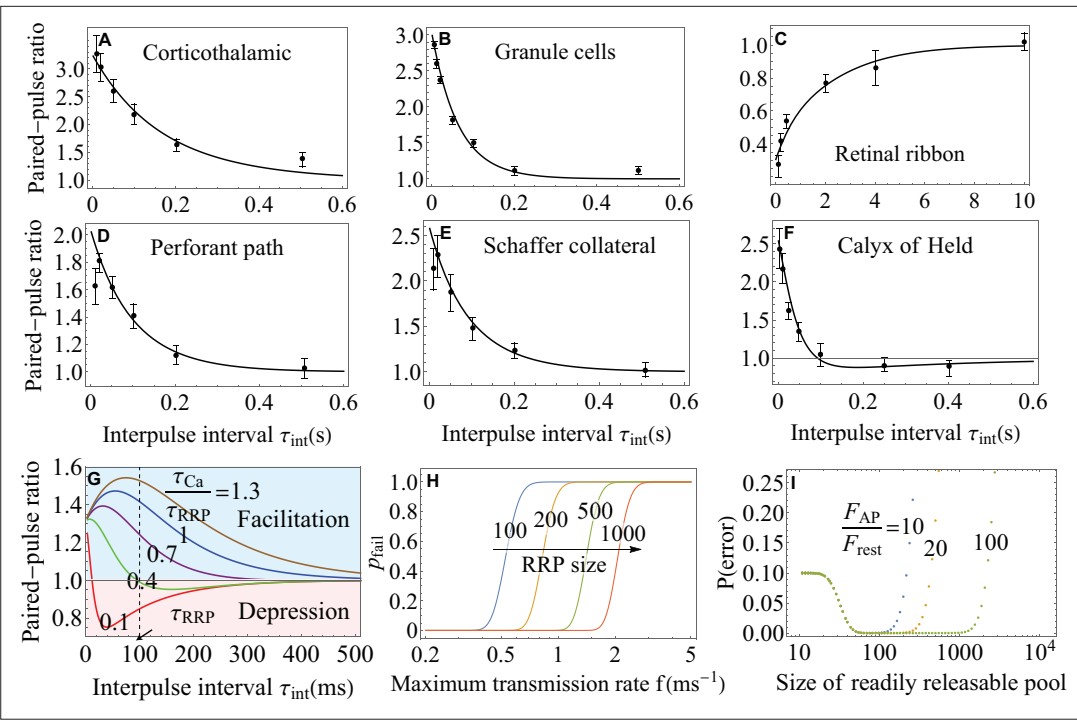

**Figure 4.** Functional implications of the theory. (**A–F**) The paired-pulse ratio as a measure of short-term plasticity from experiments (*Jackman et al., 2016*; *Turecek and Regehr, 2018*; *Müller et al., 2007*) (symbols) and theory (lines) on a variety of synapses. *Equation 7* for the syt7-mediated facilitation captures the data in A-E, and *Equation 8* for the buffer saturation mechanisms captures the data in F over most of the interstimulus timescales probed in the experiments. The theory also identifies the regimes where particular mechanisms fail to account for the observed facilitation (**A, B, D, E**). (**G**) Paired-pulse ratio predicted by *Equation 8*. Synapses exhibit short-term facilitation or depression depending on the relative timescales of the recovery of the readily releasable pool, $\tau_{RRP}$, and dissociation of calcium, $\tau_{Ca}$. A given synapse can exhibit multiple forms of short-term plasticity as the time interval $\tau_{int}$ is varied. (**H**) Trade-off between the maximum transmission rate $f = 1/T$ and fidelity $1 - p_{fail}$ from *Equation 9* for different RRP sizes. (**I**) Synaptic efficacy, $1 - P(error)$, from *Equation 11*. The plateau around the optimal synaptic strength (*Equation 12*) indicates that no fine-tuning is required for near-optimal transmission of large synapses. Higher $Ca^{2+}$-sensitivity $F_{AP}/F_{rest}$ results in broader plateau for near-optimal performance. Parameters are shown in Appendix 3.

the fusion machinery. Synaptotagmin isoform syt7 has been shown to act as a calcium sensor for facilitation for multiple synapses in the brain (*Jackman et al., 2016*; *Chen et al., 2017*; *Turecek and Regehr, 2018*). According to the syt7-mediated facilitation scenario proposed in *Jackman and Regehr, 2017*, let us assume that syt7 is activated by the residual calcium supplied by the first action potential, and this activation transiently increases the rate of conformational transition $k_1([Ca^{2+}])$ of the main calcium sensor (syt1) by a factor of $\sigma > 1$. Let $\tau_{res}$ denote the characteristic timescale on which the new rate $\sigma k_1([Ca^{2+}])$ decays due to the removal of intracellular residual calcium, and let $\tau_{RRP}$ denote the recovery timescale of RRP. Assuming the first-order kinetics of calcium removal and RRP recovery, the change in synaptic strength due to the facilitation sensor mechanism can be obtained from *Equation 2* as (see Appendix 1)

$$\frac{\langle n_f(T)\rangle}{\langle n_i(T)\rangle} \simeq \left[1 - e^{-\frac{\tau_{int}}{\tau_{RRP}}}\left(1 - e^{-k_1([Ca^{2+}])T}\right)^N\right]\left(\frac{1 - e^{-\left(1 + (\sigma-1)e^{-\tau_{int}/\tau_{res}}\right)k_1([Ca^{2+}])T}}{1 - e^{-k_1([Ca^{2+}])T}}\right)^N, \quad (7)$$

where the rate constant $k_1([Ca^{2+}])$ is given by *Equation 4*.

*Equation 7* enables a quantitative comparison with existing experimental data on a variety of synapses where the activation of syt7 by residual calcium has been proposed as the primary mechanism of facilitation (*Luo et al., 2015*; *Jackman et al., 2016*; *Turecek and Regehr, 2018*). *Figure 4 (A–E)* shows that the facilitation sensor model in *Equation 7* successfully explains, with no additional

assumptions, the experimental data on Schaffer collateral, perforant path, corticothalamic, cerebellar granule cell, and retinal ribbon synapses over most of the interstimulus timescales probed in the experiments. At the same time, the comparison between the data and theory shows that the facilitation sensor mechanism alone fails to explain the data on short ($< 10ms$) timescales for Schaffer collateral and perforant path synapses (*Figure 4D and E*) as well as on the timescales $> 500ms$ for corticothalamic and granule cell synapses (*Figure 4A and B*), indicating that other facilitation mechanisms are present and dominate on these timescales. It is worth emphasizing that *Equation 7* provides a quantitative model for the syt7-syt1 mechanism, and enables a quantitative test of the facilitation sensor hypothesis, for different synapses through a single unifying analytic expression. Furthermore, the analytic tractability of the present theory allows the extraction of the parameters that govern the syt7-syt1 mechanism. In particular, the extracted parameters indicate that the syt7-syt1 interaction is strongest ($\sigma = 2.05$) in cerebellar granule cell synapses and weakest ($\sigma = 1.49$) in perforant path synapses. The full list of parameters is included in *Appendix 3—table 3*.

According to the buffer saturation hypothesis of facilitation (*Neher, 1998*; *Babai et al., 2014*), $Ca^{2+}$ buffer captures some of the $Ca^{2+}$ ions supplied by the first action potential thereby decreasing the calcium signal for the sensor that triggers fusion. Upon arrival of the second action potential, the fully or partially saturated buffer no longer constrains calcium concentration so that the signal becomes larger, $[Ca^{2+}]_f > [Ca^{2+}]_i$, and can produce facilitation. Let $\tau_{Ca}$ denote the characteristic timescale on which the increment in calcium concentration decays due to the dissociation of calcium from the buffer. Assuming the first-order kinetics of the calcium concentration increment and RRP vesicle replenishment, the change in synaptic strength due to the buffer saturation mechanism can be obtained from *Equation 2* as (see Appendix 1)

$$\frac{\langle n_f(T) \rangle}{\langle n_i(T) \rangle} \simeq \left[ 1 - e^{-\frac{\tau_{int}}{\tau_{RRP}}} \left( 1 - e^{-k_1([Ca^{2+}]_f)T} \right)^N \right] \left( \frac{1 - e^{-k_1([Ca^{2+}]_f)T}}{1 - e^{-k_1([Ca^{2+}]_i)T}} \right)^N, \tag{8}$$

where the rate constant $k_1([Ca^{2+}])$ is given by *Equation 4*, calcium concentrations during the first and second action potentials are $[Ca^{2+}]_i$ and $[Ca^{2+}]_f = [Ca^{2+}]_i + I_{Ca}e^{-\frac{\tau_{int}}{\tau_{Ca}}}$, and $I_{Ca}$ is the amplitude of the calcium concentration increment due to buffer saturation.

*Figure 4F* shows a quantitative comparison between *Equation 8* and the experimental data on calyx of Held (*Müller et al., 2007*) where buffer saturation has been proposed as the primary mechanism of facilitation (*Babai et al., 2014*; *Luo and Südhof, 2017*). The buffer saturation model in *Equation 8* successfully explained, with no assumptions of additional mechanisms, the data over all interstimulus timescales probed in the experiment, thus supporting buffer saturation as the dominant mechanism in mature calyx of Held synapses. Furthermore, the theory enabled the extraction of the dissociation constant for the local calcium buffer and the rate of RRP replenishment from the experimental data (Appendix 3).

The analytic expressions in *Equations 7; 8* can be used to explore, quantitatively, how short-term plasticity is affected by other factors, such as the interplay between the key timescales and the sensitivity of the underlying calcium sensors. For example, *Equation 8* predicts that, for fixed interpulse interval $\tau_{int}$, the synapse will exhibit short-term facilitation or short-term depression depending on the ratio of the timescales, $\tau_{Ca}/\tau_{RRP}$, as illustrated in *Figure 4G* (*Tank et al., 1995*). *Equation 8* further shows that a given synapse may exhibit multiple forms of short-term plasticity when the interpulse interval $\tau_{int}$ is varied (*Figure 4G*). Such coexistence of multiple forms of plasticity has been observed experimentally (*Regehr, 2012*).

A notable feature of *Equation 8* is the existence of an optimal value of interpulse interval at which facilitation (at large $\tau_{Ca}/\tau_{RRP}$) or depression (at small $\tau_{Ca}/\tau_{RRP}$) of synaptic transmission is maximal (*Figure 4G*). The optimality becomes less pronounced at intermediate values of $\tau_{Ca}/\tau_{RRP}$ where the synapse exhibits both facilitation and depression (note the curve at $\tau_{Ca}/\tau_{RRP} = 0.4$ in *Figure 4G*), suggesting a more subtle role of short-term plasticity in transmitting transient signals (*Tsodyks and Markram, 1997*; *Fuhrmann et al., 2002*).

*Equation 8* further reveals that a higher $Ca^{2+}$-sensitivity of the calcium sensor leads to larger facilitation (*Appendix 1—figure 3B*), indicating that a high $Ca^{2+}$-sensitivity of synaptic fusion machinery is essential for the large dynamic range of short-term plasticity. An example of this relationship can be found in *Rozov et al., 2001*, and it generally applies to the facilitation synapses where the second spike is associated with higher $Ca^{2+}$ influx, as is the case for the residual $Ca^{2+}$ and buffer saturation

mechanisms. Higher $[Ca^{2+}]$ at the second spike causes a larger increase in rate constant $k_1([Ca^{2+}])$ for a more sensitive synapse compared to the corresponding increase in $k_1([Ca^{2+}])$ for a less sensitive synapse, thus triggering more neurotransmitter release.

Finally, *Equation 8* reveals how the molecular-level properties of synapses regulate the facilitation and depression modes of short-term plasticity (*Appendix 1—figure 3C*). The unique properties of neurotransmitter release machinery in different synapses are captured through unique sets of parameters $\{\Delta G^{\ddagger}, n_{Ca}^{\ddagger}, k_0\}$ and $\tau_{Ca}$ for each synapse and can reflect different isoforms of synaptotagmin in SNAREs (*Hui et al., 2005*; *Wolfes and Dean, 2020*), different coupling mechanisms between SNAREs and the scaffolding proteins at release sites (*Vyleta and Jonas, 2014*; *Gramlich and Klyachko, 2019*), or different types of $Ca^{2+}$ buffering proteins present at the presynaptic terminal (*Schwaller, 2020*). These results highlight how the diversity of the molecular machinery for vesicle fusion enables the diverse functions of short-term plasticity (*Südhof, 2013*).

## Transmission rate vs. fidelity

An important characteristic of neuronal communication is fidelity of synaptic transmission. Two measures of fidelity can be considered at the single-synapse level for different types of synapses. The probability of spike transmission is a natural measure of fidelity for giant synapses in sensory systems (*Borst and Soria van Hoeve, 2012*) and neuromuscular junctions. The probability of a postsynaptic voltage/current response, beyond the noise level, to a presynaptic spike is a measure of fidelity for small synapses in the central nervous system (CNS) (*Dobrunz and Stevens, 1997*). The probabilistic nature of release mechanisms at synapses is a common origin of synaptic failure (*Allen and Stevens, 1994*).

Although the two definitions of fidelity apply to different types of synapses, the present theory allows for a unifying treatment of both phenomena. We assume that the desired postsynaptic response – a postsynaptic spike or a postsynaptic current beyond the noise level – is generated only if the number of released vesicles in response to an action potential exceeds some threshold $M$. The value of $M$ depends on the density of postsyanptic receptors and the excitability of the postsynaptic neuron (*Biederer et al., 2017*). For both types of the postsynaptic response, the probability that the synaptic transmission fails is then obtained from the probability $P\{n(t) = m\}$ that $m$ vesicles fuse by time $t$ as

$$p_{fail}\left(T, k_1([Ca^{2+}]), n_{tot1}\right) = \sum_{m=0}^{M} P\{n(T) = m\} \simeq \sum_{m=0}^{M} \binom{n_{tot1}}{m} F_1(T)^m \left(1 - F_1(T)\right)^{n_{tot1}-m}, \quad (9)$$

where $F_1(T) = \left(1 - e^{-k_1([Ca^{2+}])T}\right)^N$. Since the presynaptic neuron cannot generate a second spike during time $[0,T]$, $f \equiv 1/T$ represents the maximum transmission rate. *Equation 9* predicts that a higher maximum transmission rate $f$ results in a higher probability of transmission failure $p_{fail}$ and thus lower fidelity $(1 - p_{fail})$. This trade-off between the maximum rate and fidelity in synaptic transmission is shown in *Figure 4H*. Consistent with intuitive expectation, *Equation 9* further predicts that, for a given maximum transmission rate, the probability of transmission failure can be constrained by the RRP size $n_{tot1}$ and/or SNARE conformational rate $k_1([Ca^{2+}])$ (*Figure 4H*).

*Equation 9* allows us to make a quantitative statement regarding the molecular-level constraints on the fidelity of synapses of different sizes. Faithful spike transmission implies that the threshold $M$ for postsynaptic response is smaller than the average cumulative release, $M < \langle n(T) \rangle = n_{tot1}F_1(T)$. Then, by the Chernoff bound for *Equation 9* (*Vershynin, 2018*),

$$p_{fail}\left(T, k_1([Ca^{2+}]), n_{tot1}\right) \leq e^{-\alpha n_{tot1}\left(\frac{F_1(T)}{\alpha} + \ln\frac{\alpha}{F_1(T)} - 1\right)}, \quad (10)$$

where $\alpha \equiv M/n_{tot1}$. Because both $M$ and $n_{tot1}$ scale linearly with the area of synaptic junctions (*Nakamura et al., 2015*; *Miki et al., 2017*; *Holler et al., 2021*), it is reasonable to assume that $\alpha = M/n_{tot1} < F_1(T)$ is kept at an approximately constant level for different synapses. Since $F_1(T)/\alpha + \ln\left(\alpha/F_1(T)\right) - 1 > 0$, the probability of synaptic failure decreases exponentially as the RRP size $n_{tot1}$ increases. Thus, it follows from *Equation 10* that larger synapses tend to be significantly more reliable, i.e., have an exponentially smaller probability to fail, than smaller synapses in transmitting signals (*Dobrunz and Stevens, 1997*).

## Synaptic efficacy

*Equations 9–10* show that synaptic strength can be increased, that is, failure suppressed, by increasing the RRP size or decreasing the threshold for eliciting postsynaptic response. However, a high synaptic strength increases the probability of an error read, that is, a postsynaptic response generated without a presynaptic spike. We will now establish the condition for the optimal synaptic strength through the balance of probabilities of failure (no postsynaptic response to an action potential) and error read (postsynaptic response in the absence of an action potential). Let $[Ca^{2+}]_{rest}$ and $[Ca^{2+}]_{AP}$ be the calcium concentrations at rest and during the action potential and $q$ the probability of firing an action potential by the presynaptic neuron. The total probability of transmission error is

$$P(error) = \underbrace{qp_{fail}\left(T, k_1([Ca^{2+}]_{AP}), n_{tot1}\right)}_{\text{no postsynaptic response after presynaptic spike}} + \underbrace{(1-q)\left(1 - p_{fail}\left(T, k_1([Ca^{2+}]_{rest}), n_{tot1}\right)\right)}_{\text{postsynaptic response without presynaptic spike}}. \quad (11)$$

Here, we consider the long-term (minutes to days) change in synaptic strength, known as long-term plasticity, through the presynaptic mechanisms and predominantly due to changes in the RRP size, $n_{tot1}$, which has been shown to be regulated through retrograde signaling according to the threshold $M$ on the postsynaptic side (*Haghighi et al., 2003*; *Yang and Calakos, 2013*; *Mayford et al., 2012*; *Bailey et al., 2015*). Synaptic efficacy, $1 - P(error)$, measures the ability of the synapse to faithfully transmit signal. The optimal RRP size is obtained by minimizing the transmission error in *Equation 11*:

$$n_{tot1}^* = \left\lceil M\left(1 + \frac{\ln\frac{F_{AP}}{F_{rest}}}{\ln\frac{1-F_{rest}}{1-F_{AP}}}\right) + \frac{\ln\frac{q}{1-q}}{\ln\frac{1-F_{rest}}{1-F_{AP}}}\right\rceil, \quad (12)$$

where $\lceil x \rceil$ denotes ceiling, i.e. the smallest integer greater than or equal to $x$, and $F_{AP} = \left(1 - e^{-k_1([Ca^{2+}]_{AP})T}\right)^N$ and $F_{rest} = \left(1 - e^{-k_1([Ca^{2+}]_{rest})T}\right)^N$ are the fusion probabilities during the action potential and at rest. *Equation 12* predicts that, as the synapse is stimulated more frequently ($q$ increases), a larger RRP size is needed for the optimal performance, that is, the optimal RRP size and hence the optimal synaptic strength increase, resulting in long-term potentiation on the presynaptic side.

How far can the RRP size deviate from its optimal value without a significant loss of synaptic efficacy? The range of RRP sizes for near-optimal performance can be estimated through the Chernoff bound for *Equation 11*:

$$P(error) \leq qe^{-\alpha n_{tot1}\left(\frac{F_{AP}}{\alpha} + \ln\frac{\alpha}{F_{AP}} - 1\right)} + (1-q)e^{-(1-\alpha)n_{tot1}\left(\frac{1-\alpha}{F_{rest}} + \ln\frac{F_{rest}}{1-\alpha} - 1\right)}. \quad (13)$$

According to *Equation 13*, for synapses that are large ($n_{tot1} \gg 1$) and sufficiently sensitive to $Ca^{2+}$ ($F_{AP}/F_{rest} \gg 1$), the error probability is exponentially small and thus insensitive to changes in the RRP size $n_{tot1}$. Specifically, the near-optimal range for $n_{tot1}$ can be estimated from $F_{rest} \lesssim \alpha \lesssim F_{AP}$ to be $M/F_{AP} \lesssim n_{tot1} \lesssim M/F_{rest}$. Since $1/F_{AP} \ll 1/F_{rest}$, this range is broad, indicating that large synapses do not need to fine-tune their RRP size in order to maintain near-optimal transmission. This robustness in synaptic transmission is illustrated in *Figure 4I*.

## Discussion

The capacity of neurons to transmit information through synapses rapidly and precisely is the key to our ability to feel, think, or perform actions. Despite the challenge posed for experimental studies by the ultrashort timescale of synaptic transmission, a number of recent experiments *in vivo* (*Heidelberger et al., 1994*; *Schneggenburger and Neher, 2000*; *Lou et al., 2005*; *Bollmann et al., 2000*; *Miki et al., 2018*; *Duncan et al., 2010*; *Kochubey et al., 2009*; *Wölfel et al., 2007*; *Sun et al., 2007*; *Voets, 2000*; *Sakaba, 2008*; *Yang and Gillis, 2004*; *Beutner et al., 2001*; *Fukaya et al., 2021*) and in reconstituted systems (*Kyoung et al., 2011*; *Diao et al., 2012*) demonstrated the ability to probe the kinetics of synaptic transmission at the single-synapse level. By design, these experiments generate pre-averaged data that encode unprecedented information on the molecular mechanisms of synaptic function; this information is lost once the data are averaged over multiple synaptic inputs. However, decoding this information requires a quantitative framework that would link the quantities that are

measured in the experiments to the microscopic parameters of the synaptic release machinery. Here, we presented a statistical-mechanical theory that establishes these links.

## Analytic theory for synaptic transmission

Our theory casts the synaptic fusion scenarios observed in different experimental setups into a unifying kinetic scheme. Each step in this scheme has its mechanistic origin in the context of a given experimental setup. In the context of *in vivo* experiments, distinct vesicle pool dynamics are taken into account (*Alabi and Tsien, 2012*; *Yang and Calakos, 2013*; *Kaeser and Regehr, 2017*) to quantitatively explain the different timescales observed in the vesicle release dynamics (*Kaeser and Regehr, 2014*; *Neher, 2017*; *Rozov et al., 2019*): vesicles from the readily releasable pool (RRP) fuse readily once the critical number of SNARE complexes undergo conformational transitions upon $Ca^{2+}$ influx (fast step), while the reserve pool supplies vesicles to the RRP (slow step). In the context of *in vitro* experiments, different timescales in vesicle release dynamics are due to the observed distinct states of docked vesicles (*Diao et al., 2012*; *Kweon et al., 2017*; *Gipson et al., 2017*): the vesicles that are in a point contact with the membrane fuse readily upon $Ca^{2+}$-triggered SNARE conformational transition (fast step), while the vesicles that are in an extended contact become trapped in a hemifusion diaphragm state prior to fusing with the membrane (slow step). Although the presence of these distinct docked states *in vivo* is still under debate (*Neher and Brose, 2018*; *Brunger et al., 2018*), the realization that both of the fusion scenarios can in fact be mapped onto the same kinetic scheme allowed us to capture these scenarios through a unifying analytical theory. The fact that each fusion step in the kinetic scheme has a concrete mechanistic interpretation makes the theory directly predictive in both *in vitro* and *in vivo* experiments.

The calculated measurable quantities include: (i) cumulative release, which quantifies the number of vesicles fused during a given time interval following the action potential, (ii) temporal profile of the release rate, which measures the rate of change in the number of fused vesicles, (iii) peak release rate, which is a measure of sensitivity of a synapse to the trigger, and (iv) the calcium-dependent rate of SNARE conformational change. A least-squares fit of data with these expressions yields the activation energy barrier and rate constant for SNARE conformational change at any calcium concentration of interest, the critical number of SNARE assemblies necessary for fusion, and the sizes of the readily releasable and reserve vesicle pools.

Since the pioneering efforts to quantitatively describe synaptic transmission (*Katz and Miledi, 1965*; *Dodge and Rahamimoff, 1967*), multiple models have been developed, such as the "five-site" model and its variants (*Klingauf and Neher, 1997*; *Schneggenburger and Neher, 2000*; *Bollmann et al., 2000*; *Sakaba, 2008*; *Kochubey et al., 2009*; *Voets, 2000*; *Beutner et al., 2001*) and the dual $Ca^{2+}$ sensor models (*Sun et al., 2007*; *Pan and Zucker, 2009*). These models provided valuable insights into the action-potential-triggered neurotransmitter release in the particular synapses for which they have been developed. However, the existing models have at least two fundamental limitations. First, the system-specific nature of these models limits their applicability beyond specific systems, so that the description of synapses with different calcium-response properties requires the use of different models. In contrast, the present theory is applicable to a wide variety of synaptic types, despite the differences in their fusion pathways, different calcium sensors that they implement (*Wolfes and Dean, 2020*) and different couplings between their regulatory proteins (*Kasai et al., 2012*; *Gramlich and Klyachko, 2019*). Indeed, recent experiments have suggested that the calcium-response properties of synapses are much more diverse than had been thought previously (*Özçete and Moser, 2021*; *Gómez-Casati and Goutman, 2021*; *Schröder et al., 2021*). Second, the existing models did not produce analytic expressions for the key observables that emerge from the experiments, which limits the predictive value of these models, their utility in extracting information from the experiments, and their ability to reveal the organizational principles of synaptic transmission. In contrast, the present theory yields analytic expressions for the key measurable characteristics of synaptic transmission, which can be used as the tools for extracting the essential molecular parameters of synaptic release machinery through a direct fit to experimental data. Thus, the predictive power of the present theory in describing synaptic transmission in vastly different synapses through a unifying framework is complemented by the utility of the theory as a tool for extracting the molecular parameters that uniquely identify each synapse. The theory links the underlying molecular diversity of synapses to the distinct phenomenological responses observed in experiments, and thus

constitutes a constructive step toward a yet more complete description of synaptic transmission (*Stevens, 2000*).

The theory presented here has several limitations. (i) Our treatment of the vesicle replenishment rate $k_2$ as a constant is justified by its weak sensitivity to the intracellular calcium concentration compared to that of $k_1$, as found in recent experiments (*Wölfel et al., 2007*; *Kobbersmed et al., 2020*; *Kusick et al., 2020*). However, in the response to a tetanic stimulus, where the asynchronous component of the release becomes dominant, the calcium-dependence of $k_2$ may no longer be negligible. Explicitly taking this dependence into account in the theory will allow the extraction of the parameters for post-tetanic potentiation. (ii) The theory describes synaptic transmission at the level of a single synapse. The theory was motivated by the experimental setups that are capable of probing synaptic transmission at the single-synapse level and is applicable both to giant synapses with many active zones in sensory systems (*Borst and Soria van Hoeve, 2012*) and to small synapses with few active zones in the brain (*Harris and Weinberg, 2012*; *Figure 2*). However, a postsynaptic neuron usually receives inputs from many synaptic connections, and the cellular response is an integration of these inputs. The analytic expressions presented above can be directly applied to integrated multiple synaptic inputs in the cases where the molecular features of the presynaptic and postsynaptic sides are similar across the synapses, for example when the synapses originate from the same axon and connect to nearby dendritic regions of a postsynaptic neuron (*Branco and Staras, 2009*). The theory can be extended to account for the effects of heterogeneous presynaptic inputs by applying the derived expressions to each synapse separately with an individual set of microscopic parameters for each synapse. (iii) We treated the postsynaptic response as a linear function of neurotransmitter release (*Equation 6*). Such a treatment is sufficient to explain the experimental data on neurotransmitter release (*Figure 2* and *Figure 3*) and the paired-pulse ratio in short-term plasticity (*Figure 4*) through a single, unifying framework. The theory can be extended to account for the nonlinearity of postsyanptic response by replacing *Equation 6* with a relevant nonlinear function. Such an extension will enable the elucidation of the details of active dendritic integration of heterogeneous synaptic inputs.

## $Ca^{2+}$-dependent rate of SNARE conformational transition from Kramers theory

The rate-limiting step in the initiation of fusion of the synaptic vesicles that are docked on the presynaptic membrane is the conformational transition of the critical number of SNARE assemblies tethering the vesicles to the membrane (*Kaeser and Regehr, 2014*). We derived the calcium-dependence of the SNARE conformational rate from the classical reaction-rate theory (*Kramers, 1940*) which we generalized to include an external trigger – calcium influx. The resulting analytic expression reveals that the SNARE conformational rate, and hence both the vesicle release rate and the peak of the release rate, are all exponentially sensitive to the force that drives the release – the logarithm of calcium concentration (the logarithmic scale arises naturally due to the several-orders-of-magnitude changes in $[Ca^{2+}]$ following an action potential). This result provides a quantitative explanation for the remarkable synchrony of synaptic vesicle fusion: since the rising of calcium concentration after an action potential occurs on a microsecond timescale and is thus essentially instantaneous on the timescale of synaptic release, the exponential sensitivity of the release rate to this nearly instantaneous trigger ensures an ultra-rapid initiation of vesicle fusion upon calcium influx. Likewise, the exponential sensitivity of the release rate to the trigger ensures that the fusion process terminates rapidly upon calcium depletion (*Brunger et al., 2018*).

Unlike the conventional model that postulates $k_{SNARE} \sim [Ca^{2+}]^4$, our expression in *Equation 4* naturally accounts for the saturation effect at intermediate-to-high calcium concentrations (*Figure 2* and *Figure 3B*), which is the typical regime for the AP-evoked neurotransmitter release. In the limit of $\ln\left(\frac{[Ca^{2+}]}{[Ca^{2+}]_0}\right) \ll 1$, the asymptotic expansion of *Equation 4* recovers the power-law $k_1([Ca^{2+}]) \sim \left(\frac{[Ca^{2+}]}{[Ca^{2+}]_0}\right)^{n_{Ca}^{\ddagger}}$, indicating that a power-law description is only valid for the initial rise of the release rate in response to calcium. Moreover, the power exponent $n_{Ca}^{\ddagger}$ is not a universal number (e.g., 4) but rather it depends on the details of the molecular constitutes of the SNARE complexes in a given synapse, such as different calcium sensors from synaptotagmin family (*Wolfes and Dean, 2020*) and different couplings between the regulatory proteins (*Kasai et al., 2012*; *Stanley, 2016*) (*Appendix 3—table 2*).

## Critical number of SNARE assemblies for vesicle fusion

The theory further reveals how the kinetics of vesicle fusion are affected by the critical number of SNARE assemblies per vesicle. Given the lack of general consensus (*Südhof, 2013*; *Sinha et al., 2011*; *van den Bogaart and Jahn, 2011*; *Brunger et al., 2018*), the theory makes no assumptions about the specific number of SNAREs necessary for fusion, and the number itself can serve as a free parameter when sufficient data is available for a robust fit. Interestingly, however, the theory suggests that $N = 2$ independent SNARE assemblies per vesicle provide the optimal balance between stability and precision of release dynamics. Indeed, on the one hand, in the presence of a single SNARE, the high values and an exponentially-steep temporal dependence of the release rate make the rate highly sensitive to sub-millisecond calcium fluctuations, and thus a very fine tuning of the calcium concentration would be necessary to prevent instability of the fusion process. On the other hand, the values of $N$ greater than two lead to longer delays in the peak of the release rate following an action potential, thus reducing the temporal precision of vesicle release. Furthermore, a least-squares fit of the release rate from the experiment (*Kochubey et al., 2009*) with the theory at different values of $N$ reveals that $N = 2$ indeed results in the smallest fitting errors for all calcium concentrations. The generality of this result can be determined as more data on the release dynamics for different synapses becomes available. The theory further suggests that incorporating additional SNARE assemblies beyond $N = 2$ may be advantageous for the synapses that require robustness against slow $[Ca^{2+}]$ fluctuations (*Mohrmann et al., 2010*).

The theory can account for cooperativity between SNAREs and can help identify the presence of SNARE super-assemblies (*Radhakrishnan et al., 2021*). Mathematically, this is due to the formal definition of the parameter $N$ as the number of *independent* reaction steps needed for fusion. Each such step may represent a conformational transition of a single SNARE (in the absence of cooperativity) or of a multi-SNARE super-assembly (i.e. an assembly of cooperative SNAREs). The calcium-dependent release rate $k_1([Ca^{2+}])$ in *Equation 4* should be regarded as the transition rate for each independent SNARE unit: if individual SNAREs act independently, $k_1$ is the transition rate of a single SNARE and $N$ is the number of SNAREs per vesicle; alternatively, if multiple SNAREs undergo conformational change cooperatively, $k_1$ is the effective transition rate of a super-assembly and $N$ is the number of the super-assemblies per vesicle. The theory allows one to detect the presence of super-assemblies through the values of $n_{Ca}^{\ddagger}$ extracted from the fit: if $n_{Ca}^{\ddagger}$ is larger than the number of $Ca^{2+}$ binding sites for a single SNARE ($n_{Ca_{max}} = 5$), it is an indication that a super-assembly of more than one SNARE is present. Applying this criterion produced evidence for the presence of such super-assemblies in several experimental data sets analyzed in this study. More detailed measurements will be needed to get a more direct estimate of the number of SNAREs in each super-assembly. One approach is to perform single-molecule measurements of the kinetics of a single SNARE under different calcium concentrations, fit the resulting rate $k_1([Ca^{2+}])$ with *Equation 4* to extract the value of $n_{Ca}^{\ddagger}$ for the single SNARE, and to compare this value with the value of $n_{Ca}^{\ddagger}$ extracted from a fit with *Equation 4* of *in vivo* data to get an estimate for the number of SNAREs in each super-assembly. The theory suggests that synapses may have more than 2 SNAREs while still having the optimal value of $N = 2$: the SNAREs in these synapses may form $N = 2$ super-assemblies, each comprising more than one SNARE.

## Universality vs. specificity in synaptic transmission

The fact that, in all chemical synapses, the delay time from the action potential triggering to vesicle fusion is determined by the conformational transition of preassembled SNARE complexes, and that the conformational transition itself occurs through a single rate-limited step, suggests possible universality in synaptic transmission across different synapses despite their structural and kinetic diversity. Our theory made this intuition precise through a non-dimensionalized scaling relationship between the peak release rate and calcium concentration (*Equation 5*), which is predicted to hold for all synapses irrespective of their variability on the molecular level. In statistical physics, the significance of universality is that it indicates that the observed phenomenon (here, synaptic transmission) realized in different systems is governed by common physical principles that transcend the details of particular systems.

The universal relation was tested using published experimental data on a variety of synapses, including *in vivo* measurements on the calyx of Held studied at different developmental stages, parallel fiber-molecular layer interneuron, the photoreceptor synapse, the inner hair cell, the hippocampal

mossy fiber, the cerebellar basket cell, the retina bipolar cell, the chromaffin cell, and the insulin-secreting cell, as well as a reconstituted system. Despite more than an order of magnitude difference in the size of these synapses, ten orders of magnitude variation in the dynamic range of synaptic preparations, and a range of calcium concentrations spanning more than three orders of magnitude, the data for the sensitivity of the synapses to the trigger collapsed onto a universal curve, as predicted by the theory. The collapse serves as an evidence that the established scaling of the normalized peak release $r$ with calcium concentration $c$, $r = \exp\left[1 - (1-c)^{3/2}\right]$, is indeed universal across different synapses. At the same time, the unique properties of specific synapses are captured by the theory through the distinct sets of parameters of their molecular machinery: the critical number of SNAREs, their kinetic and energetic characteristics, and the sizes of the vesicle pools. The practical value of the theory as a tool for extracting microscopic parameters of synapses was further illustrated by fitting *in vivo* and *in vitro* data for cumulative release and for the average release rate at different calcium concentrations. Compared to previous work based on phenomenological formulas (**Kochubey et al., 2011**), the mechanistic nature of the present theory allows it to be further tested by independently measuring the microscopic parameters of synaptic fusion machinery $\{\Delta G^{\ddagger}, n_{Ca}^{\ddagger}, k_0\}$ through single-molecule experiments (**Gao et al., 2012**; **Oelkers et al., 2016**) and the postsynaptic response through electrophysiological recording experiments.

## From molecular mechanisms to synaptic function

We applied the theory to establish quantitative connections between the molecular constituents of synapses and synaptic function. Previous quantitative analyses of the experimental data on short-term plasticity were based either on the empirical fourth-power model (**Magleby, 1973**) or on custom models that are only applicable to specific calcium sensors (**Klingauf and Neher, 1997**; **Pan and Zucker, 2009**). The present theory provides analytic expressions for the paired-pulse ratio (**Equations 7 and 8**) that can be directly compared with the existing experimental data on a variety of synapses (**Müller et al., 2007**; **Jackman et al., 2016**; **Turecek and Regehr, 2018**). As an illustration of the functional implications of the theory, we tested two prevalent hypotheses for the mechanism of synaptic facilitation: syt7-mediated facilitation and buffer saturation. Our results support the facilitation sensor (syt7) as the dominant mechanism for short-term facilitation over most of the interstimulus timescales in the Schaffer collateral, perforant path, corticothalamic, cerebellar granule cell, and retinal ribbon synapses, in agreement with (**Jackman et al., 2016**; **Turecek and Regehr, 2018**) but contrary to an earlier study that has suggested other mechanisms for facilitation in the retinal ribbon synapse (**Luo et al., 2015**). The theory also identified the regimes where the proposed mechanisms fail to account for the observed facilitation. In particular, the syt7-mediated facilitation cannot explain data at $> 500ms$ for cerebellar granule cell and corticothalamic cell synapses, plausibly due to a dominant effect of buffer saturation in this regime (**Kawaguchi and Sakaba, 2017**; **Rebola et al., 2019**). Likewise, the failure of the syt7 mechanism to explain facilitation in Schaffer collateral and perforant path synapses at $< 10ms$ suggests a significant contribution of the calcium current facilitation in this regime (**Nanou et al., 2016**). We limited the discussion of the short-term plasticity to the two mechanisms of synaptic facilitation and to the data on the paired-pulse ratio as illustrative examples, but other mechanisms can be explored in an analogous manner. For example, spike-broadening effects (**Cho et al., 2020**) and calcium-dependent vesicle recycling (**Marks and McMahon, 1998**) can be incorporated into the theory by introducing variations in $T$ and $k_2$, respectively.

The theory enabled a quantitative description of how short-term facilitation, depression, or coexistence of multiple forms of plasticity in a given synapse emerge from the interplay between the molecular-scale factors such as the timescales of RRP recovery and buffer dissociation as well as the sensitivity of $Ca^{2+}$-sensors. In contrast to phenomenological models of short-term plasticity (**Tsodyks and Markram, 1997**; **Fuhrmann et al., 2002**; **Rosenbaum et al., 2012**), the mechanistic nature of the present theory reveals the connection between temporal filtering of synaptic transmission and calcium-sensitivity of synaptic fusion machinery, and shows how diverse short-term facilitation/depression modes emerge from the diversity of the molecular constituents.

While one intuitively expects that there must be a tradeoff between the maximum transmission rate and fidelity of a synapse, our theory turns this intuition into a quantitative relation (**Equation 10**). The trade-off relation shows how transmission failure can be controlled by changing the microscopic properties of the vesicle pool and SNARE complexes. The relation further shows that the

probability of synaptic failure decreases exponentially with increasing the synapse size, which makes large synapses significantly more reliable than small synapses in transmitting signals. Furthermore, the established condition for the maximal synaptic efficacy (*Equation 12*) reveals that, for large synapses, the parameter range of near-optimal performance is broad, indicating that no fine tuning is needed for these synapses to maintain near-optimal transmission (*Figure 4I*). This finding may also be relevant to small synapses: although a small size of their individual RRPs makes them less reliable in transmitting signals individually, trans-synaptic interactions that couple many nearby small synapses may result in a large 'effective' RRP (*Bailey et al., 2015*) and thus enable small synapses to collectively maintain near-optimal transmission without fine-tuning. Altogether, the results of the theory provide a quantitative basis for the notion that the molecular-level properties of synapses are not merely details but are crucial determinants of the computational and information-processing synaptic functions (*Südhof, 2013*). Limitations of the theory and possible routes to generalize it to other settings are also discussed.

Other biological processes, including infection by enveloped viruses, fertilization, skeletal muscle formation, carcinogenesis, intracellular trafficking, and secretion, have features that are very similar to those in synaptic transmission, despite the bewildering number and structural diversity of the molecular constituents involved (*Harrison, 2017*). These processes occur through membrane fusion that (i) requires overcoming high energy barriers, (ii) is controlled by proteins that undergo a conformational transition once exposed to a trigger, (iii) is facilitated by the energy released during this transition, which reduces the fusion timescale by orders of magnitude. The theory presented here can be generalized to encompass these processes while engaging with the diversity of specific systems. The mapping from molecular mechanisms to cellular function, provided by the present theory, is a step toward a more complete framework that would bridge mechanisms with function at the multicellular scale (e.g. neuronal circuits and tissues) and further at the scale of an organism.

## Materials and methods

Details of the derivations for analytical results, simulation methods and fitting procedures are described in Appendices 1, 2 and 3.

## Acknowledgements

We are grateful to Bill Bialek for insightful discussions, and Tynan Kennedy and Palka Puri for critical comments on the manuscript. This research was supported by the National Science Foundation Grant No. MCB-1411884.

## Additional information

### Funding

| Funder | Grant reference number | Author |
| --- | --- | --- |
| National Science Foundation | MCB-1411884 | Olga K Dudko |

The funders had no role in study design, data collection and interpretation, or the decision to submit the work for publication.

### Author contributions

Bin Wang, Conceptualization, Data curation, Formal analysis, Investigation, Methodology, Software, Validation, Visualization, Writing – original draft, Writing – review and editing; Olga K Dudko, Conceptualization, Data curation, Formal analysis, Funding acquisition, Investigation, Methodology, Project administration, Resources, Software, Supervision, Validation, Visualization, Writing – original draft, Writing – review and editing

## Author ORCIDs

Bin Wang http://orcid.org/0000-0003-3390-8210
Olga K Dudko http://orcid.org/0000-0001-8944-8538

## Decision letter and Author response

Decision letter https://doi.org/10.7554/eLife.73585.sa1
Author response https://doi.org/10.7554/eLife.73585.sa2

---

## Additional files

### Supplementary files

• Transparent reporting form

### Data availability

This paper is a theoretical study, no data have been generated. Fitting code is provided in Appendix.

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

# Appendix 1

## Derivations for analytic results

### Synaptic fusion dynamics and fusion times

As discussed in the main text, even though the reaction schemes for calcium-triggered synaptic release *in vivo* and *in vitro* differ in the interpretation of the individual states, their mathematical equivalence enables the treatment through a unified theory. Independent of the details of the experimental situation, synaptic fusion generally involves a "fast" pool and a "slow" pool of synaptic vesicles (***Figure 1*** in the main text).

We assume that fusion of a vesicle from the fast pool requires a conformational change of each of the $N$ SNARE assemblies attached to it and that the SNAREs undergo their conformational changes independently with the corresponding times $\tau_1, \tau_2$. Since the conformational change of a SNARE assembly is dominated by a single barrier (***Hui et al., 2005***), $\tau_i$ satisfies the exponential distribution

$$P(\tau_i) = k_1 e^{-k_1 \tau_i} \qquad \text{for} \qquad i = 1, 2 \cdots N, \tag{14}$$

where $k_1 \equiv k_1([Ca^{2+}])$ is the calcium-dependent rate constant for the conformational change of each SNARE assembly. The fusion time for a vesicle in the fast pool, $T_1$, is therefore determined by the largest value of $\tau_i$. For an action potential, and the rise of calcium concentration, triggered at time $t = 0$, the probability that fusion has occurred by time $t$ is

$$
\begin{aligned}
F_1(t) \equiv \mathbb{P}\left\{T_1 \le t\right\} \quad &= \mathbb{P}\left\{\max(\tau_1, \tau_2 \cdots \tau_N) \le t\right\} \\
&== \prod_{i=1}^{N} \mathbb{P}\left\{\tau_i \le t\right\} \\
&= \left(\int_0^t P(\tau_i) d\tau_i\right)^N \\
&= (1 - e^{-k_1 t})^N.
\end{aligned}
\tag{15}
$$

***Equation 15*** is the cumulative distribution for the fusion times $T_1$ of vesicles in the fast pool. The corresponding probability density function is

$$p_1(t) \equiv \frac{dF_1(t)}{dt} = Nk_1(1 - e^{-k_1 t})^{N-1} e^{-k_1 t}. \tag{16}$$

The fusion process for vesicles in the slow pool consists of two sequential steps: conformation changes of $N$ SNAREs with rate constant $k_1$ and a slow reaction step with rate constant $k_2$. *In vivo*, the slow step corresponds to the replenishment of the readily releasable pool (RRP) with vesicles from the reserve pool through their docking and priming. *In vitro*, the slow step corresponds to the escape from the metastable "hemifusion diaphragm" state of the vesicle. Thus, the fusion time for vesicles in the slow pool is $T_2 = T_1' + t_2$, where $T_1'$ has the same probability distribution as $T_1$ (***Equations 15; 16***) and $t_2$ satisfies the exponential distribution with parameter $k_2$. Since $T_1'$ and $t_2$ are mutually independent, the probability density distribution for $T_2$ is the following convolution:

$$
\begin{aligned}
p_2(t) \quad &= \int_0^t k_2 e^{-k_2(t-\tau)} p_1(\tau) d\tau \\
&= \int_0^t Nk_1 k_2 e^{-k_2(t-\tau)} (1 - e^{-k_1 \tau})^{N-1} e^{-k_1 \tau} d\tau \\
&= Nk_1 k_2 e^{-k_2 t} \sum_{i=0}^{N-1} \binom{N-1}{i} (-1)^i \int_0^t e^{-((i+1)k_1 - k_2)\tau} d\tau \\
&= Nk_1 k_2 \sum_{i=0}^{N-1} \binom{N-1}{i} (-1)^i \frac{e^{-k_2 t} - e^{-(i+1)k_1 t}}{(i+1)k_1 - k_2}.
\end{aligned}
\tag{17}
$$

We use $(1 + x)^N = \sum_{i=0}^{N} \binom{N}{i} x^i$ in the third line. The cumulative distribution for $T_2$ is obtained by integration of ***Equation 17***:

$$F_2(t) \equiv \int_0^t p_2(t) dt = \sum_{i=0}^{N-1} \binom{N}{i+1} (-1)^i \left(1 - \frac{(i+1)k_1 e^{-k_2 t} - k_2 e^{-(i+1)k_1 t}}{(i+1)k_1 - k_2}\right) \tag{18}$$

## Synaptic release statistics and the average release rate

The cumulative distributions for fusion time, *Equations 15; 18*, allow us to calculate the probability distribution for the number of vesicles from the fast and slow pathways, $n_1(t)$ and $n_2(t)$, that have fused by time $t$. Assuming that fusion of vesicles within each pool is independent and random, we have

$$n_1(t) \sim B(n_{tot1}, F_1(t)), \qquad n_2(t) \sim B(n_{tot2}, F_2(t)), \tag{19}$$

where $B(n, p)$ is binomial distribution with parameters $n$ and $p$, and $n_{tot1}$ and $n_{tot2}$ are the sizes of the fast and slow pools. Furthermore, assuming that vesicles in the fast and slow pools are released independently, the distribution of the total number of released vesicles, $n(t) = n_1(t) + n_2(t)$, is found as the convolution of two binomial distributions:

$$
\begin{aligned}
P\{n(t) = m\} &= \sum_{i=0}^{m} P\{n_1(t) = i\} P\{n_2(t) = m - i\} \\
&= \sum_{i=0}^{m} \binom{n_{tot1}}{i} \binom{n_{tot2}}{m-i} F_1(t)^i (1 - F_1(t))^{n_{tot1}-i} F_2(t)^{m-i} (1 - F_2(t))^{n_{tot2}-m+i}.
\end{aligned}
\tag{20}
$$

*Equation 20* gives the probability that $m$ vesicles fuse by time $t$. When the slow step $k_2$ is neglected, $F_2(t) \sim 0$, *Equation 20* reduces to the binomial distribution (*Malagon et al., 2016*): $P\{n(t) = m\} = \binom{n_{tot1}}{m} F_1(t)^m (1 - F_1(t))^{n_{tot1}-m}$. Using *Equation 20*, the critical number of SNAREs, $N$, can be accurately determined by Bayesian model comparison from the release statistics.

The average cumulative release is then

$$\langle n(t) \rangle = \langle n_1(t) \rangle + {}_2(t) \rangle = n_{tot1} F_1(t) + n_{tot2} F_2(t), \tag{21}$$

where $F_1(t)$ and $F_2(t)$ are given by *Equations 15; 18*. This is *Equation 2* in the main text.

The average release rate is obtained by differentiating *Equation 21*:

$$\frac{d\langle n(t) \rangle}{dt} = n_{tot1} p_1(t) + n_{tot2} p_2(t), \tag{22}$$

where $p_1(t)$ and $p_2(t)$ are given by *Equations 16; 17*. This is *Equation 1* in the main text.

## Average release in various asymptotic regimes

Examining the asymptotic behavior of the exact solutions derived above yields approximate yet simple and accurate expressions for experimentally measurable quantities.

On the short timescale, $t \ll 1/k_1 \ll 1/k_2$, keeping the leading-order term $k_1 t$ and neglecting all the terms related to $k_2 t \ll k_1 t$ in *Equation 22*, we have

$$
\begin{aligned}
\frac{d\langle n(t) \rangle}{dt} &= Nk_1 \left[ n_{tot1}(1 - e^{-k_1 t})^{N-1} e^{-k_1 t} + n_{tot2} k_2 \sum_{i=0}^{N-1} \binom{N-1}{i} (-1)^i \frac{e^{-k_2 t} - e^{-(i+1)k_1 t}}{(i+1)k_1 - k_2} \right] \\
&\approx N n_{tot1} k_1^N t^{N-1}.
\end{aligned}
\tag{23}
$$

*Equation 23* provides a means to test the kinetic scheme in *Scheme 1* and distinguish it from the previously proposed scheme (*Bollmann et al., 2000*; *Miki et al., 2018*; *Schneggenburger and Neher, 2000*) that assumed that the binding of each calcium ion to a SNARE complex constitutes a separate rate-limiting step along the fusion reaction. Specifically, the schemes may be distinguished through the scaling behavior of $\frac{d\langle n(t) \rangle}{dt}$ at short-times, $t \ll 1/k_1$. It can be shown that, for a general stochastic trajectory with sequential transitions that are characterized by rate constants $\{k_i\}_{i=1}^{M}$, $\frac{d\langle n(t) \rangle}{dt} \sim t^{M-1}$ at times $t \ll 1/\max(k_i)$. *Scheme 1* corresponds to $M = N \sim 2$. In contrast, the scheme based on individual calcium ion binding (*Bollmann et al., 2000*; *Miki et al., 2018*; *Schneggenburger and Neher, 2000*) corresponds to $M \geq 4$. The time resolution of the current experiments $t_c \sim 1/k_1 \sim 1ms$ may not be sufficient to resolve these different behaviors, but future experiments may make this test possible.

On the intermediate timescale $1/k_1 \ll t \ll 1/k_2$, we have $F_1(t) \approx 1$ and thus

$$
\begin{aligned}
F_2(t) \ &\approx \sum_{i=0}^{N-1} \binom{N}{i+1}(-1)^i \left(1 - \frac{(i+1)k_1 e^{-k_2 t}}{(i+1)k_1 - k_2}\right) \\
&\approx \sum_{i=0}^{N-1} \binom{N}{i+1}(-1)^i \left(1 - \frac{(i+1)k_1 e^{-k_2 t}}{(i+1)k_1}\right) \\
&= (1 - e^{-k_2 t}) \approx k_2 t.
\end{aligned}
\tag{24}
$$

Therefore, *Equation 21* becomes

$$
\langle n(t) \rangle = n_{tot1} F_1(t) + n_{tot2} F_2(t) \approx n_{tot1} + n_{tot2} k_2 t.
\tag{25}
$$

On the long timescale $t \gg 1/k_2 \gg 1/k_1$, we have $F_1(t) \approx 1$ and $F_2(t) \approx 1 - e^{-k_2 t}$ (see *Equation 24*), and thus

$$
\langle n(t) \rangle = n_{tot1} F_1(t) + n_{tot2} F_2(t) \approx n_{tot1} + n_{tot2}(1 - e^{-k_2 t}).
\tag{26}
$$

## Peak release rate

The time $t_{max}$ at which the release rate is maximal can be obtained by setting the derivative of the average release rate in *Equation 22* to be zero:

$$
\begin{aligned}
0 \ &= n_{tot1} \frac{dp_1(t)}{dt} + n_{tot2} \frac{dp_2(t)}{dt} \\
&= N k_1 \sum_{i=0}^{N-1} \binom{N-1}{i}(-1)^{i-1} \left[ n_{tot1}(i+1)k_1 e^{-(i+1)k_1 t} + n_{tot2}k_2 \frac{k_2 e^{-k_2 t} - (i+1)k_1 e^{-(i+1)k_1 t}}{(i+1)k_1 - k_2} \right] \\
&= N n_{tot1} k_1^2 \sum_{i=0}^{N-1} \binom{N-1}{i}(-1)^{i+1}(i+1) e^{-(i+1)k_1 t} \left[ 1 + \frac{n_{tot2}k_2}{(i+1)n_{tot1}k_1} \frac{\frac{k_2}{(i+1)k_1} e^{(i+1)(1 - \frac{k_2}{(i+1)k_1})k_1 t} - 1}{1 - \frac{k_2}{(i+1)k_1}} \right] \\
&\equiv N n_{tot1} k_1^2 \sum_{i=0}^{N-1} \binom{N-1}{i}(-1)^{i+1}(i+1) e^{-(i+1)k_1 t} A_i,
\end{aligned}
\tag{27}
$$

where we used $p_1(t) = N k_1 (1 - e^{-k_1 t})^{N-1} e^{-k_1 t} = N k_1 \sum_{i=0}^{N-1} \binom{N-1}{i}(-1)^i e^{-(i+1)k_1 t}$ in the second line and introduced $A_i$ in the last line for convenience. The time $t_{max}$ is the solution of *Equation 27*, which is a transcendental equation that is challenging to solve for $N \geq 2$. In practice, $k_2/k_1 \ll 1$, which allows us to expand $A_i$ up to the first order of $k_2/k_1$:

$$
\begin{aligned}
A_i \ &\approx 1 + \frac{n_{tot2}k_2}{(i+1)n_{tot1}k_1}\left(1 + \frac{k_2}{(i+1)k_1}\right)\left(\frac{k_2}{(i+1)k_1} e^{(i+1)(1 - \frac{k_2}{(i+1)k_1})k_1 t} - 1\right) \\
&\approx 1 + \frac{n_{tot2}k_2}{(i+1)n_{tot1}k_1}\left(\frac{k_2}{(i+1)k_1} e^{(i+1)(1 - \frac{k_2}{(i+1)k_1})k_1 t} - 1\right),
\end{aligned}
\tag{28}
$$

where high order terms are dropped in the second line. After dropping the constant factor $N n_{tot1} k_1^2$, *Equation 27* becomes

$$
\begin{aligned}
0 \ &= \sum_{i=0}^{N-1} \binom{N-1}{i}(-1)^{i+1}(i+1)\left[ e^{-(i+1)t} - \frac{n_{tot2}k_2}{(i+1)n_{tot1}k_1} e^{-(i+1)t} + \frac{n_{tot2}}{n_{tot1}}\left(\frac{k_2}{(i+1)k_1}\right)^2 e^{-k_2 t} \right] \\
&\approx \sum_{i=0}^{N-1} \binom{N-1}{i}(-1)^{i+1}(i+1) e^{-(i+1)t} - \frac{n_{tot2}k_2}{n_{tot1}k_1} \sum_{i=0}^{N-1} \binom{N-1}{i}(-1)^{i+1} e^{-(i+1)k_1 t} \\
&= (1 - e^{-k_1 t})^{N-2}(N - e^{+k_1 t}) + \frac{n_{tot2}k_2}{n_{tot1}k_1} e^{-k_1 t}(1 - e^{-k_1 t})^{N-1} \\
&= (1 - e^{-k_1 t})^{N-2}\left(N - e^{+k_1 t} + \frac{n_{tot2}k_2}{n_{tot1}k_1} e^{-k_1 t} - \frac{n_{tot2}k_2}{n_{tot1}k_1} e^{-2k_1 t}\right).
\end{aligned}
\tag{29}
$$

In the second line, the last term is dropped due to $\left(\frac{k_2}{(i+1)k_1}\right)^2 e^{-k_2 t} = O\left(\left(\frac{k_2}{k_1}\right)^2\right)$. In the third line, we used $\sum_{i=0}^{N-1} \binom{N-1}{i}(i+1)x^{i+1} = (1+x)^{N-2}(N - \frac{1}{x})$ for $N \geq 2$ and $\sum_{i=0}^{N-1} \binom{N-1}{i}x^{i+1} = (1+x)^{N-1}$.

Now let $x = e^{-k_1 t}$ and $\epsilon = \frac{n_{tot2}k_2}{n_{tot1}k_1} \ll 1$. The equation for $t_{max}$ up to order $O(\epsilon)$ becomes

$$
\epsilon x^3 - \epsilon x^2 - N x + 1 = 0.
\tag{30}
$$

Assuming $x = x_0(1 + \epsilon x_1 + O(\epsilon^2))$ and comparing the coefficients of $\epsilon^0$ and $\epsilon^1$ in **Equation 30**, we find: $x_0 = \frac{1}{N}$ and $x_1 = \frac{1-N}{N^3}$. Therefore,

$$t_{max} \approx -\frac{\ln x_0 + \ln(1 + \epsilon x_1)}{k_1} + O(\epsilon^2) \approx \frac{\ln N}{k_1} + \frac{(N-1)n_{tot2}k_2}{N^3 n_{tot1} k_1^2} + O(\epsilon^2). \tag{31}$$

With $e^{-k_1 t_{max}} = \frac{1}{N} + \frac{n_{tot2}k_2}{n_{tot1}k_1}\frac{1-N}{N^4} + O(\epsilon^2)$, the peak release rate, up to the first order of $\epsilon = \frac{n_{tot2}k_2}{n_{tot1}k_1}$, is

$$
\begin{aligned}
\left.\frac{d\langle n(t)\rangle}{dt}\right|_{t=t_{max}} &= n_{tot1}p_1(t_{max}) + n_{tot2}p_2(t_{max})\\
&= Nn_{tot1}k_1\left[(1 - e^{-k_1 t_{max}})^{N-1}e^{-k_1 t_{max}} + \frac{n_{tot2}k_2}{n_{tot1}k_1}\sum_{i=0}^{N-1}\binom{N-1}{i}(-1)^i\frac{e^{-k_2 t_{max}} - e^{-(i+1)k_1 t_{max}}}{(i+1)k_1 - k_2}\right]\\
&\approx Nn_{tot1}k_1(1 - \frac{1}{N} - \frac{n_{tot2}k_2}{n_{tot1}k_1}\frac{1-N}{N^4})^{N-1}(\frac{1}{N} - \frac{n_{tot2}k_2}{n_{tot1}k_1}\frac{N-1}{N^4})\\
&\quad + Nn_{tot2}\sum_{i=0}^{N-1}\binom{N-1}{i}(-1)^i\frac{\frac{k_2}{(i+1)k_1}}{1 - \frac{k_2}{(i+1)k_1}}e^{-(i+1)k_1 t_{max}}(e^{(i+1)k_1 t_{max}(1 - \frac{k_2}{(i+1)k_1})} - 1)\\
&\approx n_{tot1}k_1(1 - \frac{1}{N})^{N-1}(1 + \frac{N-1}{N^3}\frac{n_{tot2}k_2}{n_{tot1}k_1})(1 - \frac{n_{tot2}k_2}{n_{tot1}k_1}\frac{N-1}{N^3})\\
&\quad + Nn_{tot2}k_1\sum_{i=0}^{N-1}\binom{N-1}{i}(-1)^i\frac{k_2}{(i+1)k_1}e^{-(i+1)k_1 t_{max}}(e^{(i+1)k_1 t_{max}} - 1)\\
&\approx Nn_{tot1}k_1(1 - \frac{1}{N})^{N-1} + Nn_{tot2}k_2(\frac{1}{N} + \frac{(1 - e^{-k_1 t_{max}})^N - 1}{N})\\
&\approx n_{tot1}k_1(1 - \frac{1}{N})^{N-1} + n_{tot2}k_2(1 - \frac{1}{N})^N = n_{tot1}k_1(1 - \frac{1}{N})^{N-1}(1 + \frac{n_{tot2}k_2}{n_{tot1}k_1}\frac{N-1}{N}),
\end{aligned}
\tag{32}
$$

which is **Equation 3** in the main text.

## Calcium-dependent rate constant $k_1([Ca^{2+}])$

Analogous to the expression for the $pH$-dependent Gibbs free energy of a protein (**Schaefer et al., 1997**), the calcium-dependent free energy can be written as

$$G(n_{Ca}, [Ca^{2+}]) = G(n_{Ca}, [Ca^{2+}]_0) - k_B T n_{Ca}\ln(\frac{[Ca^{2+}]}{[Ca^{2+}]_0}), \tag{33}$$

where $n_{Ca}$ is the average occupancy (relative to the reference value) of calcium ions on a SNARE complex. The generic free energy profile $G(n_{Ca}, [Ca^{2+}]_0)$ at reference calcium concentration $[Ca^{2+}]_0$ with a well and a barrier can be captured by a cubic polynomial (**Dudko et al., 2006**):

$$G(n_{Ca}, [Ca^{2+}]_0) = \frac{3}{2}\Delta G^{\ddagger}\frac{n_{Ca}}{n_{Ca}^{\ddagger}} - 2\Delta G^{\ddagger}\left(\frac{n_{Ca}}{n_{Ca}^{\ddagger}}\right)^3, \tag{34}$$

where $\Delta G^{\ddagger}$ is the barrier height and $n_{Ca}^{\ddagger}$ is the average occupancy number of calcium ions at the top of the barrier (**Appendix 1—figure 1**).

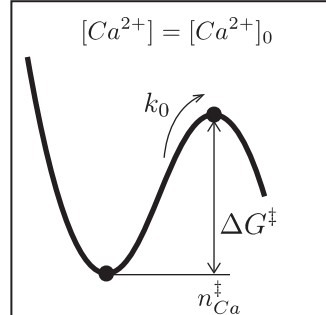 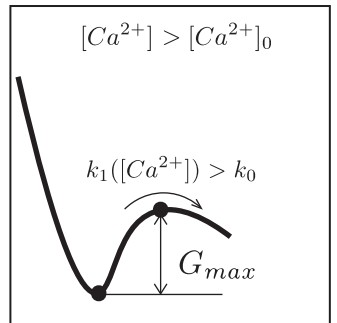

**Appendix 1—figure 1.** Schematic representation of the free energy profile of a SNARE assembly. (Left) The free energy profile at a reference calcium level $[Ca^{2+}] = [Ca^{2+}]_0$. The activation barrier
*Appendix 1—figure 1 continued on next page*

*Appendix 1—figure 1 continued*

$\Delta G^{\ddagger}$ and the average number $n_{Ca}^{\ddagger}$ of calcium ions bound to the SNARE assembly at transition state are indicated. (Right) An elevation of the calcium level lowers the free energy barrier and thereby increases the rate $k_1$ of the SNARE conformation transition.

The reaction rate $k_1([Ca^{2+}])$ can then be derived from the *Kramers, 1940* formalism generalized to the presence of a bias field (*Dudko et al., 2006*). For given calcium concentration $[Ca^{2+}]$, the maximum of the free energy can be found from *Equations 33 and 34* by solving $dG(n_{Ca}, [Ca^{2+}])/dn_{Ca} = 0$:

$$G_{max} = \frac{\Delta G^{\ddagger}}{2}\left(1 - \frac{2k_B T n_{Ca}^{\ddagger}}{3\Delta G^{\ddagger}}\ln(\frac{[Ca^{2+}]}{[Ca^{2+}]_0})\right)^{\frac{3}{2}} \tag{35}$$

at

$$\frac{n_{Ca}^{max}}{n_{Ca}^{\ddagger}} = \frac{1}{2}\sqrt{1 - \frac{2k_B T n_{Ca}^{\ddagger}}{3\Delta G^{\ddagger}}\ln(\frac{[Ca^{2+}]}{[Ca^{2+}]_0})}. \tag{36}$$

Due to symmetry in *Equation 34*, the Kramers rate can be written as (*Kramers, 1940*)

$$k([Ca^{2+}]) = A\left(\left|\frac{d^2G}{dn_{Ca}^2}\right|\right)_{n_{Ca}^{max}} e^{-2G_{max}}, \tag{37}$$

where $A = k_0 e^{\frac{\Delta G^{\ddagger}}{k_B T}}$ is a constant, independent of $[Ca^{2+}]$ and $\left(\left|\frac{d^2G}{dn_{Ca}^2}\right|\right)_{n_{Ca}^{max}}$ is the calcium-dependent curvature of $G(n_{Ca}, [Ca^{2+}])$ at $n_{Ca} = n_{Ca}^{max}$. Substituting *Equations 35 and 33* into *Equation 37* yields *Equation 4* for the calcium-dependent reaction rate.

*Appendix 1—figure 2A* shows the temporal profiles of the average release rate from the theory (*Equations 1 and 4*) for different values of calcium concentration. *Appendix 1—figure 2B* shows the average release rate for different values of the critical number of SNARE assemblies, $N$.

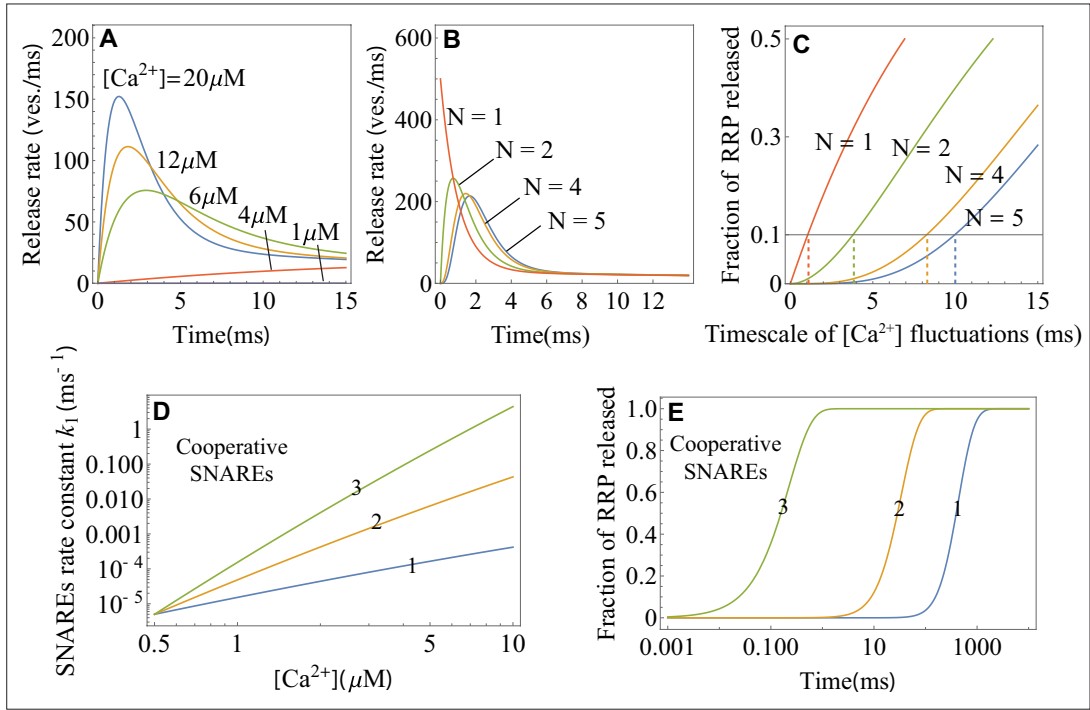

**Appendix 1—figure 2.** The effects of calcium concentration, the critical number of independent SNARE assemblies, and the number of cooperative SNAREs within a super-assembly on the
*Appendix 1—figure 2 continued on next page*

*Appendix 1—figure 2 continued*

neurotransmitter release dynamics, as predicted by the theory. (**A**) Temporal profiles of the average release rate from *Equations 1 and 4* across the range of calcium concentrations (indicated) typical for an action potential. Due to the exponential factor in *Equation 4*, the release turns on rapidly upon calcium influx and terminates rapidly with calcium depletion, resulting in a high temporal precision of synaptic release. (**B**) Temporal profile of the average release rate (*Equation 1*) when $N$ independent SNARE assemblies per vesicle are required for fusion. $N = 2$ provides the optimal balance between stability with respect to fluctuations in calcium concentration (low release rate on the sub-millisecond timescale) and temporal precision (the fastest rise of average release rate). (**C**) Fraction of the RRP vesicles released due to spontaneous calcium fluctuations of varying timescales when $N$ SNARE assemblies per vesicle are required for fusion. Synapses with the larger $N$ exhibit robustness (low release fraction) against slower $\left[Ca^{2+}\right]$ fluctuations. The timescales of $\left[Ca^{2+}\right]$ fluctuations for which the synapse with a given $N$ is robust (fraction of released RRP = 0.1 , horizontal line) are indicated. (**D**) Transition rate for a single-SNARE assembly and for a super-assembly of 2 or 3 SNAREs as a function of calcium concentration, from *Equation 4*. Cooperativity between SNAREs within a super-assembly results in a steeper increase of the rate of conformational change with $\left[Ca^{2+}\right]$ and hence in a faster vesicle release. Every additional SNARE within the super-assembly increases the release rate by a factor of $\sim 100$ at $\left[Ca^{2+}\right] \sim 10\mu M$. (**E**) Temporal profiles of the fraction of RRP vesicles released for a (super-)assembly of $1, 2$ or $3$ SNAREs, from *Equation 2*, at $\left[Ca^{2+}\right] = 10\mu M$. The effect of cooperativity between two or three SNAREs is incorporated through the parameter values for the transition barrier, $n_{Ca}^{\ddagger}$ and $\Delta G^{\ddagger}$, which are, respectively, 2 and 3 times the values for a single SNARE. The parameter values for the 1 SNARE curve are matched to the *in vitro* experiment on syt1 (*Hui et al., 2005*). Parameter values are given in Appendix 3.

## Universal scaling form for the peak release rate

Combining *Equation 3* and *Equation 4*, we have

$$\left.\frac{d\langle n(t)\rangle}{dt}\right|_{t=t_{max}} \approx n_{tot1}\left(1-\frac{1}{N}\right)^{N-1} k_0 \left(1-\frac{2}{3}\ln\left(\frac{[Ca^{2+}]}{[Ca^{2+}]_0}\right)\frac{k_B T n_{Ca}^{\ddagger}}{\Delta G^{\ddagger}}\right)^{\frac{1}{2}}$$
$$\times \exp\left(\frac{\Delta G^{\ddagger}}{k_B T}\left[1-\left(1-\frac{2}{3}\ln(\frac{[Ca^{2+}]}{[Ca^{2+}]_0})\frac{k_B T n_{Ca}^{\ddagger}}{\Delta G^{\ddagger}}\right)^{\frac{3}{2}}\right]\right),$$

(38)

where $\frac{n_{tot2}k_2}{n_{tot1}k_1} \ll 1$ allowed us to ignore the second term in *Equation 3*.

Defining $c = \frac{2k_B T n_{Ca}^{\ddagger}}{3\Delta G^{\ddagger}}\ln\frac{[Ca^{2+}]}{[Ca^{2+}]_0}$ and $a = \frac{(1+\frac{1}{N-1})^{N-1}}{n_{tot1}k_0}$, the above equation becomes

$$\left.\frac{d\langle n(t)\rangle}{dt}\right|_{t=t_{max}} = \frac{(1-c)^{\frac{1}{2}}}{a}\exp\left[\frac{\Delta G^{\ddagger}}{k_B T}\left(1-(1-c)^{\frac{3}{2}}\right)\right].$$

(39)

Let $\mathfrak{r}=\left(\frac{a}{(1-c)^{1/2}}\left.\frac{d\langle n(t)\rangle}{dt}\right|_{t=t_{max}}\right)^{k_B T/\Delta G^{\ddagger}}$. *Equation 39* then gives *Equation 5*.

## Peak postsynaptic current and cumulative release

Following the conductance-based model for postsynaptic response (*Destexhe et al., 1994*), the postsynaptic current caused by an ion channel of a given type can be written as

$$I(t) = g(t)\left(V(t) - E_{rev}\right),$$

(40)

where $g(t)$ is the conductance of the channel, $V(t)$ is the postsynaptic membrane potential and $E_{rev}$ is the reversal potential of the ion corresponding to the ion channel. Different types of channels may have different $g(t)$ and $E_{rev}$. The peak value of postsynaptic current is usually dominated by a single type of channel, e.g. AMPA receptor for excitatory synapses or GABA receptor for inhibitory synapses. Thus, in the following, the postsynaptic current will be assumed to be caused by the dominate channel type.

For an action potential triggered at $t = t_s$, the conductance has a pulse at $t_s$ of amplitude proportional to the number $n(T)$ of neurotransmitters released during the action potential:

$$g(t) = g_0 n(T) e^{-\frac{t-t_s}{\tau}}. \tag{41}$$

Here, $g_0$ depends on the intrinsic properties of the ion channel and the channel density, and $\tau$ is the relaxation timescale of the channel. Since we are concerned with the response to a few (probably one or two) action potentials, *Equation 41* assumes that the postsynaptic receptors are not saturated. Since the time scale for an action potential $T \sim 1ms$ is much shorter than the relaxation time scale of the ion channel ($\tau_{AMPA}, \tau_{GABA} \sim 20ms - 30ms$ [*Destexhe et al., 1994*]), the action potential can be regarded as a delta-function pulse.

The membrane voltage is usually far from the reversal potential when responding to a few action potentials. When the reversal potential is close to the membranes voltage, *Equation 6* is still true but it takes more work to calculate the current kinetics. According to *Equation 40*, the current $I(t)$ is therefore proportional to the conductance $g(t)$, and the peak of $g(t)$ is at $t = t_s$. The peak value of postsynaptic current is

$$I_{PSC} \simeq g_0(V_0 - E_{rev})n(T), \tag{42}$$

where $V_0$ is the resting membrane potential. By letting $\gamma = g_0(V_0 - E_{rev})$ and taking the average of both sides of the above equation, we obtain *Equation 6* in the main text.

## Paired-pulse ratio in short-term plasticity

The number of RRP vesicles $N_1(t)$ is assumed to follow the first-order recovery kinetics:

$$\frac{dN_1(t)}{dt} = \frac{n_{tot1} - N_1(t)}{\tau_{RRP}}, \tag{43}$$

where the RRP pool of total capacity $n_{tot1}$ is assumed to be full initially: $N_1(0) = n_{tot1}$. For a pair of spikes with interpulse interval $\tau_{int}$, this equation yields a solution for the number of vesicles in RRP by the time of arrival of the second spike:

$$n_{1f} \equiv N_1(\tau_{int}) = n_{tot1} - \langle n_1(T) \rangle e^{-\frac{\tau_{int}}{\tau_{RRP}}}, \tag{44}$$

which contains the vesicles left after the first AP-triggered release and the vesicles replenished from the reserve pool during the interspike interval.

To capture the syt7-mediated facilitation scenario, we assume that the rate constant $k_1([Ca^{2+}])$ during the first spike increases by a factor of $\sigma$ following the first spike (the timescale on which $Ca^{2+}$ ions bind to syt7 is negligible compared to the typical values of $\tau_{int}$) and that the new rate constant $\sigma k_1([Ca^{2+}])$ decays with timescale $\tau_{res}$. Therefore, the rate constant during the second spike is

$$k_{1f} = \left(1 + (\sigma - 1)e^{-\frac{\tau_{int}}{\tau_{res}}}\right) k_1([Ca^{2+}]). \tag{45}$$

Using *Equation 2*, the pair-pulse ratio is thus

$$\frac{\langle n_f(T) \rangle}{\langle n_i(T) \rangle} \simeq \frac{n_{1f}}{n_{tot1}} \left(\frac{1 - e^{-k_{1f}T}}{1 - e^{-k_1([Ca^{2+}])T}}\right)^N, \tag{46}$$

where the release from the reserve pool (second term in *Equation 2*) has been ignored. Using *Equation 44* and *Equation 45* for $n_{1f}$ and $k_{1f}$, we arrive at *Equation 7*.

To describe the buffer saturation scenario, let $[Ca^{2+}]_i$ denote the calcium concentration sensed by the calcium sensor during the first spike. Let us assume that the amplitude of the increment in the calcium concentration due to the partial buffer saturation is $I_{Ca}$. Let us further assume that the dissociation of calcium from the buffer follows the first-order kinetics with dissociation time constant $\tau_{Ca}$. Therefore, the calcium concentration sensed by the calcium sensor after time $\tau_{int}$ is

$$\left[Ca^{2+}\right]_f = \left[Ca^{2+}\right]_i + I_{Ca} e^{-\frac{\tau_{int}}{\tau_{Ca}}}. \tag{47}$$

Again using *Equation 2* and ignoring the release from the reserve pool (second term in *Equation 2*), the paired-pulse ratio is

$$\frac{\langle n_f(T)\rangle}{\langle n_i(T)\rangle} \simeq \frac{n_{1,f}}{n_{tot1}}\left(\frac{1-e^{-k_1([Ca^{2+}]_f)T}}{1-e^{-k_1([Ca^{2+}]_i)T}}\right)^N. \tag{48}$$

Using *Equation 44* for $n_{1,f}$, we arrive at *Equation 8*.

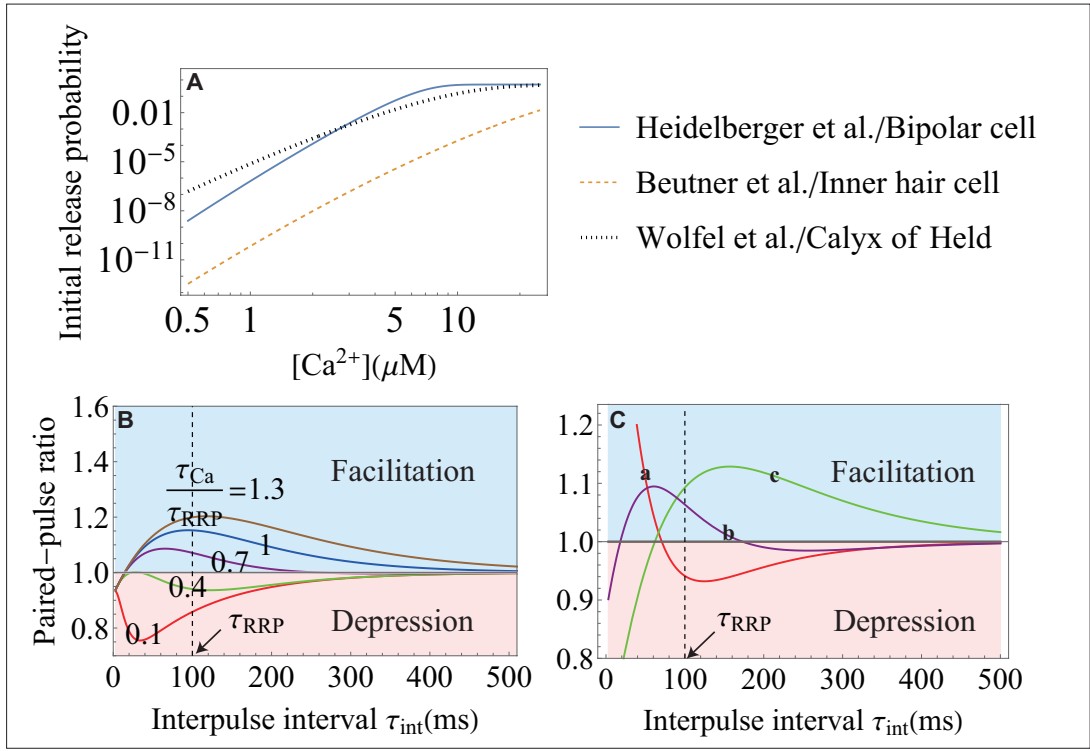

**Appendix 1—figure 3.** The effect of the molecular-level properties of synapses on release probability and short-term plasticity. (**A**) The initial vesicle release probability, $\langle n(T)\rangle / (n_{tot1} + n_{tot2})$, over a range of $\left[Ca^{2+}\right]$ typical in experiments, for different synapses. The release probability can vary significantly in different synapses because of different types of $Ca^{2+}$ - sensors, different coupling between regulatory proteins and SNAREs, and different amount of the initial $Ca^{2+}$ entry. The parameters for each curve correspond to the data from the studies that are indicated on the right. (**B**) Weaker $Ca^{2+}$-sensitivity (here, it is $\sim 1/3$ of that in *Figure 4G*) of a SNARE assembly results in a smaller dynamic range of short-term facilitation. The $Ca^{2+}$-sensitivity of a SNARE is defined as the ratio of the conformational rate constants (*Equation 4*) during the action potential, $k_1\left([Ca^{2+}]_0 + \Delta[Ca^{2+}]\right)$, and at rest, $k_1\left([Ca^{2+}]_0\right)$. The value of $\Delta[Ca^{2+}]$ is set at $10\mu M$. (**C**) Distinct short-term facilitation/depression modes in synapses that differ on the molecular level, from theory (*Equation 8*). Three different sets of parameters $\{\Delta G^{\ddagger}, n_{Ca}^{\ddagger}, k_0\}$ and $\tau_{Ca}$ are used for curves a, b and c, representing different properties of the molecular constituents for the three synapses. In curve a, the high frequency transient input (with small $\tau_{int}$) is facilitated and the low frequency input (with large $\tau_{int}$) is depressed. The effects are reversed in curve c with depression at high frequency and facilitation at low frequency. In curve b, inputs with intermediate frequencies are facilitated and inputs with high and low frequencies are depressed. The dynamic range for each curve can be amplified by changing the timescale for RRP replenishment, $\tau_{RRP}$. Parameter values are given in Appendix 3.

## Optimal Synaptic Strength

We derive the condition for optimal RRP size as follows. Typically $k_2 T \ll 1$, hence we can ignore the contribution of the reserve pool. The probability that there is no response in the postsynaptic neuron for an action potential of duration $T$ can be written as

$$p_{fail}(T, k_1([Ca^{2+}]), n_{tot1}) = \sum_{m=0}^{M} \binom{n_{tot1}}{m} F_1(T)^k (1 - F_1(T))^{n_{tot1} - m}, \tag{49}$$

where $[Ca^{2+}]$ is calcium concentration in pre-synaptic neuron and $F_1(T)$ is defined in **Equation 15**.

The error probability can be written as

$$P(error) = q p_{fail}(T, k_1([Ca^{2+}]_{AP}), n_{tot1})) + (1 - q)(1 - p_{fail}(T, k_1([Ca^{2+}]_{rest}), n_{tot1}) \equiv P(n_{tot1}), \tag{50}$$

as shown in **Equation 11** in the main text. Here we focus on the dependence of $P(error)$ on $n_{tot1}$.

To see whether $P(n_{tot1})$ has a minimum, we solve the following inequality:

$$
\begin{aligned}
&P(n_{tot1} + 1) \geq P(n_{tot1}) \\
&\Leftrightarrow q \left[ p_{fail}(T, k_1([Ca^{2+}]_{AP}), n_{tot1} + 1) - p_{fail}(T, k_1([Ca^{2+}]_{AP}), n_{tot1}) \right] \\
&\geq (1 - q) \left[ p_{fail}(T, k_1([Ca^{2+}]_{rest}), n_{tot1} + 1) - p_{fail}(T, k_1([Ca^{2+}]_{rest}), n_{tot1}) \right].
\end{aligned}
\tag{51}
$$

But

$$
\begin{aligned}
&p_{fail}(T, k_1([Ca^{2+}]), n_{tot1} + 1) - p_{fail}(T, k_1([Ca^{2+}]), n_{tot1}) \\
&= \sum_{k=0}^{M} \binom{n_{tot1} + 1}{k} F_1(T)^k (1 - F_1(T))^{n_{tot1} + 1 - k} - \sum_{k=0}^{M} \binom{n_{tot1}}{k} F_1(T)^k (1 - F_1(T))^{n_{tot1} - k} \\
&= \sum_{k=0}^{M} \left( \binom{n_{tot1}}{k} + \binom{n_{tot1}}{k-1} \right) F_1(T)^k (1 - F_1(T))^{n_{tot1} + 1 - k} - \sum_{k=0}^{M} \binom{n_{tot1}}{k} F_1(T)^k (1 - F_1(T))^{n_{tot1} - k} \\
&= -F_1(T) \sum_{k=0}^{M} \binom{n_{tot1}}{k} F_1(T)^k (1 - F_1(T))^{n_{tot1} - k} + \sum_{k=0}^{M-1} \binom{n_{tot1}}{k} F_1(T)^{k+1} (1 - F_1(T))^{n_{tot1} - k} \\
&= -\binom{n_{tot1}}{M} F_1(T)^{M+1} (1 - F_1(T))^{n_{tot1} - M}.
\end{aligned}
\tag{52}
$$

Therefore, **Equation 51** becomes $q \binom{n_{tot1}}{M} F_a^M (1 - F_a)^{n_{tot1} - M} \leq (1 - q) \binom{n_{tot1}}{M} F_r^M (1 - F_r)^{n_{tot1} - M}$, where $F_a \equiv (1 - e^{-k_1([Ca^{2+}]_{AP})T})^N$ and $F_r \equiv (1 - e^{-k_1([Ca^{2+}]_{rest})T})^N$ are the probabilities that a vesicle in RRP is fused during the action potential and at rest, respectively.

We can solve for $n_{tot1}$ from the above inequality as follows:

$$n_{tot1} \geq M(1 + \frac{\ln \frac{F_a}{F_r}}{\ln \frac{1 - F_r}{1 - F_a}}) + \frac{\ln \frac{q}{1-q}}{\ln \frac{1 - F_r}{1 - F_a}}. \tag{53}$$

Since $n_{tot1}$ is an integer, the optimal RRP size $n_{tot1}^*$ can be written as **Equation 12**.

## Appendix 2

### Simulations set-up

We use Gillespie algorithm to model the fusion dynamics in the simulations. To validate the analytic expressions derived within the framework of the model in *Scheme 1* and *Figure 1* , we performed simulations of this model and examined whether the analytical expressions can accurately recover the input parameters of the simulations when used as a fitting tool. Next, to test the limitations of the assumptions of our model, we performed a modified set of simulations and compared their results with the original model. Specifically, the simulations were modified to incorporate the effects that are thought to be relevant for synaptic transmission *in vivo*: (i) the finite-capacity effect of the readily-releasable pool and (ii) the spatial coupling between voltage-gated calcium channels and vesicle release sites.

### Testing the analytic theory

Numerical simulations were carried out for the vesicle fusion model in *Scheme 1* and *Figure 1* with the following parameter values: $n_{tot1} = 500$, $n_{tot2} = 1000$, $k_2 = 0.027 ms^{-1}$, $\Delta G^{\ddagger} = 18.4 k_B T$, $n_{Ca}^{\ddagger} = 3.48$, $k_0 = 1.67 \times 10^{-7} ms^{-1}$, and $N = 2$. Calcium concentration was varied from $0.05 \mu M$ to $20 \mu M$, with the reference value set at $[Ca^{2+}]_0 = 50 nM$. The number of vesicles that fuse by time $t$, $n(t)$, was recorded from $t = 0 ms$ to $t = 100 ms$ with time interval $0.4 ms$, and the average over 40 trajectories was calculated to obtain the average cumulative release $\langle n(t) \rangle$. The data generated through simulations were then fitted with the analytic expression for the average cumulative release (*Equation 2* ). The theory was found to accurately reproduce the input parameters used in the simulations (*Appendix 2—figure 1*). For low calcium concentration ($[Ca^{2+}] < 1 \mu M$), the recording time for $\langle n(t) \rangle$ had to be extended to $\sim 1000 ms$ in order to reliably extract the rate constant $k_1$. This is because, at low calcium concentration, $k_1$ and $k_2$ become comparable and the term $k_1 - k_2$ in the denominator in the expression of $\langle n(t) \rangle$ (*Equation 2* ) tends to cause numerical instability. Nevertheless, the fit to $k([Ca^{2+}])$ in *Equation 4* in the main text was found to be always reliable as long as there are enough data points at high calcium concentrations ($[Ca^{2+}] > 1 \mu M$), which is the case for the experimental data in *Figure 2*.

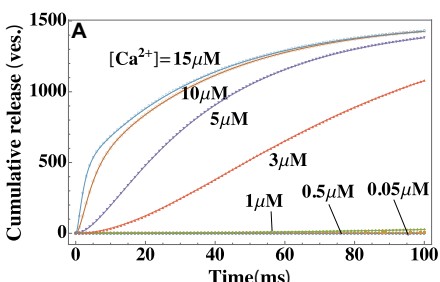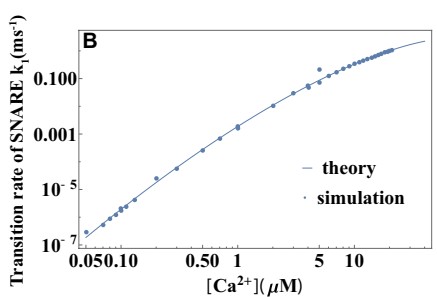

**Appendix 2—figure 1.** Validation of the theory through simulations. (A) Temporal profiles of average cumulative release at different values of calcium concentrations from simulations (symbols) and a fit to the theory in *Equation 2* (lines). (B) Calcium dependence of the rate constant of SNARE conformational change from simulations in (A), and a fit to *Equation 4* . The reference concentration $[Ca^{2+}]_0 = 50 nM$ is set at a typical resting value of a synapse *in vivo* (*Kaeser and Regehr, 2014*) and $N = 2$. The fit yields the height and width of the activation barrier and the rate constant of SNARE conformational change at $[Ca^{2+}]_0$, which accurately recover the input parameters of the simulations. Parameters are listed in Appendix 3.

To test the validity of the analytical expression for peak release rate in *Equation 3*, the kinetic scheme in *Scheme 1* was simulated at the parameter values indicated above and the peak release rate was computed at different values of $[Ca^{2+}]$. *Equation 3* was used to extract the microscopic parameters $\Delta G^{\ddagger}$, $n_{Ca}^{\ddagger}$, $n_{tot1} k_0 \left(1 - \frac{1}{N}\right)^{N-1}$, which were then compared with the input parameters of the simulations. This procedure was repeated at different values of $n_{tot1}/n_{tot2}$ and $k_2$. For the biologically relevant ratio $n_{tot1}/n_{tot2} \sim 1$ and in the range of calcium concentrations $0.1 \mu M \leq [Ca^{2+}] \leq 0.1 mM$, the analytic expression in *Equation 3* was found to be highly accurate:

the parameters returned by the fit were within less than 5% from the exact parameters used in the simulations. *Appendix 2—figure 2* shows that the accuracy of *Equation 3* does not depend significantly on the value of $k_2$. Even for the very low ratio $n_{tot1}/n_{tot2} \sim 0.5$, the parameters extracted from *Equation 3* still have > 90% accuracy.

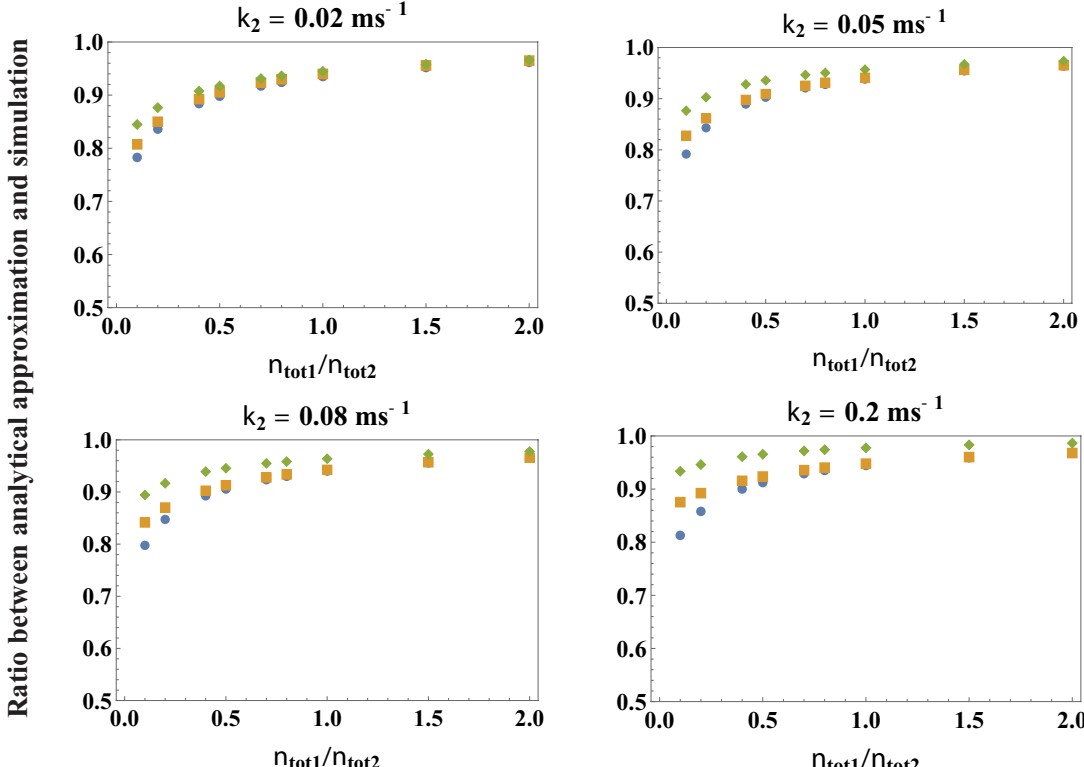

**Appendix 2—figure 2.** Testing the accuracy of the analytical theory for the peak release rate, *Equation 3*. Plotted are the ratios between the parameters extracted from a fit to *Equation 3* with $k_1$ given by *Equation 4* and the exact parameters of the simulations, across a broad range of values of $n_{tot1}/n_{tot2}$ and $k_2$ (indicated in the figure), over the range of calcium concentrations $0.1 \mu M \leq \left[ Ca^{2+} \right] \leq 0.1 mM$. Blue circle is the ratio for $\ln \left[ (1 - \frac{1}{N})^{N-1} n_{tot1} k_0 \right]$, orange square for $\Delta G^{\ddagger}$, green diamond for $n_{Ca}^{\ddagger}$. For the biologically relevant value $n_{tot1}/n_{tot2} \sim 1$, the ratios are very close to 1, indicating that the theory is highly accurate (> 95% accuracy).

## Finite-capacity effect of the readily releasable pool

As found in recent experiments (*Biederer et al., 2017*), synaptic vesicles are released at specialized sites, known as active zones, at the presynaptic terminal. Because there are only a finite number of active zones in each presynaptic button, the maximal number of docked vesicles (state $D$ in *Figure 1D*) is finite. Let $n_{max}$ be the number of release sites on the presynapic membrane and set $n_{tot1} = n_{max}$ in the simulations, which corresponds to the release sites being initially fully occupied by the docked vesicles. To incorporate this finite-capacity effect in the simulations, we now assume that the vesicles in the reserve pool (state $R$ in *Figure 1D*) can be docked to presynaptic membrane only if there is a vacant release site ($n_{max} - n_1(t) > 0$).

We note that the model with the finite size of the readily releasable pool corresponds to the $G/G/N/N/k$ queue model in queueing theory (*Gautam, 2012*). Few results are known for the general $G/G/N/N/k$ queue model, although bounds and approximation methods have been developed for various situations. When the capacity $N \to +\infty$, the $G/G/N/N/k$ queue model converges to our model in Appendix 1 with infinite capacity of the RRP pool, and is exactly solvable. Here, rather than seeking analytic approximations for the finite-capacity effect, we use simulation to explore its properties and the validity of our model in the light of this effect. We define the dimensionless ratio $f \equiv \frac{n_{tot1} k_1}{n_{tot2} k_2}$ and change it from 10 to 0.5. When $f \gg 1$, the depletion rate of the

readily releasable pool is much larger than the replenishing rate, and the readily releasable pool is effectively of an infinite capacity. As expected, the theory (*Equation 2* ) is valid in this regime (*Appendix 2—figure 3A*). When $f$ decreases, deviations from the analytic theory appear, with the cumulative release being slower than that predicted by *Equation 2* (*Appendix 2—figure 3A*). In real neuron, action potential-evoked calcium elevation will lead to $k_1 \gg k_2$ and thus $f \gg 1$, therefore, *Equation 2* is expected to perform well in the biologically relevant range of parameters.

## The Effect of Heterogeneity among Release Sites

It has been pointed out that the docked vesicles in the same synapse may have different release rates due to their different distances to the voltage-gated calcium channels (*Trommershäuser et al., 2003*; *Neher, 2015*). Action potential-evoked calcium influx forms a so-called nanodomain around each channel. Let us assume that diffusion and buffering are the dominant factors that shape the concentration profile of calcium. When the channel is open, the steady state of calcium concentration profile can be described by the following reaction-diffusion equation:

$$D_{[Ca^{2+}]}\nabla^2 c(\mathbf{r}) - \kappa c(\mathbf{r}) = 0, \tag{54}$$

where $D_{[Ca^{2+}]} \approx 200 \mu m^2/s$ is the diffusion coefficient of a calcium ion inside the cell and $\kappa = k_-[B] \approx 0.3 \mu s^{-1}$(*Delvendahl et al., 2015*) is the binding rate that characterizes calcium buffers. *Equation 54* can be solved by assuming spherical symmetry (*Neher, 1998*):

$$c(\mathbf{r}) = \alpha([Ca^{2+}]_{out})\frac{e^{-r/\lambda}}{r}, \tag{55}$$

where $\lambda = \sqrt{D_{[Ca^{2+}]}/\kappa} \approx 30nm$ sets the characteristic length scale of the calcium nanodomain, $\alpha([Ca^{2+}]_{out})$ measures the magnitude of the calcium current through the channel at extracellular calcium concentrations $[Ca^{2+}]_{out}$, and $r$ is the distance from the channel. Due to the decaying concentration profile in *Equation 55*, vesicles that are closer to the channel experience higher calcium concentration and thus have higher release rate. The relative positions of the release site and voltage-gated calcium channels on the presynaptic membrane may therefore have a significant impact on the action-potential-evoked vesicle release dynamics.

Recently, several studies established the nanoscale organization of the molecular apparatus around the vesicle release sites (*Stanley, 2016*; *Gramlich and Klyachko, 2019*; *Biederer et al., 2017*). It has been found that release sites and channels together form clusters within active zones on the presynaptic membrane (*Maschi and Klyachko, 2017*; *Miki et al., 2017*; *Nakamura et al., 2015*). The typical size of an active zone is about 250 nm (*Gramlich and Klyachko, 2019*), with multiple release sites present within a single active zone (*Maschi and Klyachko, 2017*). Multiple channels cluster around a single release site and their distances to the release site are regulated by scaffold proteins (*Böhme et al., 2016*).

Based on these experimental facts, we set up our modified simulations as follows. Each active zone is modeled as a disk of radius $r = 125$ nm, the total number of active zones on the presynaptic synapse is $N_{AZ} = 300$, the number of release sites in an active zone is $N_r = 3$ and the number of channels around each release site is $N_c = 3$. These values are chosen to mimic the organization of release sites in the calyx of Held (*Borst and Soria van Hoeve, 2012*). We further assume that the release sites are uniformly distributed within each active zone (*Maschi and Klyachko, 2017*), and the channels are uniformly distributed around each release site within the range of distances ($\lambda_{min} = 30nm, \lambda_{max} = 40nm$). The calcium current $\alpha([Ca^{2+}]_{out}) = \alpha_{max}\frac{[Ca^{2+}]_{out}}{K_D+[Ca^{2+}]_{out}} \approx \alpha_{max}[Ca^{2+}]_{out} \approx 15\mu m \cdot \mu M \times (\frac{[Ca^{2+}]_{out}}{10\mu M})$(*Schneggenburger et al., 1999*), and the concentration $[Ca^{2+}]_{out}$ is varied from $0.5\mu M$ to $10\mu M$. The release rates for docked vesicles are determined by *Equation 4* in the main text and the parameter values are the same as in the above simulations (See "Testing the Analytic Theory"). Each release site is assumed to have its corresponding reserve pool of size $n_{tot2} = 3$, and the release rate for vesicles in the reserve pool is $k_2 = 0.02ms^{-1}$. Simulation results and the corresponding fits to *Equation 2* in the main text are shown in *Appendix 2—figure 3B*. The simulations show that, as long as $\lambda_{max} - \lambda_{min}$ is not too large, our theory is accurate. Further simulations (not shown) show that, if $\lambda_{max} - \lambda_{min}$ is too large, the cumulative release curve in *Appendix 2—figure 3B* is no longer double-exponential. The fact that cumulative release has in fact been observed to be double-exponential (*Miki et al.,*

*2018*) indicates that the distance between the channel and the release sites is likely to be tightly regulated by the scaffold proteins. We conclude that, in the range of parameters that correspond to real biological systems, the spatial heterogeneity of calcium concentration has no significant effect on the accuracy of the results presented in the main text.

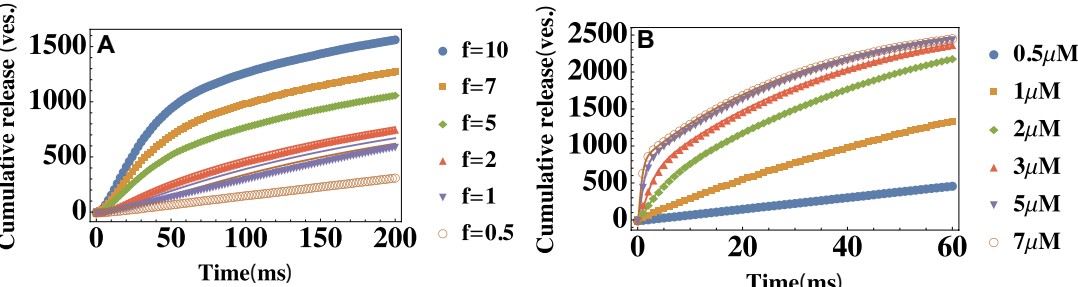

**Appendix 2—figure 3.** Testing the limitations of the theory: Finite capacity of the readily releasable pool and heterogeneity in vesicle pools. (A) Temporal profiles of the average cumulative release $\langle n(t) \rangle$ at different ratios $f = \frac{n_{tot1}k_1}{n_{tot2}k_2}$. Parameter values are $n_{tot2} = 1000$, $k_2 = 0.005 ms^{-1}$ and $k_1 = 0.05 ms^{-1}$, and $n_{tot1}$ is varied to change the ratio $f$ of the depletion and replenishment rates of the readily releasable pool. Symbols: data generated from modified simulations that introduced the finite-capacity effect of the readily releasable pool, lines: *Equation 2* with the same parameter values as those used in the simulations. (B) Temporal profiles of the average cumulative release $\langle n(t) \rangle$ at different values of the extracellular calcium concentration $[Ca^{2+}]_{out}$. The parameters are described in the text. $[Ca^{2+}]_{out}$ is shown in the legend. Data is fitted with *Equation 2*. The cumulative release rate exhibits a double-exponential shape if $\lambda_{max} - \lambda_{min}$ is not too large ($\lesssim 10 nm$).

## Appendix 3

### Critical Number of SNARE assemblies, $N$

To achieve robustness of the fits with limited experimental data available, the fits in the main text were performed at a fixed value of the critical number of SNARE assemblies, $N$, necessary for vesicle fusion. The choice of $N = 2$ was based on the indirect experimental evidence (**Sinha et al., 2011**). Additionally, we performed the least square fits of the *in vivo* data in **Figure 3** with **Equation 2** with the values of $N = 1, 2, 3, 4, 5$. Values $N \geq 6$ would be too large to be biologically realistic (**Brunger et al., 2018**). The fitting errors for different values of $N$ are reported in **Appendix 3—table 1**.

It can be seen that $N = 2$ results in the consistently smallest fitting errors across all values of calcium concentrations used in the experiment, and thus represents the optimal fit. This numerical test provides an additional support for $N = 2$ representing the critical number of SNAREs necessary for fusion.

**Appendix 3—table 1.** Errors from the least square fit for different values of N.
The fits are performed for the experimental data from **Wölfel et al., 2007** to **Equation 2** in the main text. The fitting error is calculated according to $\sum_i |y_i - f(x_i)|^2$. The fit with N = 2 results in consistently smallest fitting errors across all calcium concentrations used in the experiment.

| Least squares error | N = 1 | N = 2 | N = 3 | N = 4 | N = 5 |
|---|---|---|---|---|---|
| $[Ca^{2+}] = 27$ μM | 27.431 | 25.896 | 26.742 | 29.016 | 33.234 |
| $[Ca^{2+}] = 11$ μM | 19.662 | 13.968 | 14.656 | 15.374 | 16.737 |
| $[Ca^{2+}] = 7$ μM | 10.241 | 8.115 | 8.189 | 8.260 | 8.416 |
| $[Ca^{2+}] = 6$ μM | 84.908 | 84.732 | 85.255 | 85.553 | 85.701 |

## Parameter Values Extracted from the Fits or Used for Illustration

**Appendix 3—table 2.** Microscopic parameters of synaptic fusion machinery extracted from the fits in **Figure 2** in the main text.

Rate constant $k_o$ is in ms$^{-1}$, and $v \equiv \left(1 - \frac{1}{N}\right)^{N-1}$.

| $\Delta G^{\ddagger}(k_B T)$ | $n^{\ddagger}_{Ca}$ | $\log_{10}[n_{tot1}k_0 v]$ | Data source |
|---|---|---|---|
| 34.1 ± 3.0 | 7.84 ± 1.92 | −11.5 ± 1.3 | Calyx of Held (**Schneggenburger and Neher, 2000**) |
| 35 ± 19 | 4.48 ± 5.98 | −7.3 ± 0.8 | Calyx of Held (**Lou et al., 2005**) |
| 27.0 ± 1.31 | 6.22 ± 0.50 | −10.7 ± 0.5 | Calyx of Held (**Bollmann et al., 2000**) |
| 22.6 ± 1.7 | 5.58 ± 1.18 | −5.4 ± 0.4 | Calyx of Held (**Sun et al., 2007**) |
| 16.6 | 2.66 | −3.7 | PF-MLI (**Miki et al., 2018**) |
| 7.63 ± 3.34 | 3.34 ± 2.36 | −2.0 ± 0.7 | Photoreceptor (**Duncan et al., 2010**) |
| 19.8 ± 1.5 | 3.54 ± 1.00 | −4.7 ± 1.2 | Calyx of Held (**Wölfel et al., 2007**) |
| 20.3 ± 1.5 | 3.86 ± 0.40 | −5.3 ± 0.6 | Calyx of Held P8-9 (**Kochubey et al., 2009**) |
| 25.2 ± 1.9 | 5.28 ± 0.54 | −7.2 ± 0.7 | Calyx of Held P12 (**Kochubey et al., 2009**) |
| 26.3 ± 2.1 | 5.96 ± 0.56 | −11.1 ± 0.9 | Inner hair cell (**Beutner et al., 2001**) |
| 19.5 ± 5.1 | 5.92 ± 2.26 | −8.7 ± 2.5 | Hippocampal mossy 1ber (**Fukaya et al., 2021**) |
| 20.3 ± 2.0 | 4.40 ± 0.74 | −8.8 ± 0.8 | Cerebellar basket cell (**Sakaba, 2008**) |
| 30.3 ± 3.4 | 4.92 ± 0.56 | −9.6 ± 1.3 | Retina bipolar cell (**Heidelberger et al., 1994**) |
| 20.5 | 1.28 | −5.1 | Chromaffin cell (**Voets, 2000**) |
| 25.7 | 1.02 | −7.6 | Insulin-secreting cell (**Yang and Gillis, 2004**) |
| 20 | 2.82 | −9.3 | in vitro (**Diao et al., 2012**) |

**Appendix 3—table 3.** Parameter values extracted from the fit in *Figure 4A-E* in the main text for syt7-mediated facilitation.

In addition, $\Delta G^{\ddagger} = 20k_B T$ and $n_{Ca}^{\ddagger} = 4$ were fixed at the values typical for the synapse with syt1 as the main $[Ca^{2+}]$-sensor (see *Appendix 3—table 2*, $T = 1ms$ and $[Ca^{2+}] = 10\mu M$. The value of $\tau_{RRP}$ was set at $100ms$ for the facilitation-dominated synapses (*Figure 4A, B, D and E*) and 2000ms for the retinal ribbon synapse (*Figure 4C*).

| $\sigma$ | $\tau_{res}$ (*ms*) | $\log_{10} [k_0(ms^{-1})]$ | Data source |
|---|---|---|---|
| 1.90 | 200 | −8.0 | Corticothalamic *Jackman et al., 2016* |
| 1.66 | 108 | −8.1 | Schaffer collateral *Jackman et al., 2016* |
| 1.47 | 110 | −8.0 | Perforant path *Jackman et al., 2016* |
| 2.05 | 70 | −7.6 | Granule cell *Turecek and Regehr, 2018* |
| 1.80 | 860 | −6.7 | Retinal ribbon *Luo et al., 2015* |

In *Figure 3B*, the fit yields the height and width of the activation barrier and the rate constant of the SNARE conformational change in the resting state ($[Ca^{2+}]_0 = 50nM$): $\Delta G^{\ddagger} \approx (18.7 \pm 1.0)k_BT$, $n_{Ca}^{\ddagger} \approx 3.54 \pm 0.13$ and $\log_{10}\left(\frac{k_0}{ms^{-1}}\right) \approx -6.8 \pm 1.0$. Based on the results in *Appendix 3—table 1*, the fit was performed with $N = 2$.

In *Figure 3C*, the fit yields $k_1 \approx 2.32s^{-1}$ and $k_2 \approx 0.0117s^{-1}$ ($250\mu M$) and $k_1 \approx 2.71s^{-1}$ and $k_2 \approx 0.0198s^{-1}$ ($500\mu M$).

In *Figure 3D*, parameter values used: $\Delta G^{\ddagger} = 35k_BT$, $n_{Ca}^{\ddagger} = 2$, $k_0 = 10^{-5}s^{-1}$, $\frac{n_{tot1}}{n_{tot2}} = \frac{3}{7}$ and $k_2 = 0.0105ms^{-1}$.

In *Figure 3E*, parameter values used: $\Delta G^{\ddagger} = 24k_BT$ and $n_{Ca}^{\ddagger} = 4$. Facilitation ratio is defined as $\left[k_1\left([Ca^{2+}]_i + \Delta[Ca^{2+}]\right)\right]/k_1\left([Ca^{2+}]_i\right)$(*Barrett and Stevens, 1972*). Normalized release is defined as the ratio of residual release, $k_1\left(\Delta[Ca^{2+}]\right)$, and control release, $k_1\left([Ca^{2+}]_i\right)$. The value $[Ca^{2+}]_i$ was set at $10\mu M$ and $\Delta[Ca^{2+}]$ was varied when plotting the curve for the present theory.

In *Figure 4F*, parameter values used: $T = 1.8ms$, $\Delta G^{\ddagger} = 18.7k_BT$, $n_{Ca}^{\ddagger} = 3.54$, $k_0 = 1.67 \times 10^{-7}ms^{-1}$, $\tau_{RRP} = 350ms$, $\tau_{Ca} = 37ms$, $[Ca^{2+}]_0 = 50nM$ and$I_{ca} = 10\mu M$.

In *Figure 4G*, parameter values used: $T = 3ms$, $\Delta G^{\ddagger} = 18.7k_BT$, $n_{Ca}^{\ddagger} = 3.54$, $k_0 = 1.67 \times 10^{-7}ms^{-1}$, $n_{tot1} = 1000$, $\tau_{RRP} = 40ms$, $[Ca^{2+}]_0 = 50nM$ and $I_{Ca} = 10\mu M$.

In *Figure 4H*, parameter values used: $k_1 = 0.5ms$, $M = 100$.

In *Figure 4I*, parameter values used: $M = 10$, $k_1([Ca^{2+}]_a) = 0.32ms^{-1}$, $q = 10^{-1}$, $T = 2.5ms$.

In *Appendix 1—figure 2A*, parameter values are $k_0 = 2.3 \times 10^{-3}ms^{-1}$, $\Delta G^{\ddagger} = 18.5k_BT$, $n_{Ca}^{\ddagger} = 3.20$, $n_{tot1} = 500$, $n_{tot2} = 1000$, $k_2 = 0.027ms^{-1}$, $N = 2$. Parameters values used in *Appendix 1—figure 2B* are $n_{tot1} = 500$, $n_{tot2} = 1000$, $k_2 = 0.027ms^{-1}$ and $k_1 = 1ms^{-1}$ (corresponds to $[Ca^{2+}] \approx 20\mu M$ for the calyx of Held).

In *Appendix 1—figure 2C*, parameter values are $k_1 = 0.1ms^{-1}$ and $n_{tot1} = 500$.

In *Appendix 1—figure 2D*, the parameter values $\Delta G = 10k_BT$, $n_{Ca}^{\ddagger} = 1.7$ and $k_0 = 6 \times 10^{-5}ms^{-1}$ are approximately matched to the data in *Hui et al., 2005*.

In *Appendix 1—figure 2E*, the values of $k_1$ correspond to $[Ca^{2+}] = 10\mu M$ in *Appendix 1—figure 2D*.

In *Appendix 1—figure 3B*, the values for $T$, $n_{Ca}^{\ddagger}$, $n_{tot1}$, $\tau_{RRP}$, $[Ca^{2+}]_0$ and $I_{Ca}$ are as in *Figure 4G* in the main text. The values $\Delta G^{\ddagger} = 15.4k_BT$ and $k_0 = 5.45 \times 10^{-7}ms^{-1}$ are chosen such that $Ca^{2+}$-sensitivity is $\sim 1/3$ of that in *Figure 4G* while the response to a single action potential is the same.

In *Appendix 1—figure 3C*, parameter values are $\Delta G^{\ddagger} = 18.5k_BT$, $n_{Ca}^{\ddagger} = 3.54$, $k_0 = 1.67 \times 10^{-7}ms^{-1}$ and $\tau_{Ca} = 12ms$ for curve a; $\Delta G^{\ddagger} = 16.5k_BT$, $n_{Ca}^{\ddagger} = 3.14$, $k_0 = 1.4 \times 10^{-6}ms^{-1}$ and $\tau_{Ca} = 48ms$ for curve b; $\Delta G^{\ddagger} = 20k_BT$, $n_{Ca}^{\ddagger} = 4$, $k_0 = 4 \times 10^{-8}ms^{-1}$ and $\tau_{Ca} = 48ms$ for curve c. For all three curves, $T = 3ms$, $n_{tot1} = 1000$, $\tau_{RRP} = 40ms$, $[Ca^{2+}]_0 = 50nM$ and $I_{Ca} = 10\mu M$.

In *Appendix 2—figure 1*, the fit yields the height and width of the activation barrier and the rate constant for the SNARE conformational change at $[Ca^{2+}]_0$, $n_{Ca}^{\ddagger} = 3.48 \pm 0.01$ and $k_0 = (1.88 \pm 0.07) \times 10^{-7}ms^{-1}$. The fitting parameters accurately recover the input parameters of the simulations: $\Delta G^{\ddagger} = 18.4k_BT$, $n_{Ca}^{\ddagger} = 3.54$ and $k_0 = 1.67 \times 10^{-7}ms^{-1}$.

## Fitting algorithms for extracting microscopic parameters from experimental data

Data on temporal profiles of release rate and cumulative release (*Figure 3A, C*) can be used to extract the sizes of the vesicle pools, $n_{tot1}$ and $n_{tot2}$, the number of independent SNARE assemblies required for fusion, $N$, and the rate constants, $k_1$ and $k_2$. Suppose there are $m$ measured data points for the release rate or for cumulative release as a function of time: $\{(t_i, y_i)\}_{i=1}^{m}$, where $t_i$ is the time and $y_i$ is the release rate at time $t_i$ or cumulative release up to time $t_i$.

For the release rate, the target function used in the least-squares fit is *Equation 1*:

$$y(t) = a_1(1 - e^{-k_1 t})^{N-1} e^{-k_1 t} + a_2 \sum_{j=0}^{N-1} (-1)^j \binom{N-1}{j} \frac{e^{-k_2 t} - e^{-(j+1)k_1 t}}{(j+1)k_1 - k_2}, \tag{56}$$

where the coefficients $(a_1, a_2, k_1, k_2, N)$ will be determined by minimizing the squared error $\sum_{i=1}^{m} |y_i - y(t_i)|^2$. The minimization can be realized by the following code in Mathematica (alternatively, any similar program can be used):

```
data = {{t_1, y_1},{t_2, y_2},{t_3, y_3}…,{t_m, y_m}};
N = N_0;

rate =
Function[{a_1,a_2,k_1,k_2,t},
Piecewise[{{a_1 PDF[HypoexponentialDistribution[Table[ik_1,{i,1,N}]],t]
+a_2 PDF[HypoexponentialDistribution[Join[Table[ik_1,{i,1,N}],{k_2}]],t]
                                   ,t ≥ 0}},0]];
fit = NonlinearModelFit[data,rate[a_1,a_2,k_1,k_2,t],
         {{a_1,a10},{a_2,a20},{k_1,k10},{k_2,k20}},t,Method -> "NMinimize"];
fit["ParameterTable"]
Mean[Map[Abs, fit["FitResiduals"]]]
```

where $a_{10}, a_{20}, k_{10}, k_{20}$ are the initial guess values for the parameters. The code will yield the least squared error and thus the best-fit values for $(a_1, a_2, k_1, k_2)$ at $N = N_0$. The code should be run for different values of $N_0$. The set of parameters that corresponds to the minimal least squared error is then chosen (see *Appendix 3—table 1*). The sizes of the two vesicle pools can be calculated as $n_{tot1} = \frac{a_1}{Nk_1}$ and $n_{tot2} = \frac{a_2}{Nk_1k_2}$.

For the cumulative release, the target function used in the least-squares fit is *Equation 2*:

$$y(t) = b_1(1 - e^{-k_1 t})^N + b_2 \sum_{j=1}^{N} \binom{N}{j}(-1)^{j-1}(1 - \frac{jk_1 e^{-k_2 t} - k_2 e^{-jk_1 t}}{jk_1 - k_2}). \tag{57}$$

Similarly, the coefficients $(b_1, b_2, k_1, k_2, N)$ are determined by minimizing the squared error $\sum_{i=1}^{m} |y_i - y(t_i)|^2$. The corresponding Mathematica code is the same as the one above with two modifications: $(a_1, a_2)$ should be replaced with $(b_1, b_2)$ and PDF should be changed to CDF.

Data for peak release rate vs. intracellular calcium concentration (*Figure 2*) can be used to extract the parameters listed in *Appendix 3—table 2*: the activation barrier of SNARE conformational transition at reference calcium level, $\Delta G^{\ddagger}$; the average number of calcium ions bound to a SNARE assembly at the transition state, $n_{Ca}^{\ddagger}$; and the parameter combination $\left(1 - \frac{1}{N}\right)^{N-1} n_{tot1} k_0$, which measures the release rate at the reference calcium level. If $k_1$ and $N$ are extracted independently from the temporal profile of the release rate or cumulative release as discussed above, then $k_0$ can be calculated from these parameters. Rather than directly using *Equations 3 and 4* for the least squares fit, we write the logarithmic version of *Equation 4* as follows

$$R(x) = A + \frac{1}{2}\ln\left(1 - \frac{2}{3}Bx\right) + C\left(1 - \left(1 - \frac{2}{3}Bx\right)^{\frac{3}{2}}\right), \tag{58}$$

where $R$ is the logarithm of peak release rate and $x$ is the logarithm of the normalized intracellular calcium level, $\ln\frac{[Ca^{2+}]}{[Ca^{2+}]_0}$. The coefficients $A, B$, and $C$ can be determined by minimizing the squared error. Suppose the measured data points are $\{x_i, R_i\}_{i=1}^{m}$, where $x_i$ is the logarithm of the normalized intracellular calcium level and $r_i$ is the peak release rate for $x_i$. Then the following Mathematica code can be used for the least squares fit with *Equation 58*:

```
data = {{x₁,R₁},{x₂,R₂},{x₃,R₃}...,{xₘ,Rₘ}};
loglog = Function[{x,y},{Log[x/[Ca²⁺]₀],Log[y]}];
datalog = Apply[loglog, data, {1}];
```

```
peakr = Function[{A,B,C,x},Piecewise[{{A+(1/2)*Log[1-(2/3)*Bx]
                 + C*(1-(1-(2/3)*Bx)^(3/2), x<3/2/B, Infinity}}]];
```

```
fit = NonlinearModelFit[datalog, {peakr[A,B,C,x],{A > 0, B > 0, C > 0}},
          {{A,A₀},{B,B₀},{C,C₀}},x,Method -> NMinimize];
fit["ParameterTable"]
```

where $[Ca^{2+}]_0$ is the reference calcium level ($50\mu M$ in *Figure 2*) and $A_0, B_0, C_0$ are the initial guess values for the parameters. The code will yield the best-fit values for $A, B$, and $C$. The parameters for SNAREs can now be obtained as follows:

$$\frac{\Delta G^{\ddagger}}{k_B T} = C, \qquad n_{Ca}^{\ddagger} = BC, \qquad \left(1 - \frac{1}{N}\right)^{N-1} n_{tot1} k_0 = A. \tag{59}$$

