## [Editor Report]

The present manuscript describes an effort to create a general mathematical model of synaptic neurotransmission. The authors invested great efforts to create a model of the presynaptic mechanisms. This is an exceptionally challenging task and this model makes substantive progress, and highlights where further opportunities lie.

---

## [Decision Letter]

[Editors’ note: the authors submitted for reconsideration following the decision after peer review. What follows is the decision letter after the first round of review.]

Thank you for submitting your work entitled "A Theory of Synaptic Transmission" for consideration by *eLife*. Your article has been reviewed by 2 peer reviewers, and the evaluation has been overseen by a Reviewing Editor and a Senior Editor. The reviewers have opted to remain anonymous.

We are sorry to say that, after consultation with the reviewers, we have decided that your work will not be considered further for publication by *eLife*.

Given the effort and mathematical expertise of the authors, it is with considerable chagrin that I relay the comments of the reviewers and news of rejection. It is clear that the task you have undertaken is important and that your efforts are appreciated. I should emphasize that both reviewers are experts in the field and highly qualified to evaluate your study. As you can see from the reviews, there is consensus regarding the applicability to certain areas of synaptic transmission, but also concerns and limitations of the model as a general theory of synaptic transmission. We hope that the reviewer comments are helpful for the further refinement of this study and eventual publication in a more specialized journal.

*Reviewer #1:*

Wang and Dudko derive analytical equations for one special case of a model of Ca-dependent vesicle fusion, in the attempt to find a "general theory" of synaptic transmission. They use a model with 2 kinetically distinct fast and slow pools (equation 1).

Critique

1) Overall, the analytical approach applied here remains limited to the quite arbitrarily chosen 2-pool model. Thus, while the authors are able to re-capitulate the kinetics of transmitter release under a series of defined intracellular Ca-concentration steps, [Ca]i (see Figure 2B; data from Woelfel et al. 2007 J. Neuroscience), this is nevertheless not surprising because the data by Woelfel et al. was originally also fit with a 2-pool model. More importantly, the 2-pool model is valid for describing release kinetics at high [Ca]i, but it cannot account for other important phenomena of synaptic transmission like e.g. spontaneous and asynchronous release which happen at lower [Ca]i, with different Ca cooperativity (Lou et al., 2005). Along the same lines, the derivations of the equations by Wang and Dudko are not valid in the range of low [Ca]i below about 1 μM (see "private recommendations" for details). This, however, limits the applicability of the model to AP-driven transmitter release, and it shows that based on one specific arbitrarily chosen model (here: the 2-pool model), one cannot claim to build a realistic and full "theory" for synaptic transmission.

2) In their derivations, Wang and Dudko collapse the intracellular Ca-concentration [Ca]i, a parameter directly quantified in the several original experiments that went into Figure 2A, into a dimensionless relative [Ca]i "c" (see equation 7). Similarly, the release rates are collapsed into a dimensionless quantity. With these normalizations, Ca-dependent transmitter release measured in several preparations seems to fall onto a single theoretical prediction (Figure 2A). The deeper meaning behind the equalization of the data was unclear, except a demonstration that the data from these different experiments can in general be described with a two-pool model, which is at the core of the dimensionless equations. One issue might be that many of the original data sets used here derive from the same preparation (the calyx of Held), and therefore the previous data might not scatter strongly between studies. This could be clarified by the authors by also plotting the data from all studies on the non-normalized [Ca]i axis for comparison. Furthermore, it would be useful to include data from other preparations, like the inner hair cells (Beutner et al. 2001 Neuron; their Figure 3) which likely have a lower Ca-sensitivity, i.e. are right-shifted as compared to the calyx (see discussion in Woelfel and Schneggenburger 2003 J. Neuroscience). Thus, it is unclear why normalization of [Ca]i to "c" should be an advantage, because differences in the intracellular Ca sensitivity of vesicle fusion exist between synapses (see above), and likely represent important physiological differences between secretory systems.

3) Finally, the authors use their model to derive the number of SNARE proteins necessary for vesicle fusion, and they arrive at the quite strong conclusion that N = 2 SNAREs are required. Nevertheless, this estimate doesn't fit with the number of n = 4-5 ca^2+^ ions which the original studies of Figure 2A consistently found. The Ca-sensitivity at the calyx of Held, and the steepness of the release rate versus [Ca]i relation is determined by Ca-binding to Synatotagmin-2 (the specific Ca sensor isoform found at the calyx synapse), as has been determined in molecular studies at the calyx synapse (see Sun et al. 2007 Nature; Kochubey and Schneggenburger 2011 Neuron). Furthermore, in other secretory cells, the number of SNARE proteins has been estimated to be {greater than or equal to} 3 (Mohrmann et al., Science 2010).

Taken together, the derivation of the analytical equations for the kinetic scheme of a 2-pool model is mathematically interesting, and the scholarly derived equations are trustworthy. Nevertheless, the derived analytical model in fact captures only a specific stage of synaptic transmission focusing on Ca-dependent fusion of vesicles from two pools at [Ca]i >1 microM. Other important processes and mechanistic components (e.g. spontaneous, asynchronous release, Ca-dependent pool replenishment, postsynaptic factors) are either over-simplified or remained out of the scope of the theory. Therefore, the paper is far from providing a general "theory for synaptic transmission", as the title promises.

Specific Recommendations:

1) The authors discuss that the analytical equations can lead to imprecise predictions for [Ca]i > ~1 microM (see line 925). Due to the steep dependence of the rate k1 on [Ca]i (Appendix 2, Figure 1B), however, it is possible that the limiting value of 1 microM might need to be further increased. This is because otherwise, the simplifying assumptions that k1 » k2 , and that ɛ«1 (used to enable the derivations of Equations 24-29 and 31 respectively) might be violated, especially taking into account that ɛ = (k2/k1)*(ntot2/ntot1) is a function of k2 as well as the pool sizes. Although this concern was touched upon in the simulations of cumulative release (Appendix 2 Figure 2) showing that theoretical predictions cannot well describe the result of direct numerical simulations of the kinetic schemes for smaller "f" (lines 953-961; where f=1/ɛ from Equation 31), the authors should more critically assess the range of validity of the theoretical formalism in the low range of [Ca]i values as applied to the parameter space (k2, ntot2, ntot1) relevant in different synaptic preparations.

2) It is not clear from Appendix 3, which [Ca]i peak value was assumed for the simulations in Appendix 1 Figure 1B showing that for N=2 the timecourse of the release rate is more realistic. Was it Ica=10 microM, as in other simulations? If so, then the peak [Ca]i during the action potential varies between preparations and across literature, and it can as high as 25 microM. One expects that the curves in Appendix 1 Figure 1B would change if a higher peak [Ca]i was assumed, thus higher values of N would become more appropriate. This would then strongly affect the discussion and conclusions of the paper on the inferred value of N.

*Reviewer #2:*

The present manuscript describes an effort to create a general mathematical model of synaptic neurotransmission. The authors invested great efforts to create a complex model of the presynaptic mechanisms, but their approach of the postsynaptic mechanisms is way oversimplified. The authors claim that their model is consistent with lots of in vivo and in vitro experimental data, but this night be true for a small sub-selection of experimental papers (they cite 7 experimental papers regularly in the MS!). The authors also indicate that their modeling has a realistic foundation, namely they can relate some parameters in their equations to molecules/molecular mechanisms. One example is the parameter N, which they claim indicate the number of SNARE complexes requires for fusion. The reviewer finds it rather misleading because it alludes that there is a parameter for complexin, Rim1, Rim-BP, Munc13-1 etc… The equations clearly cannot formulate and reflect diversity due to different isoforms of even the above mentioned key presynaptic molecules.

The model uses a very simple equation for calculating the postsynaptic responses (Equation 8). If the reviewer understood it correctly, the postsynaptic response only depends on a constant and on the width of the spike. The width of the spike clearly affects the Ca entry, but each synapse response differently to the [ca^2+^] change. Not to talk about the way postsynaptic receptors sense the release transmitters. How would the authors reconcile data clearly showing full and incomplete postsynaptic receptor occupancy? The time course of the PSC affects the PSP just as well as the peak amplitude. Where is the kinetic of the current taken into account?

The short-term plasticity part is way oversimplified. The authors consider only two time-dependent mechanisms: residual [ca^2+^] removal rate and replenishment of the RRP. Three decades (or more) of literature demonstrates multiple mechanisms of short-term plasticity, some involve pre- and some others postynaptic changes. The reviewer cannot even see how such a simple fact is taken into the equation that the initial vesicle release probability can span two-order of magnitude. The RRP size also varies several orders of magnitudes (not only the rate of replenishment, but the size as well, which has been shown in many instances that could be a limiting factor). The presence of a special Ca sensor has been suggested that contribute to the STP. Postsynaptic receptor desensitization, diffusion of postsynaptic receptors out and in the postsynaptic density have also been shown to affect STP. None of these can be described by the model.

Modelling the 'fidelity' of the synapses is another example of oversimplification. It is well known that the probability of spike transmission varies tremendously: in some cells a large repertoire of postsynaptic active conductances contribute to the generation of suprathreshold postsynaptic responses, and in some other, the synaptic integration is passive (sublinear).

[Editors’ note: the authors submitted for reconsideration following the decision after peer review. What follows is the decision letter after the second round of review.]

Thank you for submitting the paper entitled "A Theory of Synaptic Transmission" for consideration at *eLife*. Your article and your letter of appeal have been considered by a Senior Editor, and we regret to inform you that we are upholding our original decision.

In response to your appeal, we took several steps. Most importantly, we sought the advice of an additional reviewer who was asked to review your manuscript as well as respond to the prior reviews. As you pointed out in your appeal, the approach you have taken is novel and is based in the practice of physics, not biology. The possibility that your work was not properly appreciated due to difference among the fields of physics and biology resonated with us. Thus, we sought a reviewer trained in classical physics who is also well-versed in neurobiology. This took a bit of time, but we are confident of the expertise that has been brought to bear on your paper, and we sincerely hope that the feedback provided will be useful. We are in general agreement that your work would have the most significant impact in a biological journal as opposed to a physics journal. However, all three reviewers honed in on a similar weakness that will necessitate significant revisions to the manuscript, beyond what would normally be considered appropriate for revision and re-submission at e*Life*. With this said, e*Life* might be willing to consider a future manuscript that was substantially revised along the lines of what the reviewers suggest, or along different lines based on your own interests and intuitions.

*Reviewer #3 (Recommendations for the authors):*

The manuscript by Wang and Dudko attempts something new in the arena of synaptic biophysics and modeling. Our current 'models' of synaptic vesicle fusion and plasticity have been developed over the past several decades and are generally well accepted. But, it remains possible that the acceptance of existing models is also a reflection of the fact that the people who generated the underlying data are also those who develop and test the robustness of these models. New approaches would be welcome, even if only for stimulating debate about fundamental assumptions.

The current study is potentially well suited to publication in *eLife* as open access venue that is open to publishing broadly. It is also apparent that the current study would have significantly more impact in a biological journal as opposed to a physics journal. This is clear.

I have broken my review into 'positives' and 'negatives' in the hope that this might stimulate the authors to consider how their study might be revised for eventual publication, whether at e*Life* or elsewhere.

Positives: The theoretical underpinnings of the work are strong, driven by senior physicists. The approach is a bottom-up theoretical approach to synaptic transmission that draws significantly from the author's prior work examining viral fusion. I really have very little to criticize regarding the mathematical formulation, which is clearly delineated. The success of the author's approach is nicely shown in Figure 2. I am struck by the ability of this approach to accurately fit diverse, previously published data. Given this success, I am left wondering about the underlying assumptions and which elements, when manipulated, would cause the approach to deviate from observed data. For example, the concept of 2 SNARES being minimally necessary is interesting and there are some data that might be consistent with this concept from the laboratory of Reinhardt Jan. What is the value of incorporating additional SNAREs, as is likely to occur biologically? A similar query could be made regarding the calcium-dependent cooperativity of release, a topic that has been with us since the days of Dodge and Rahamimoff. Similarly, the formulation implicitly assumes that you can sequentially activate SNAREs with no consequence with respect to the time-interval between activations and the ability to actually drive membrane fusion, implying there is also no form of cooperativity between SNARE assemblies. This is worth exploring in more detail as for example the newest models from the Rothman group imply a large cooperative action of a super-assembly of SNAREs assembled into a ring. It would be especially valuable if the stat-mech formulation they use here could "constrain" whether such super-assemblies of SNAREs is warranted in any of the existing synaptic data.

Negatives: In my view, mirroring comments by the other reviewers, the major downside of the work is that it fails to tell us something new. The approach in itself is new, but the implications of this novel approach are not realized. There is an attempt, and this is what has drawn the greatest criticism. The authors attempt to address the calcium-dependence of short-term plasticity, specifically residual calcium as it applies to short-term facilitation and post-tetanic potentiation of vesicle release. There are considerable data, from the laboratories of Wade Regehr, Erwin Neher and Ralph Schneggenburger, that argue strongly for mechanisms other than a residual calcium hypothesis. If the authors wish to pursue this topic in depth, these papers and their underlying data need to be incorporated to the level of the first half of the paper. Alternatively, could the authors pivot and remove the extension to short-term plasticity? The might, instead, focus on other underlying assumptions, perhaps with a focus on the robustness of the fusion mechanism as opposed to short term plasticity?

---

## [Author Response]

[Editors’ note: The authors appealed the original decision. What follows is the authors’ response to the first round of review.]

Reviewer #1 (Recommendations for the authors):Wang and Dudko derive analytical equations for one special case of a model of Ca-dependent vesicle fusion, in the attempt to find a "general theory" of synaptic transmission. They use a model with 2 kinetically distinct fast and slow pools (equation 1).Critique1) Overall, the analytical approach applied here remains limited to the quite arbitrarily chosen 2-pool model. Thus, while the authors are able to re-capitulate the kinetics of transmitter release under a series of defined intracellular Ca-concentration steps, [Ca]i (see Figure 2B; data from Woelfel et al. 2007 J. Neuroscience), this is nevertheless not surprising because the data by Woelfel et al. was originally also fit with a 2-pool model. More importantly, the 2-pool model is valid for describing release kinetics at high [Ca]i, but it cannot account for other important phenomena of synaptic transmission like e.g. spontaneous and asynchronous release which happen at lower [Ca]i, with different Ca cooperativity (Lou et al., 2005). Along the same lines, the derivations of the equations by Wang and Dudko are not valid in the range of low [Ca]i below about 1 μM (see "private recommendations" for details). This, however, limits the applicability of the model to AP-driven transmitter release, and it shows that based on one specific arbitrarily chosen model (here: the 2-pool model), one cannot claim to build a realistic and full "theory" for synaptic transmission.

Our two-pool description is far from being “arbitrarily chosen”. It is based on experimental facts that have been established by multiple independent laboratories: namely, the observed two distinct vesicle fusion kinetics due to the presence of the readily releasable and reserve pools in vivo and due to the presence of two dominant vesicle morphologies in vitro. The two-pool picture has been confirmed and successfully used in numerous experimental papers previously. That being said, our two-pool description refers to a more general notion of separation of timescales and is thus more flexible than a literal interpretation might suggest.

The data from Woelfel et al. 2007 J. Neuroscience, while of excellent quality, are not the only measured kinetics of the action-potential triggered vesicle fusion that our theory has been able to recapitulate (see other experimental data in Figure 2 and Figure 3 of the manuscript). The theory also recapitulates the kinetic measurements from fifteen other independent experimental studies, on ten different types of synapses. The dynamic range (peak release rate) of these synapses vary by 10 orders of magnitude, and the range of ca^2+^ concentrations span more than 3 orders of magnitude. Our work recapitulates these 16 datasets not through 16 different ad-hoc models but through a single, fully analytically solved, theoretical framework. Importantly, beyond recapitulating the existing data, our analytically tractable theory enables one to extract the unique sets of microscopic parameters for particular synapses, such as the activation energies and kinetic rates of their synaptic machinery, the sizes of the vesicle pools and the critical number of SNAREs. We verify that these predictions from our theory have reasonable values for each of the data sets; this is an additional, non-trivial check of our theory. The fact that our theory reproduces observations on such strikingly diverse systems, and has such a degree of predictive power, cannot be dismissed as an artifact or coincidence. We are not aware of any other theory, nor fitting model, of comparable generality and the ability to generate concrete predictions.

Reviewer #1 is mistaken in stating that the derivations of our equations are not valid below 1 μM ca^2+^ concentrations. It is evident already from Figure 2 that the theory performs flawlessly at concentrations as low as 0.1μM. There are indeed non-linear effects at ultra-low ca^2+^ concentrations that are not displayed by the experimental data in Figure 2. Our theory is also applicable in that regime: one simply needs to include a second coordinate (in addition to the number of ca^2+^ ions bound, *n*_*ca*_) to account for the multidimensionality of the free energy landscape, analogous to the calculations of the rate constants for multidimensional activated rate processes in chemical physics. This illustrates just one of the many ways in which our theory will enable detailed studies of mechanistic aspects of synaptic transmission.

With further regards to generality, as stated in our Abstract, this paper is concerned with providing a physical theory to describe “rapid and precise neuronal communication” enabled by “a highly synchronous release” of neurotransmitters. Typically, more than 90% of the neurotransmitters are released through synchronous release during the action potential. By applying our theory to each of the multiple Ca^2+^ sensors one will be able to cover the remaining <10% of the neurotransmitters and thus simultaneously describe spontaneous, asynchronous and synchronous release. While detailed studies of these effects are clearly beyond the scope of this work, our theory opens a door for such studies by providing a foundation in the form of a conceptual, analytically tractable framework.

2) In their derivations, Wang and Dudko collapse the intracellular Ca-concentration [Ca]i, a parameter directly quantified in the several original experiments that went into Figure 2A, into a dimensionless relative [Ca]i "c" (see equation 7). Similarly, the release rates are collapsed into a dimensionless quantity. With these normalizations, Ca-dependent transmitter release measured in several preparations seems to fall onto a single theoretical prediction (Figure 2A). The deeper meaning behind the equalization of the data was unclear, except a demonstration that the data from these different experiments can in general be described with a two-pool model, which is at the core of the dimensionless equations. One issue might be that many of the original data sets used here derive from the same preparation (the calyx of Held), and therefore the previous data might not scatter strongly between studies. This could be clarified by the authors by also plotting the data from all studies on the non-normalized [Ca]i axis for comparison. Furthermore, it would be useful to include data from other preparations, like the inner hair cells (Beutner et al. 2001 Neuron; their Figure 3) which likely have a lower Ca-sensitivity, i.e. are right-shifted as compared to the calyx (see discussion in Woelfel and Schneggenburger 2003 J. Neuroscience). Thus, it is unclear why normalization of [Ca]i to "c" should be an advantage, because differences in the intracellular Ca sensitivity of vesicle fusion exist between synapses (see above), and likely represent important physiological differences between secretory systems.

We thank the Reviewer for challenging our work with the hypothesis that the demonstrated universal scaling of the experimental data could in fact be an artefact caused by pre-selecting the data with the same preparation – addressing this hypothesis is indeed a compelling test to probe the true limits of generality of our theory. We carried out this test. We implemented the two suggestions of the Reviewer: (i) we added datasets on markedly different synaptic preparations, including the inner hair cells as suggested by the Reviewer, as well as retina bipolar cell, hippocampal mossy fiber, cerebella basket cell, chromaffin cell, insulin-secreting cell, and additional data on the calyx of Held from multiple laboratories, and (ii) we plotted the data on the nonnormalized axis of [ca^2+^] to reveal the full extent of scatter among the data sets. The resulting plot (Figure 2) speaks for itself: *in vivo* data for the release rate span 4 orders of magnitude at low [ca^2+^] and 6 orders of magnitude at high [ca^2+^], and there is a 10 orders of magnitude difference between the release rates from in vivo and in vitro data. The scatter across 4-10 orders of magnitude allows one to appreciate the vastly different sensitivities to [ca^2+^] between synaptic preparations (Figure 2, left). Yet, all these data collapse beautifully on the master curve established by our theory (Figure 2, right).

What the Reviewer refers to as “the equalization of the data” is known in statistical physics as universality. The deeper meaning of a universal scaling is its indication that the observed phenomena realized in seemingly unrelated systems are in fact governed by common physical principles. The collapse of the data onto the universal curve in Figure 2 is a demonstration that the present theory has uncovered, quantitatively, unifying physical principles underneath the striking diversity and bewildering complexity of chemical synapses. The Referee is of course correct that the differences in [ca^2+^] sensitivities among synapses likely represent important physiological differences between distinct synapses and distinct secretory systems. The present theory does not negate these differences, but it in fact allows one to quantify these differences through the unique sets of extracted parameters for individual synapses (see Appendix 3 Table 2). We are not aware of any other theory that has demonstrated universality in synaptic transmission through a simple, single scaling relation across 10 orders of magnitude in dynamic range and at the same time allowed the extraction of the microscopic parameters that are unique for the individual synapses and thus reflect the diversity of their synaptic machinery.

3) Finally, the authors use their model to derive the number of SNARE proteins necessary for vesicle fusion, and they arrive at the quite strong conclusion that N = 2 SNAREs are required. Nevertheless, this estimate doesn't fit with the number of n = 4-5 ca^2+^ ions which the original studies of Figure 2A consistently found. The Ca-sensitivity at the calyx of Held, and the steepness of the release rate versus [Ca]i relation is determined by Ca-binding to Synatotagmin-2 (the specific Ca sensor isoform found at the calyx synapse), as has been determined in molecular studies at the calyx synapse (see Sun et al. 2007 Nature; Kochubey and Schneggenburger 2011 Neuron). Furthermore, in other secretory cells, the number of SNARE proteins has been estimated to be {greater than or equal to} 3 (Mohrmann et al., Science 2010).

The Reviewer is incorrect in their claim that there is any discrepancy here. The number of SNAREs *N* and the number of ca^2+^ ions Q^‡^, extracted from the fit to our theory, are actually in a good agreement with the findings from the studies mentioned by the Reviewer. To clarify, the parameter Q^‡^ is the number of ca^2+^ ions bound to a SNARE at the transition state (not final state) of the free energy landscape of a SNARE complex. Appendix 3 Table 2 shows that, for all synaptic preparations, the extracted values at the transition state areQ^‡^< 4 - 5, which is indeed consistent with n = 4 - 5 at the final state. We note that, in addition, our theory enables one to extract the key energetic parameter that governs synaptic vesicle fusion: the activation free energy barrier ΔG^‡^ of SNARE conformational transition (in the range 8-34 k_B_T for different synaptic preparations, see Appendix 3 Table 2), which, to our knowledge, has not been possible to extract from these experiments before.

The specific value *N*=2 was extracted from a particular data set for Calyx of Held (*Woelfel et al. 2007*), for which the temporal curves of cumulative release at different ca^2+^ concentrations were available. It is quite possible that the value of *N* will be different for some other synapses. As we emphasize in the manuscript (see Discussion), the present theory does not declare the same value of *N* for all types of synapses; the power of the theory lies in providing a fitting tool for extracting this value for a system of interest.

Taken together, the derivation of the analytical equations for the kinetic scheme of a 2-pool model is mathematically interesting, and the scholarly derived equations are trustworthy. Nevertheless, the derived analytical model in fact captures only a specific stage of synaptic transmission focusing on Ca-dependent fusion of vesicles from two pools at [Ca]i >1 microM. Other important processes and mechanistic components (e.g. spontaneous, asynchronous release, Ca-dependent pool replenishment, postsynaptic factors) are either over-simplified or remained out of the scope of the theory. Therefore, the paper is far from providing a general "theory for synaptic transmission", as the title promises.

We appreciate that the Reviewer sees our analytical derivations as being mathematically interesting, scholarly derived, and trustworthy. We believe that we have convincingly refuted the Reviewer’s criticisms regarding perceived limitations. We have shown that our universal scaling and collapse is not limited to high calcium concentrations and have presented checks using data from vastly different synaptic preparations. As noted above, the generality of a theory is determined not by the amount of details packed in it but by the ability of the theory to reproduce observations and generate predictions regarding the phenomenon of interest (here: rapid and precise neuronal communication) while containing as few details as possible. Our theory accomplishes just that; it delivers precisely what our title promises.

Specific Recommendations:1) The authors discuss that the analytical equations can lead to imprecise predictions for [Ca]i > ~1 microM (see line 925). Due to the steep dependence of the rate k1 on [Ca]i (Appendix 2, Figure 1B), however, it is possible that the limiting value of 1 microM might need to be further increased. This is because otherwise, the simplifying assumptions that k1 » k2, and that ɛ«1 (used to enable the derivations of equations 24-29 and 31 respectively) might be violated, especially taking into account that ɛ = (k2/k1)*(ntot2/ntot1) is a function of k2 as well as the pool sizes. Although this concern was touched upon in the simulations of cumulative release (Appendix 2 Figure 2) showing that theoretical predictions cannot well describe the result of direct numerical simulations of the kinetic schemes for smaller "f" (lines 953-961; where f=1/ɛ from Equation 31), the authors should more critically assess the range of validity of the theoretical formalism in the low range of [Ca]i values as applied to the parameter space (k2, ntot2, ntot1) relevant in different synaptic preparations.

The Reviewer appears to have confused Appendix 2 Figure 2 in the original manuscript (testing the accuracy of the theory in the light of the finite capacity of the readily releasable pool) and the derivation of the peak release rate in Equation (7) (Equations (28)-(33)). In the original manuscript, we have already demonstrated that the theory is valid and accurate with respect to the finite capacity of the readily releasable pool (see Appendix 2 Figure 3A in the revised manuscript, which corresponds to Appendix 2 Figure 2A in the original manuscript). In the revised Appendix (Appendix 2 Figure 2) we now assess the validity of the analytical expression in Equation (7) as suggested by the Reviewer. As demonstrated in Appendix 2;figure 2, we found that, for the biologically relevant ratio ntot1ntot2∼1 and in the range of calcium concentrations [Ca^2+^] < 0.1μM, the analytic expression in Equation 7 is exceedingly accurate (>95%), with the parameters returned by the fit being within less than 5% from the exact parameters used in the simulations. Appendix 2;figure 2 further shows that the accuracy of Equation (7) doesn’t depend significantly on the value of k_2_. Additionally, we found that, even for the ratio ntot1ntot2∼0.5 (possibly too small to be physiologically relevant), the parameters extracted from Equation (7) still have > 90% accuracy.

2) It is not clear from Appendix 3, which [Ca]i peak value was assumed for the simulations in Appendix 1 Figure 1B showing that for N=2 the timecourse of the release rate is more realistic. Was it Ica=10 microM, as in other simulations? If so, then the peak [Ca]i during the action potential varies between preparations and across literature, and it can as high as 25 microM. One expects that the curves in Appendix 1 Figure 1B would change if a higher peak [Ca]i was assumed, thus higher values of N would become more appropriate. This would then strongly affect the discussion and conclusions of the paper on the inferred value of N.

The [Ca^2+^] peak value in Appendix 1 Figure 1B is 20uM (we indicated this value in the revised manuscript). For reference, this value corresponds to a rate constant, k1∼1ms−1 for Calyx of Held. We would like to emphasize that, when inferring the value of N from fitting the experimental data in (Woelfel, Lou and Schneggenburger 2007 J. Neuroscience) to our theory, we did not make any assumptions about the peak [Ca^2+^]. Rather, we directly used the values of [Ca^2+^] that are indicated in Figure 4 of their paper. In general, the value of N will indeed depend on the [Ca^2+^] peak value in a given experiment as the Reviewer pointed out, and the theory will naturally account for this dependence when it is applied to each experiment.

Reviewer #2 (Recommendations for the authors):The present manuscript describes an effort to create a general mathematical model of synaptic neurotransmission. The authors invested great efforts to create a complex model of the presynaptic mechanisms, but their approach of the postsynaptic mechanisms is way oversimplified. The authors claim that their model is consistent with lots of in vivo and in vitro experimental data, but this night be true for a small sub-selection of experimental papers (they cite 7 experimental papers regularly in the MS!). The authors also indicate that their modeling has a realistic foundation, namely they can relate some parameters in their equations to molecules/molecular mechanisms. One example is the parameter N, which they claim indicate the number of SNARE complexes requires for fusion. The reviewer finds it rather misleading because it alludes that there is a parameter for complexin, Rim1, Rim-BP, Munc13-1 etc… The equations clearly cannot formulate and reflect diversity due to different isoforms of even the above mentioned key presynaptic molecules.

We appreciate that the Reviewer found 7 different experimental papers – covering different synapses and different experimental setups – to be “a small sub-selection”. We believe that Figure 2 , which uses 16 different experimental papers, leaves no further doubts that the claims about the consistency between the theory and data are fully justified. Despite up to 10 orders of magnitude variation in the release rate of different synaptic preparations and more than 3 orders of magnitude range of calcium concentrations (Figure 2, left), all the data collapse onto a universal curve predicted by our theory (Figure 2, right). These data represent different systems – from the central nervous system to the secretory system – and come from in vivo and in vitro experiments. The data we have used cover the measurements on all synaptic systems that we could find in the literature on the action potential-driven neurotransmitter release. If the Reviewer is aware of any existing data on other synaptic systems that we might have missed, we will gratefully appreciate the opportunity to apply the theory to those data as well.

The diversity of the molecular components in different synapses is captured in our theory through different values of the microscopic parameters*ΔG^‡^, n*^‡^_*Ca*_ and k_0._ These parameters describe, respectively, the activation energy barrier, the number of bound *Ca^2+^* ions, and the intrinsic rate of the conformational transition of the SNARE complexes that drive synaptic vesicle fusion in a given synapse. Different isoforms of the individual components of SNARE complexes and scaffold proteins, including the proteins mentioned by the Reviewer, will be reflected in different values of*ΔG^‡^, n*^‡^_*Ca*_ and k_0._ for specific synaptic preparations, as can be seen in Appendix 3 Table 2 in the manuscript. These parameters capture the energetic and kinetic properties of the synaptic fusion machinery as a complex rather than as a collection of isolated molecules. Because the molecular components within a SNARE complex act collectively (hence the name ;complex;) to drive vesicle fusion, it is natural (and indeed fortunate) that the predictive power of the theory can be preserved with only a few key parameters of the molecular machinery as opposed to requiring a long list of parameters for every specific isoform of each of the many individual molecular components.

The model uses a very simple equation for calculating the postsynaptic responses (Equation 8). If the reviewer understood it correctly, the postsynaptic response only depends on a constant and on the width of the spike. The width of the spike clearly affects the Ca entry, but each synapse response differently to the [ca^2+^] change. Not to talk about the way postsynaptic receptors sense the release transmitters. How would the authors reconcile data clearly showing full and incomplete postsynaptic receptor occupancy? The time course of the PSC affects the PSP just as well as the peak amplitude. Where is the kinetic of the current taken into account?

The Reviewer did not understand it correctly. The postsynaptic response (Equation 8) in our theory depends not only on the width of the spike, but also on other presynaptic factors, including the Ca^2+^-sensitivity due to different Ca^2+^ sensors in SNAREs (captured through *ΔG^‡^, n*^‡^_*Ca*_ and k_0_ in the calcium-dependent rate *k*_1_) and the sizes of the reserve and readily-releasable vesicle pools, n*_tot1_* and n*_tot2_*. These dependencies can be found in the expression for the cumulative release < n(t) > in Equation 3. We chose to suppress these dependencies in Equation8 to avoid a cumbersome notation < n(T, ΔG^‡^, *n*^‡^_*Ca*_, k_0_, k_2_, N, n*_tot1_*, n*_tot2_*)>. We revised the text following Equation 8 in the manuscript to remind the readers that various presynaptic factors enter the postsynaptic response through the cumulative release < n(t) >.

The parameter γ in Equation 8 embodies the postsynaptic factors. It will depend on the type and density of the postsynaptic receptors and the availability of binding site on the receptors. While studying different types, densities, and occupancies of postsynaptic receptors in detail is not the focus of this manuscript, it is straightforward to incorporate these effects in Equation 8, including different forms of more complicated nonlinear dose-response curves for postsynaptic receptors. This is just one of the many ways in which the present theory opens doors to detailed quantitative studies of specific aspects of synaptic transmission.

We certainly do take into account the kinetics of the postsynaptic current: details of this can be found in the section “Peak postsynaptic current and cumulative release”, please see the derivation in Equations 41-43. This section discusses how the conductance of an ion channel of a given type, the postsynaptic membrane potential, and the reversal potential of the ion corresponding to the ion channel determine the time course of postsynaptic current.

We note that all the experimental studies on the kinetics of the action potential-driven neurotransmitter release to which we applied our theory have already taken into account the fact of postsynaptic receptor desensitization. This means that the experimental data can be directly compared to our theory at the level of neurotransmitter release (Equation 3) rather than postsynaptic response. Nevertheless, while it is not the goal of the present work to incorporate every known detail of the synaptic transmission, this work provides a quantitative, conceptual framework within which one could model, in detail, particular aspects of synaptic transmission, including different scenarios of postsynaptic response.

The short-term plasticity part is way oversimplified. The authors consider only two time-dependent mechanisms: residual [ca^2+^] removal rate and replenishment of the RRP. Three decades (or more) of literature demonstrates multiple mechanisms of short-term plasticity, some involve pre- and some others postynaptic changes. The reviewer cannot even see how such a simple fact is taken into the equation that the initial vesicle release probability can span two-order of magnitude. The RRP size also varies several orders of magnitudes (not only the rate of replenishment, but the size as well, which has been shown in many instances that could be a limiting factor). The presence of a special Ca sensor has been suggested that contribute to the STP. Postsynaptic receptor desensitization, diffusion of postsynaptic receptors out and in the postsynaptic density have also been shown to affect STP. None of these can be described by the model.

This is incorrect. Our theory can describe different Ca^2+^ sensors, different size of vesicle pools, as well as the Ca^2+^ entry. Specifically, different Ca^2+^ sensors with different Ca^2+^-sensitivities are reflected in different values of *ΔG^‡^, n*^‡^_*Ca*_ and k_0_. The Ca^2+^ sensitivity of a SNARE complex in our manuscript is defined as the ratio of the conformational rate constants (Equation 5) when [Ca^2+^] increases by a fixed amount: k1([Ca2+]0+1Ca)k1([Ca2+]0) (see Appendix 1 section “Calcium and RRP vesicle kinetics in short-term plasticity”). Different sizes of vesicle pools are accounted for through n*_tot1_* and n*_tot2_*, and there is nothing in the theory that forbids these factors from varying by orders of magnitude. As we illustrate in Appendix 1—figure 3, the variations in these factors, in turn, can give rise to significantly different initial vesicle release probability. In particular, in Appendix 1—figure 3A, we demonstrate the ability of the theory to account for the fact the initial vesicle release probability can span two orders of magnitude (significantly more, in fact), which was questioned by the Reviewer.

With respect to the short-term plasticity, not only our theory can describe the effects of Ca^2+^ removal rate, RRP replenishment rate, and different Ca^2+^-sensitivities caused by different Ca^2+^ sensors, but we in fact investigated these effects in Appendix 1 Figure 3.

Modelling the 'fidelity' of the synapses is another example of oversimplification. It is well known that the probability of spike transmission varies tremendously: in some cells a large repertoire of postsynaptic active conductances contribute to the generation of suprathreshold postsynaptic responses, and in some other, the synaptic integration is passive (sublinear).

The probability of spike transmission predicted from our theory (Equation 10) can vary tremendously, and it can do so through different RRP size (n_tot1_), different presynaptic Ca^2+^ sensors (through different values of *ΔG^‡^, n*^‡^_*Ca*_ and k_0_), and different widths of [Ca^2+^] profile. While a systematic study of different types of synaptic integration within the framework of our theory deserves a separate paper (likely a series of papers, in fact), we illustrate how our theory can bridge the gap between molecular mechanisms and synaptic function in the section “Transmission rate vs. fidelity” by applying the theory to a case where the generation of postsynaptic response is due to suprathreshold neurotransmitter release.

[Editors’ note: what follows is the authors’ response to the second round of review.]

In response to your appeal, we took several steps. Most importantly, we sought the advice of an additional reviewer who was asked to review your manuscript as well as respond to the prior reviews. As you pointed out in your appeal, the approach you have taken is novel and is based in the practice of physics, not biology. The possibility that your work was not properly appreciated due to difference among the fields of physics and biology resonated with us. Thus, we sought a reviewer trained in classical physics who is also well-versed in neurobiology. This took a bit of time, but we are confident of the expertise that has been brought to bear on your paper, and we sincerely hope that the feedback provided will be useful. We are in general agreement that your work would have the most significant impact in a biological journal as opposed to a physics journal. However, all three reviewers honed in on a similar weakness that will necessitate significant revisions to the manuscript, beyond what would normally be considered appropriate for revision and re-submission at eLife. With this said, eLife might be willing to consider a future manuscript that was substantially revised along the lines of what the reviewers suggest, or along different lines based on your own interests and intuitions.Reviewer #3 (Recommendations for the authors):The manuscript by Wang and Dudko attempts something new in the arena of synaptic biophysics and modeling. Our current 'models' of synaptic vesicle fusion and plasticity have been developed over the past several decades and are generally well accepted. But, it remains possible that the acceptance of existing models is also a reflection of the fact that the people who generated the underlying data are also those who develop and test the robustness of these models. New approaches would be welcome, even if only for stimulating debate about fundamental assumptions.The current study is potentially well suited to publication in eLife as open access venue that is open to publishing broadly. It is also apparent that the current study would have significantly more impact in a biological journal as opposed to a physics journal. This is clear.I have broken my review into 'positives' and 'negatives' in the hope that this might stimulate the authors to consider how their study might be revised for eventual publication, whether at eLife or elsewhere.Positives: The theoretical underpinnings of the work are strong, driven by senior physicists. The approach is a bottom-up theoretical approach to synaptic transmission that draws significantly from the author's prior work examining viral fusion. I really have very little to criticize regarding the mathematical formulation, which is clearly delineated. The success of the author's approach is nicely shown in Figure 2. I am struck by the ability of this approach to accurately fit diverse, previously published data. Given this success, I am left wondering about the underlying assumptions and which elements, when manipulated, would cause the approach to deviate from observed data. For example, the concept of 2 SNARES being minimally necessary is interesting and there are some data that might be consistent with this concept from the laboratory of Reinhardt Jan. What is the value of incorporating additional SNAREs, as is likely to occur biologically? A similar query could be made regarding the calcium-dependent cooperativity of release, a topic that has been with us since the days of Dodge and Rahamimoff. Similarly, the formulation implicitly assumes that you can sequentially activate SNAREs with no consequence with respect to the time-interval between activations and the ability to actually drive membrane fusion, implying there is also no form of cooperativity between SNARE assemblies. This is worth exploring in more detail as for example the newest models from the Rothman group imply a large cooperative action of a super-assembly of SNAREs assembled into a ring. It would be especially valuable if the stat-mech formulation they use here could "constrain" whether such super-assemblies of SNAREs is warranted in any of the existing synaptic data.Negatives: In my view, mirroring comments by the other reviewers, the major downside of the work is that it fails to tell us something new. The approach in itself is new, but the implications of this novel approach are not realized. There is an attempt, and this is what has drawn the greatest criticism. The authors attempt to address the calcium-dependence of short-term plasticity, specifically residual calcium as it applies to short-term facilitation and post-tetanic potentiation of vesicle release. There are considerable data, from the laboratories of Wade Regehr, Erwin Neher and Ralph Schneggenburger, that argue strongly for mechanisms other than a residual calcium hypothesis. If the authors wish to pursue this topic in depth, these papers and their underlying data need to be incorporated to the level of the first half of the paper. Alternatively, could the authors pivot and remove the extension to short-term plasticity? The might, instead, focus on other underlying assumptions, perhaps with a focus on the robustness of the fusion mechanism as opposed to short term plasticity?

Thank you for your thoughtful and efficient review of our submission and for the transparent and constructive feedback.

We are glad that Reviewer #3 was satisfied with our revisions made in response to their comments, that you have consulted two theorists, and that they found the paper potentially interesting. We followed your recommendations (1, 2, 3) and made the following changes to the manuscript:

(1) As far as you can, you try to explain what you have achieved in a way that biologists can understand. This will increase the impact of your paper (and render it more in line with what eLife usually publish)

We revised the manuscript, particularly the Introduction and Discussion, to highlight for biologists the value of our approach in analyzing and interpreting experimental data. Moreover (in line with the related question from the Senior Editor), we included in the Appendix the codes for the fitting algorithms, allowing experimentalists to fit their data with our expressions and thereby determine microscopic parameters from their data. We hope that this will facilitate the use of our results by experimentalists and increase the impact of the paper.

Below we summarize the results of our work that should appeal specifically to biologists:

– Our theory for the action-potential-evoked neurotransmitter release gives analytic expressions for the key measurable characteristics of the neurotransmitter release, including (i) the temporal profile of the release rate, (ii) the relationship between peak release rate and intracellular calcium concentration, and (iii) the effect of cooperativity among SNARE complexes on the release rate. These expressions provide a means to extract the microscopic parameters of the synapses through a simple least-squares fit. As we illustrated with existing data sets, such fits yield the activation barriers and rates of SNARE conformational transitions, the size of vesicle pools, and the number of independent SNARE assemblies necessary for fusion. The theory moreover allowed us to identify the presence of cooperative SNARE assemblies (“superassemblies”) in several existing data sets. Our theory and the fitting algorithm can be similarly applied to future experimental data to extract these new interpretations and implications.

– Our application of the theory to short-term plasticity and, in particular, two major mechanisms of synaptic facilitations (syt7-mediated facilitation and buffer saturation), shows quantitative agreement with the experimental data for the paired-pulse ratio on a variety of synapses. We clarified and identified the regimes where previously proposed facilitation mechanisms fail to explain the data quantitatively. We hope that these clarifications will stimulate further experimental analysis of the regimes and the determination of the underlying facilitation mechanisms. Furthermore, the theory provides a trade-off relationship between the transmission rate and fidelity, as well as the condition for faithful synaptic transmission without the need for the fine-tuning.

– The applicability of our theory is more general than the subset of key features of synaptic transmission that we investigated. The theory opens a door for future detailed studies of neurotransmission by providing a conceptual, quantitative, and fully analytically tractable framework that can be directly connected to experiments via the least-squares fitting of the data to our analytical results.

(2) Address the comments about the scope of claims the limitations of the theory with respect to extant data.

We agree about the importance of clearly stating the assumptions, and we have revised the manuscript to enhance the clarity of our assumptions. At the same time, we are confident that our assumptions are realistic, and that our approach delivers what it claims.

Χ2 commented that “endocytosis (vesicle recycling) is also calcium-dependent and quite critically determines short term depression – this is not included in their model”. We note that our treatment of the vesicle replenishment rate k_2_ as a constant is supported by previous experiments (e.g., Wolfel et al., 2007) which found that the sensitivity of vesicle replenishment (k_2_) to the intracellular calcium concentration is much weaker than that of SNARE conformational transition (k_1_). In response to a tetanic stimulus, where the asynchronous component of the release becomes dominant, the calcium-dependence of k_2_ may indeed no longer be negligible. However, our work focused on the action-potential-evoked release (known as synchronous release), as stated in the abstract. Nevertheless, the theory can be readily extended to incorporate the calcium-dependence of k_2_ explicitly, and such an extension will allow the extraction of the parameters for post-tetanic potentiation from the data.

Χ2 further commented that “They only show paired pulse experiments, yet claim the model explains short term plasticity generally…”. We limited the discussion of the short-term plasticity to the two mechanisms of synaptic facilitation and to the data on the paired-pulse ratio as illustrative examples, but other mechanisms can be explored in an analogous manner. For example, spike-broadening effects (Cho et al., 2020) and calcium-dependent vesicle recycling (Marks and McMahon, 1998) can be incorporated into the theory by introducing variations in T and k_2_, respectively.

To clarify the scope of our theory, we now made the discussion of the limitations of our approach more prominent in the revised manuscript. The limitations of the theory itself, and routes to extend the theory to overcome these limitations, are described in the extensive paragraph in Discussion that starts with “The theory presented here has several limitations” (Line 500). The limitations of the functional implications of the theory are described in the section “From molecular mechanisms to synaptic functions” in Discussion.

We revised the abstract and conclusion to alert the reader that the limitations are discussed in the manuscript. Specifically, in the abstract, we indicated that our theory was compared to data for the paired-pulse ratio, and added: “We discuss the limitations of the theory and propose possible routes to extend it”. An analogous statement is added in the conclusion.

(3) "it would be useful for the authors to clarify how this approach allows for the unambiguous "extraction of the kinetic and energetic parameters for SNARE complexes".

As we state in the abstract and throughout the paper, our approach allows the extraction of the parameters through a fit of the derived equations to experimental data. Most biologists should be generally familiar with the notion of fitting data to a model as a means to extract parameters from their data. As we mentioned in response to point 1 above, to facilitate experimentalists utilizing our approach, we now provide (at the end of Appendix 3) the complete algorithms and codes used in our fitting of the data. Anyone can simply copy and paste these codes in Mathematica or any other similar programs to readily perform the fit of their data. This least-squares fit will return the parameter values for the activation barriers and rates of SNARE conformational transitions (at any calcium concentration), the size of vesicle pools, and the number of independent SNARE assemblies necessary for fusion. We think that a tool for extracting these parameters from the exceedingly rich data generated by previous experiments has been missing. Providing such a tool constitutes the most valuable practical aspect of our theory. We thank the Senior Editor for the opportunity to clarify this point. We hope that the added codes and detailed instructions for fitting the data will facilitate the use of our approach by experimentalists so that they can extract from their data the wealth of valuable information that would otherwise remain hidden.